# Stratospheric aerosol characteristics from SCIAMACHY limb observations: 2-parameter retrieval

Christine Pohl[1], Felix Wrana[2], Alexei Rozanov[1], Terry Deshler[3], Elizaveta Malinina[4], Christian von Savigny[2], Landon A. Rieger[5], Adam E. Bourassa[5], and John P. Burrows[1]

[1]Institute of Environmental Physics, University of Bremen, Bremen, Germany
[2]Institute of Physics, University of Greifswald, Greifswald, Germany
[3]Department of Atmospheric Science, University of Wyoming, Laramie, Wyoming, USA
[4]Canadian Centre for Climate Modeling and Analysis, Environment and Climate Change Canada, Victoria, British Columbia, Canada
[5]Institute of Space and Atmospheric Studies, University of Saskatchewan, Saskatoon, Canada

**Correspondence:** Christine Pohl (cpohl@iup.physik.uni-bremen.de)

**Abstract.**

Stratospheric aerosols play a key role in atmospheric chemistry and climate. Their particle size is a crucial factor controlling the microphysical, radiative, and chemical aerosol processes in the stratosphere. Despite its importance, available observations on aerosol particle size are rather sparse. This limits our understanding and knowledge about the mechanisms and importance of chemical and climate aerosol feedbacks. The retrieval described by Malinina et al. (2018) provides the stratospheric particle size distribution (PSD) from SCIAMACHY limb observations in the tropics. This algorithm has now been improved and extended to work on the entire globe. Two PSD parameters of a unimodal lognormal PSD, the median radius and the geometric standard deviation, are retrieved between 18 and 35 km altitude from SCIAMACHY limb observations by a multi-wavelength non-linear regularized inversion. The approach assumes an aerosol particle number density profile that does not change during the retrieval. The effective Lambertian surface albedo pre-retrieved from coinciding SCIAMACHY nadir observations is integrated into the retrieval algorithm to mitigate the influence of the surface albedo on the retrieval results. The extinction coefficient and the effective radius are calculated from the PSD parameters. The aerosol characteristics from SCIAMACHY are compared with in-situ balloon-borne measurements from Laramie, Wyoming, and retrievals from the satellite instruments SAGE II, SAGE III, and OSIRIS. In the northern hemisphere, the median radius differs by less than 27 % and the geometric standard deviation by less than 11 % from both balloon-borne and SAGE III data. Differences are mainly attributed to errors in the assumed a priori number density profile. Globally, the SCIAMACHY extinction coefficients at 750 nm deviate by less than 35 % from SAGE II, SAGE III, and OSIRIS data. The effective radii from SCIAMACHY, balloon-borne measurements, and SAGE III agree within about 18 % while the effective radius based on SAGE II measurements is systematically larger. The novel data set containing the PSD parameters, the effective radius, and the aerosol extinction coefficient at 525, 750, and 1020 nm from SCIAMACHY observations is publicly available.

# 1   Introduction

Stratospheric aerosols are well known to play a key role in atmospheric chemistry and climate (Kremser et al., 2016). They form a distinct layer, the so-called "Junge layer", in the altitude range between 15 and 35 km. The maximum concentration is typically around 20 km. This aerosol layer consists mainly of hydrated sulfuric acid supplemented by small amounts of meteorite and other non-sulfate particles (Kremser et al., 2016).

The background aerosol loading in the stratosphere is in a quasi-stationary equilibrium due to continuous natural and anthropogenic emissions of OCS and $SO_2$ as well as evaporation and sedimentation processes, with seasonal and quasi-biennial oscillations (Deshler et al., 2006). Occasionally, the background state is overlaid by aerosols originating from transient but sulfur-rich volcanic eruptions (McCormick et al., 1995; Andersson et al., 2015; Friberg et al., 2018; Kloss et al., 2021) and biomass burning events (Siddaway and Petelina, 2011; Bourassa et al., 2019; Ohneiser et al., 2020; Das et al., 2021). The aerosols entrained in the stratosphere are dispersed on a global scale by advection and self-generated thermal convection. Microphysical processes such as nucleation, coagulation, and condensation control the spatiotemporal variation of the total aerosol number concentration and size distribution. Aerosols remain in the stratosphere for several months to years until evaporation at the top of the aerosol layer (above 32-35 km altitude) and sedimentation processes return the stratospheric aerosol level to background conditions (Kremser et al., 2016).

Volcanic and wildfire perturbations significantly increase the stratospheric aerosol optical depth (SAOD) on a global scale, which have a short-term impact on the Earth's radiative budget (Malinina et al., 2021; Sellitto et al., 2022). The enhanced stratospheric short-wave scattering and long-wave absorption typically force a cooling of the surface and a heating of the stratosphere. This triggers a series of responses in the internal dynamics responsible for the climate system, which are summarized in Marshall et al. (2022). Among others, investigations have noted changes in the North Atlantic Circulation (Hermanson et al., 2020), El Niño-Southern Oscillation (Khodri et al., 2017), Atlantic Meridional Overturning Circulation (Pausata et al., 2015), atmospheric dynamics (Toohey et al., 2014; Wallis et al., 2023), and quasi-biennial oscillation (DallaSanta et al., 2021). Furthermore, there is evidence of weaker monsoons (Liu et al., 2016), reduced precipitation (Iles et al., 2013), and a shifted Inter-Tropical Convergence Zone (Colose et al., 2016).

Even small amounts of aerosols play a significant role as catalysts in stratospheric chemistry. Heterogeneous chemical reactions on the aerosol droplet surface increase the amount of reactive chlorine and hydrogen oxide radicals as well as reduce the amount of nitrogen oxide (NO, $NO_2$) in the surrounding atmosphere (Fahey et al., 1993; Solomon et al., 1996). These reactions lead to an imbalance in the photochemical cycle of ozone loss and production and thus to ozone depletion. Furthermore, stratospheric aerosols serve as condensation nuclei for polar stratospheric cloud formation (Ebert et al., 2016).

The climatic and chemical significance of stratospheric aerosols makes an accurate knowledge of the microphysical and radiative aerosol properties important for the assessment of aerosol feedback mechanisms. This knowledge is obtained from in-situ (balloon and aircraft), ground-based, and satellite measurements, which are summarized and discussed in, e. g., Thomason and Peter (2006) and Kremser et al. (2016).

The observations of stratospheric aerosols are used for several purposes. Analysing measurements allows conclusions to be drawn about the evolution and interaction of stratospheric aerosols. Information obtained from aerosol observations contributes to the development of aerosol microphysical modules (Vignati et al., 2004), aerosol transport schemes (Grieser and Schönwiese, 1999), and volcanic forcing emulators (Toohey et al., 2016; Aubry et al., 2020). The observed aerosol characteristics are summarized in aerosol climatologies (e. g., Mills et al., 2016; Thomason et al., 2018; Kovilakam et al., 2020), which are used to create volcanic aerosol forcing sets (Stenchikov et al., 1998; Arfeuille et al., 2013; Sato et al., 2016). Those can be subsequently utilized in climate impact studies (Toohey et al., 2014; Brühl et al., 2015). Furthermore, aerosol observations are required to test the reliability of climate models. Recent studies show discrepancies between observations and simulations (e. g., Chylek et al., 2020; Tejedor et al., 2021), indicating that knowledge about stratospheric aerosols is still incomplete. Accordingly, aerosol observations are used to adjust or constrain aerosol plumes in climate models (Das et al., 2021; Schallock et al., 2023) to enable more realistic simulations and more accurate estimates of aerosol radiative forcing. In addition, observations of post-volcanic aerosol distributions provide new insights into effectiveness and impacts of potential geoengineering concepts (Robock et al., 2013).

The aerosol particle size is one of the main parameters in model simulations as it controls the microphysical, radiative, and chemical aerosol processes (Kremser et al., 2016). The knowledge of particle size is therefore a key factor in describing stratospheric aerosol evolution and associated climate response. Despite its importance, available observations on aerosol particle size are rather limited. Valuable in-situ (Deshler et al., 2019), airborne (McLinden et al., 1999), and ground-based measurements (Ugolnikov and Maslov, 2018; Zalach et al., 2019) of aerosol sizes are rare and localised. Retrievals of the size distribution from the Stratospheric Aerosol and Gas Experiment (SAGE) series data (Bingen et al., 2004; Wurl et al., 2010; Damadeo et al., 2013; Wrana et al., 2021) are global albeit the occultation measurements suffer from a rather coarse spatial sampling and limited coverage. Aerosol particle size data sets with dense spatial sampling were firstly obtained from the Optical Spectrograph and InfraRed Imager System (OSIRIS) (Bourassa et al., 2008; Rieger et al., 2014) and the Scanning Imaging Absorption Spectrometer for Atmospheric Chartography (SCIAMACHY) limb observations (Malinina et al., 2018). The latter data set is restricted to the tropics, while the former data set is no longer updated.

In this study, the algorithm from Malinina et al. (2018) developed to derive the stratospheric aerosol particle size distribution (PSD) in the tropics is extended to work on the entire globe. Similar to Malinina et al. (2018), the median radius and the geometric standard deviation of a unimodal lognormal distribution are retrieved from SCIAMACHY limb scatter observations. A number density profile is assumed that does not change during the retrieval. To mitigate the influence of surface reflection on the retrieval results, the retrieval uses the effective Lambertian surface albedo pre-retrieved from coinciding SCIAMACHY nadir measurements. From the retrieved PSD parameters, the effective radius and the extinction coefficient are calculated. The resulting data set contains the PSD parameters, as well as the effective radius and the extinction coefficient at 525, 750, and 1020 nm for the entire globe. The SCIAMACHY limb radiance in the southern hemisphere is less sensitive to PSD parameters making it difficult to retrieve them separately (see Sect. 6).

The manuscript is structured as follows: After an introduction to the relevant aerosol characteristics in Sect. 2, the SCIA-MACHY instrument and the retrieval algorithm are described in Sects. 3 and 4. Comparison data sets and satellite instruments

are presented in Sect. 5. In Sect. 6, the retrieval performance is investigated using a synthetically generated data set. The SCIAMACHY-derived aerosol characteristics are compared with balloon-borne measurements and satellite data products in Sect. 7. The comparison results are discussed in Sect. 8, followed by a conclusion in Sect. 9.

## 2 Stratospheric aerosol characteristics

The PSD of stratospheric aerosols is typically represented by a lognormal distribution, which has a different number of modes depending on the application (Deshler et al., 2003; Brühl et al., 2012; von Savigny and Hoffmann, 2020). Retrievals from space-borne measurements usually adopt a unimodal lognormal PSD (e.g., Bingen et al., 2004; Rieger et al., 2014; Malinina et al., 2018; Wrana et al., 2021):

$$\frac{\mathrm{d}n}{\mathrm{d}r} = \frac{N}{\sqrt{2\pi}\ln\sigma_{\mathrm{g}}\,r}\exp\left(-\frac{(\ln r - \ln r_g)^2}{2\ln^2\sigma_{\mathrm{g}}}\right), \tag{1}$$

where the number density $n$ of particles with radius $r$ is defined by the total number density $N$, the geometric mean or median radius $r_g$, and the geometric standard deviation $\sigma_{\mathrm{g}}$. The assumption of a unimodal PSD is advantageous because it describes the prevailing PSD relatively well with only three degrees of freedom.

Since $r_{\mathrm{g}}$ and $\sigma_{\mathrm{g}}$ represent the maximum and dispersion of the PSD in the logarithmic space, it is more convenient to use the mode radius $R_{\mathrm{mod}}$ and the width $w$ instead, which represent the maximum and spread of the PSD in the linear space:

$$R_{\mathrm{mod}} = \frac{r_{\mathrm{g}}}{\exp(\ln^2\sigma_{\mathrm{g}})}, \tag{2}$$

$$w^2 = [\exp(\ln^2\sigma_{\mathrm{g}}) - 1]\exp(2\ln r_g + \ln^2\sigma_{\mathrm{g}}). \tag{3}$$

The square of the width, $w^2$, is also called arithmetic variance. Nevertheless, we will refer to $r_{\mathrm{g}}$ and $\sigma_{\mathrm{g}}$ in this paper, since both products are direct results of the SCIAMACHY retrieval algorithm (Sect. 4) and independent of each other.

Another prevalent parameter to describe the aerosol particle size is the effective radius. It is defined by the ratio of the third to the second moment of the PSD, i.e., the ratio of the total particle volume $V$ to the total particle surface area $A$ per unit volume times 3:

$$r_{\mathrm{eff}} = \frac{3V}{A} = \frac{\int r\,\pi\,r^2\,\frac{\mathrm{d}n}{\mathrm{d}r}\,\mathrm{d}r}{\int \pi\,r^2\,\frac{\mathrm{d}n}{\mathrm{d}r}\,\mathrm{d}r}. \tag{4}$$

The right hand side of Eq. (4) indicates that $r_{\mathrm{eff}}$ is a weighted average with the cross sectional area as the weighting factor. Although, $r_{\mathrm{eff}}$ does not represent any PSD characteristics, it is useful in scattering optics. As the particle size determines the scattering probability, $r_{\mathrm{eff}}$ indicates a typical aerosol particle size in the prevailing scattered radiation field. Different PSDs can have identical effective radii and scattering properties. The effective radius thus unifies the results from stratospheric aerosol particle size retrievals independent of their PSD-based assumptions and facilitates their comparison (Mishchenko and Travis, 1997). For a unimodal lognormal PSD, $r_{\mathrm{eff}}$ is related to $r_{\mathrm{g}}$ and $\sigma_{\mathrm{g}}$ by:

$$r_{\mathrm{eff}} = r_{\mathrm{g}}\exp(2.5\ln^2\sigma_{\mathrm{g}}). \tag{5}$$

The optical properties of aerosols can be characterized by, e. g., their extinction coefficient. It is defined as:

$$Ext(\lambda) = \int \beta_{\mathrm{aer}}(r, \lambda, \mathfrak{m}) \frac{\mathrm{d}n}{\mathrm{d}r} \, \mathrm{d}r, \tag{6}$$

where $\beta_{\mathrm{aer}}$ is the aerosol extinction cross section calculated by Mie theory and $\mathfrak{m}$ is the aerosol refractive index at wavelength $\lambda$. The ratio of aerosol extinction coefficients at two different wavelengths provides the Ångstrom exponent $\alpha$ (Ångström, 1929):

$$\frac{Ext(\lambda_1)}{Ext(\lambda_2)} = \left( \frac{\lambda_1}{\lambda_2} \right)^{-\alpha}. \tag{7}$$

The Ångstrom exponent $\alpha$ is only an approximate measure of the aerosol particle size because it depends on the choice of wavelength pair. Furthermore, an infinite number of PSD parameter combinations can result in the same $\alpha$ (Malinina et al., 2019). Thus, the Ångstrom exponent in Eq. (7) is only used to calculate the extinction coefficients for desired wavelengths. Note that these kind of calculations may be subject to uncertainties for the reasons mentioned above.

## 3   SCIAMACHY observations

The Scanning Imaging Absorption Spectrometer for Atmospheric Chartography (SCIAMACHY) was operated aboard the European Environmental Satellite (Envisat). It was launched on 1 March 2002 into a sun-synchronous orbit at an altitude of about 800 km with a local descending node equator crossing time of 10 am. SCIAMACHY's measurements started in August 2002 and ended in April 2012 after communications with Envisat were lost.

SCIAMACHY measured the scattered solar radiance in limb and nadir geometry, the attenuated solar and lunar radiance in occultation geometry as well as the extraterrestrial solar irradiance and lunar radiance. The radiance in the limb and nadir geometry as well as the solar irradiance are used for the PSD retrieval in this study. Limb and nadir measurements were performed alternately on the day side of the orbit, the solar irradiance once per orbit. The radiation was detected by a grating spectrometer in eight wavelength channels covering the spectral range between 214 and 2386 nm. The spectral resolution depends on the wavelength and is between 0.2 and 1.5 nm.

In the nadir scan mode, the instrument observed the Earth's scenery below the satellite using a whisk-broom scanning system. Each orbital scan has a swath width of 960 km, i. e., a field of view of $\pm 32\,^\circ$. Typically, it consists of four scan targets (pixels) per whisk-broom line at viewing zenith angles (VZAs) of about $\pm 9°$ and $\pm 26\,°$. The footprint size of each pixel depends on the scan speed, measurement integration time, and observation geometry and is usually 30 km along-track and 240 km (60 km – best case) across-track. After 13 whisk-broom lines were recorded, SCIAMACHY switched to the limb viewing geometry.

In the limb scan mode, the instrument observed the scattered solar radiance tangential to the Earth's surface at tangent heights between -3 and about 100 km in steps of 3.3 km. The radiation was typically sampled at four different viewing azimuth angles at a constant elevation angle before the optics tilted to the next scanning elevation. The viewing azimuth angles were carefully chosen in order to match the geographic location of the limb scatter measurement with the individual scenes of subsequent nadir measurements. In total, four measurement profiles of spectral radiances were recorded with a vertical resolution of 2.6 km

and a horizontal resolution of about 400 km along-track and 240 km across-track. This results in a total swath width of 960 km. One limb observation sequence usually lasted 60 s.

According to the scanning geometry and sampling, SCIAMACHY achieved global coverage at the equator after 6 days. Further information about the instrument is provided in, e. g., Burrows et al. (1995), Bovensmann et al. (1999), and Gottwald and Bovensmann (2010).

## 4 SCIAMACHY version 2.0 aerosol PSD retrieval

The stratospheric aerosol characteristics are derived using the radiative transfer model SCIATRAN 4.1 (Rozanov et al., 2014; Mei et al., 2023). The model was run for an atmosphere-surface system. The surface was Lambertian with a surface albedo of 0.5 as a first guess. Atmospheric pressure and temperature profiles prevailing at the location and time of each SCIAMACHY observation are based on ERA-Interim reanalysis produced by the European Centre for Medium-Range Weather Forecasts (ECMWF).

Stratospheric aerosols are assumed to be located at altitudes between 12 and 46 km. They are specified as a mixture of 75 % sulphuric acid and 25 % water. Their refractive indices are based on the OPAC database with an assumed relative humidity of 0 % (Hess et al., 1998). The scattering phase functions as well as extinction and scattering coefficients are calculated employing Mie theory.

Both, the aerosol composition and the relative humidity, are idealistic assumptions. The percentage of sulphuric acid can vary slightly in reality (Turco et al., 1982; Steele et al., 2003; Doeringer et al., 2012). The Atmospheric Chemistry Experiment Fourier Transform Spectrometer (ACE-FTS) even occasionally detected sulphuric acid levels of less than 50 % after the Raikoke eruption 2019 (Boone et al., 2022). The stratospheric relative humidity is usually between 0 % («1 %) and 10 % (Steele and Hamill, 1981). However, we stick to the above-mentioned aerosol composition and relative humidity because the OPAC database does not offer more realistic compositions. The resulting retrieval uncertainty was estimated by comparing retrieved PSD parameters assuming a relative humidity of 0 % and 80 %. The latter value is exceedingly high, but allows a maximum uncertainty estimate of below 15 % for the mode radius and below 10 % for the geometric standard deviation (not shown). These values can also be regarded as an uncertainty estimate due to an incorrect aerosol composition. By increasing the relative humidity, the particles absorb water vapour, which reduces the percentage of sulphuric acid. As a result, the aerosol refractive index (Palmer and Williams, 1975) changes with a similar amplitude to that of an increase in relative humidity (Hess et al., 1998).

A vertically constant aerosol size is used as an initial condition. The mode radius is set to $R_{\mathrm{mod}} = 0.11\,\mu\mathrm{m}$. Note the convention used here – the retrieval is controlled externally by $R_{\mathrm{mod}}$ and not by $r_{\mathrm{g}}$. The geometric standard deviation is set to $\sigma_{\mathrm{g}} = 1.37$. Both values are based on balloon-borne measurements at background aerosol loadings (Deshler, 2008). If not stated otherwise, a number density profile $N_{\mathrm{ECSTRA}}$ based on the ECSTRA model climatology for aerosol background conditions (Fussen and Bingen, 1999) is assumed. It decreases exponentially from $22.83\,\mathrm{cm}^{-3}$ at 12 km to $0.03\,\mathrm{cm}^{-3}$ at 46 km altitude.

Malinina et al. (2018) retrieved the stratospheric PSD and the effective Lambertian surface albedo simultaneously to account for the influence of the surface reflectance on the measured limb radiance. However, the algorithm cannot reliably distinguish whether, e. g., an increase in the measurement signal is caused by a higher surface albedo or by a stronger scattering of stratospheric aerosols. Therefore, we split the aerosol retrieval algorithm in two steps: In the first retrieval step, the effective Lambertian surface albedo is retrieved from coinciding SCIAMACHY nadir radiances. Here, two action items have to be pointed out. First, since each nadir scan contains 13 whisk-broom lines of four pixels each, the nadir radiances are initially averaged along the track over all pixels of one nadir scan at constant VZA. This results in four averaged radiance spectra at VZAs of about $\pm 9°$ and $\pm 26°$ from which the effective surface albedo is retrieved. Second, 'coincident' means that the geographical locations of limb scatter and averaged nadir measurements may only differ by a maximum of 223 km. This corresponds to a latitudinal width of about 2 °. Otherwise, no retrieval is performed at all.

The retrieved effective surface albedo is used as a first guess in the second retrieval step. Here, vertical profiles of $r_{\mathrm{g}}$ and $\sigma_{\mathrm{g}}$ are retrieved from SCIAMACHY limb radiances. The surface albedo is additionally adjusted in this retrieval step to mitigate errors in the retrieved aerosol parameters that may arise from the incorrect assumption of a Lambertian surface. While $r_{\mathrm{g}}$ and $\sigma_{\mathrm{g}}$ are derived, $N$ remains unchanged at the initial profile for two reasons. First, the spectral signatures of the three parameters are strongly correlated. Changes in measured limb radiances can be largely described by adjusting only two PSD parameters. The third PSD parameter usually provides only little additional information. That means, a multitude of aerosol PSD profiles result in the similar measured limb radiance. Fixing one PSD parameter restricts this ambiguity and gives more weight to the other two PSD parameters when responding to the given limb radiance. Second, among the three PSD parameters, a change of $N$ has the least effect on the resulting limb radiance (Malinina et al., 2018). As a result, uncertainties caused by a fixed $N$ profile have less influence on the PSD retrieval than if $r_{\mathrm{g}}$ or $\sigma_{\mathrm{g}}$ are kept constant.

Both retrieval steps are based on the linearization of the forward model $\mathbf{F}(\mathbf{x})$ around an initial guess state $\mathbf{x_0}$ (Rodgers, 2000):

$$\mathbf{y} - \mathbf{y_0} = \mathbf{K}(\mathbf{x} - \mathbf{x_0}) + \boldsymbol{\epsilon}, \tag{8}$$

where $\mathbf{y}$ and $\mathbf{y_0}$ are the measurement and initial guess vectors, $\mathbf{K}$ is the weighting function or Jacobian matrix, $\mathbf{x}$ is the state vector, and $\boldsymbol{\epsilon}$ is the noise vector containing the modelling, measurement, and linearization errors.

The measurement vector $\mathbf{y}$ contains the logarithms of sun-normalized SCIAMACHY radiances, averaged in the six wavelength ranges $748 - 752$, $805 - 809$, $868 - 872$, $1088 - 1092$, $1225 - 1245$, and $1294 - 1306$ nm. Other wavelengths are not taken into account because radiation at shorter wavelengths or between the selected wavelength bands is too strongly influenced by Rayleigh scattering and molecular absorption and radiation at longer wavelengths has too low signal-to-noise ratios. According to the retrieval procedure, SCIAMACHY averaged radiances from the nadir geometry are used in the first step, SCIAMACHY radiances from the limb geometry are used in the second step. Here, only measured limb radiances between 18 and 35 km altitude are considered. The radiation from lower altitudes is too strongly influenced by scattering from molecules, clouds, and tropospheric aerosol contaminations while radiation above 35 km is influenced by stray-light.

The initial guess vector $\mathbf{y_0}$ contains the corresponding logarithms of sun-normalized radiances simulated by the radiative transfer model SCIATRAN 4.1 for the initial state vector $\mathbf{x_0}$.

The state vectors $\mathbf{x_0}$ and $\mathbf{x}$ contain the a priori quantities and the quantities to be retrieved, respectively. In the first step, these are the effective Lambertian surface albedo values at the six wavelength bands mentioned above. In the second step, the state vectors contain the vertical profiles of $r_g$ and $\sigma_g$ between 18 and 35 km altitude as well as the effective Lambertian surface albedo at the six wavelength bands. Above 35 km, a vertically constant PSD profile is assumed with $R_{\mathrm{mod}} = 0.11\,\mu\mathrm{m}$ and $\sigma_g = 1.37$. Below 18 km, the PSD profile is scaled with the same factor as the lowermost retrieval tangent height. The aerosol parameterizations outside the altitude range of $18 - 35$ km might be inadequate. However, they avoid unphysical aerosol size parameters in the lowermost (18 km) and uppermost retrieval height (35 km). Additionally, they have only a minor influence on the aerosol size parameters to be retrieved in between (Malinina et al., 2018).

The weighting function matrix or Jacobian matrix $\mathbf{K}$ contains the partial derivatives of the forward model operator with respect to each state vector element (retrieval quantity).

To obtain the state vector $\mathbf{x}$ in Eq. (8), we do not follow the maximum a posteriori method of Rodgers (2000), which assumes a fixed a priori state vector $\mathbf{x_0}$. Instead, we invert Eq. (8) analogous to Malinina et al. (2018) by the weighted regularized approach based on the zeroth-order Tikhonov method. In this approach, the initial guess state $\mathbf{x_0}$ in iteration $n + 1$ is replaced by the state vector $\mathbf{x}$ obtained at the previous iteration $n$. This enables the final result to move far away from a priori values while strongly constraining each particular iterative step. The latter is necessary because of a strong non-linearity of the inverse problem and a correlation between the retrieval parameters. We minimize the weighted norm:

$$\|\mathbf{y} - \mathbf{y_n}\|_{\mathbf{S_y^{-1}}}^2 + \|\mathbf{x} - \mathbf{x_n}\|_{\mathbf{S_a^{-1}}}^2 , \tag{9}$$

where the noise covariance matrix $\mathbf{S_y}$ and the a priori covariance matrix $\mathbf{S_a}$ are the weight matrices. Their contents depend on the retrieval step and are explained in detail below. This optimization method leads to the state vector solution:

$$\mathbf{x_{n+1}} = \mathbf{x_n} + \left(\mathbf{K_n^\top} \mathbf{S_y^{-1}} \mathbf{K_n} + \mathbf{S_a^{-1}}\right)^{-1} \mathbf{K_n^\top} \mathbf{S_y^{-1}} (\mathbf{y} - \mathbf{y_n}), \tag{10}$$

where $\mathbf{K_n} = \mathbf{K}(\mathbf{x_n})$ and $\mathbf{y_n} = \mathbf{F}(\mathbf{x_n})$.

In the albedo retrieval (first step), the a priori covariance matrix and the noise covariance matrix are chosen to be diagonal, i. e., no correlation between the radiances or the albedo of different wavelengths is assumed. The diagonal elements are set to $1\mathrm{e}{-2}$ for the a priori covariance and to the signal-to-noise ratio of 1000 for the noise covariance matrix.

Since three species are retrieved in the second step ($r_g$, $\sigma_g$, and the surface albedo), the a priori covariance matrix has to be set for each species individually. In case of $r_g$ and $\sigma_g$, the covariance matrix elements are calculated by:

$$[\mathbf{S_a}]_{j,k} = \varsigma^2 \exp\left(-\frac{|z_j - z_k|}{r_c}\right) , \tag{11}$$

where $r_c$ is the correlation radius that is set to 3.3 km and $z_j$ and $z_k$ are the altitudes corresponding to the element $(j, k)$. The variances are set to $\varsigma^2 = 2.5\mathrm{e}{-7}\,\mu\mathrm{m}^2$ in case of $r_g$ and to $\varsigma^2 = 2.5\mathrm{e}{-7}$ in case of $\sigma_g$. The values are selected as a trade-off between the numerical stability and a priori sensitivity. Note that instead of relative variances (Malinina et al., 2018), we use absolute values to retain a constant variance within the iterative process.

The a priori covariance matrix of the albedo in the second retrieval step is a diagonal matrix. Each diagonal entry responds to one of the six wavelength bands considered, i.e., the albedo is spectrally uncorrelated. The variances, i.e., the diagonal elements, are set to $\varsigma^2 = 1\mathrm{e}-6$. This is four orders of magnitude smaller than the variance in the first step because the surface albedo is only an auxiliary retrieval quantity. It only serves to correct errors in the pre-retrieved surface albedo resulting from the Lambertian surface assumption.

The total a priori covariance matrix results from the composition of the individual matrices responsible for $r_\mathrm{g}$, $\sigma_\mathrm{g}$, and the surface albedo $a$:

$$
\mathbf{S_a} = \begin{bmatrix} \mathbf{S_a}^{r_\mathrm{g}} & \mathbf{0} & \mathbf{0} \\ \mathbf{0} & \mathbf{S_a}^{\sigma_\mathrm{g}} & \mathbf{0} \\ \mathbf{0} & \mathbf{0} & \mathbf{S_a}^{a} \end{bmatrix} .
\tag{12}
$$

The zero sub-matrices $\mathbf{0}$ are of appropriate sizes to provide no correlation between the retrieval parameters.

In the absence of better knowledge, the noise covariance matrix is assumed to be diagonal in the second step, i.e., the noise is spectrally and spatially uncorrelated. Since the influence of stray light below 35 km is small, this assumption should not have a negative impact on the retrieval. The diagonal elements contain the signal-to-noise ratios, which are estimated from the SCIAMACHY measurements. A second order polynomial is fitted to the radiances of each considered wavelength band and the noise level is calculated from the fit residuals.

Within the iterative process, $R_\mathrm{mod}$ and the surface albedo cannot become smaller than 0.05 μm and 0.015, respectively. The radius limit is lower than the sensitivity limit of SCIAMACHY (Malinina et al., 2019). Surface albedo values below 0.015 usually do not occur. Both limits are chosen to avoid unphysical results. No limits are set for $\sigma_\mathrm{g}$.

Two convergence criteria are selected to terminate the iterative algorithm: either the root mean square deviation between all simulated and measured radiances considered within the retrieval changes by less than 0.1 % or each state vector element (retrieval quantities) changes by less than 1 % in two consecutive iterations. If both criteria are not fulfilled, the algorithm is aborted after 30 iterations.

Finally, the retrieved PSD parameters and the assumed number density are used to calculate the effective radius (Eq. (5)) and the extinction coefficient (Eq. (6)) of the aerosol particles. We calculate the aerosol extinction coefficient at 750 nm to make it comparable with our previous SCIAMACHY v1.4 $Ext$ product (Rieger et al., 2018). For the public, we also calculate the aerosol extinction coefficient at 525 and 1020 nm by Eq. (6) to enable a comparison with other satellite aerosol data products.

Note that both the retrieved ($r_\mathrm{g}$, $\sigma_\mathrm{g}$) and the calculated ($r_\mathrm{eff}$, $Ext$) parameters are slightly dependent on the choice of the a priori number density profile. However, it will be shown in Sect. 6 that the strong correlation between the PSD parameters can compensate for retrieval errors in the calculated parameters, provided that the a priori number density profile does not deviate considerably from the true profile.

## 5 Reference aerosol data products

The evaluation of the SCIAMACHY v2.0 aerosol PSD retrieval is based on data sets from balloon-borne measurements, as well as SAGE II, SAGE III-M3M, and OSIRIS observations. The instruments and retrieval approaches are briefly introduced below.

### 5.1 Optical particle counter measurements

Deshler et al. (2003, 2019) provide a long-term record of vertical PSD profiles above Laramie, Wyoming (41°N, 106°W). The PSD profiles cover the altitude range from 15 to 33 km with a vertical sampling of 0.5 km. The data are publicly available at Deshler (2023). They were obtained by balloon-borne optical particle counters (OPCs). The measurement time series began in 1971, and over time, the instruments, calibration factors, and the approach retrieving the PSDs have been improved (Kovilakam and Deshler, 2015; Deshler et al., 2019). The data recorded during SCIAMACHY's operational period were mainly provided by the OPC device based on 40 ° scattering geometry and a flow rate of 10 L/min. The number of particles is measured in 12 size bins. The instrument itself is only sensitive to particles with radii between 0.15 and 10.0 μm. Smaller particles with a size of more than 0.01 μm are first enlarged to the optical detection threshold by a connected supersaturation chamber before the total number density is recorded by a second OPC. The measurements are subsequently fitted by either a unimodal (Eq. (1)) or bimodal lognormal size distribution by minimizing the root mean squared logarithmic difference between the fit-function and bin-sized number density measurements (Deshler et al., 2003). The unimodal PSDs are used for comparison with SCIAMACHY-derived aerosol characteristics. From these, the aerosol extinction coefficients are calculated according to Eq. (6).

### 5.2 SAGE II

The Stratospheric Aerosol and Gas Experiment II (SAGE II) instrument operated aboard the Earth Radiation Budget Satellite (ERBS) between October 1984 and August 2005. It was launched on 5 October 1984 into a 57 ° inclination orbit at an altitude of 610 km.

SAGE II measured the solar irradiance attenuated by the Earth's atmosphere at each sunset and sunrise encountered by the instrument using the solar occultation technique. While the instrument was moving, the measurements were performed at different tangent altitudes ranging from cloud top to around 60 km with a vertical resolution of 0.5 km and a horizontal resolution of 2.5 x 200 km$^2$. The irradiance was measured in seven channels with center wavelengths between 385 and 1020 nm and bandwidths between 2 and 20 nm.

Determined by the satellite orbit and observation technique, SAGE II provided about 30 irradiance profiles per day, evenly distributed every 24 ° longitude with a gradual change in the latitude between 80 °N – 80 °S. Further information about the instrument is provided in, e. g., Mauldin III et al. (1985) and McCormick (1987).

The aerosol extinction coefficients and effective radii retrieved from sunset measurements by the SAGE II version 7.0 algorithm are used for the comparison. The algorithm developed by NASA (National Aeronautics and Space Administration;

Damadeo et al., 2013) converts the spectral sun-normalized observations into optical depth profiles for individual trace gases and aerosols. From those, the vertical profiles of gas concentrations and aerosol extinction coefficients at 386, 452, 525 and 1020 nm are calculated using an onion-peeling technique.

The retrieval of the effective radius is described in Thomason et al. (2008, Method 1) with additional explanations in Kovilakam and Deshler (2015) and Reeves et al. (2008). The effective radius is calculated by assuming an aerosol composition of two different aerosol particle sizes with a total number density of $20\,cm^{-3}$. From that composition, the minimum and maximum values of the surface area density and the aerosol volume density are derived using the 525 and 1020 nm extinction coefficients. The means of minimum and maximum values are then used to calculate the effective radius according to Eq. (4).

In addition to the effective radii provided in the SAGE II version 7.0 data set, we also retrieve the effective radius based on a dual-wavelength extinction (DWE) ratio. The DWE approach is based on a method described in Yue and Deepak (1983). For that, a unimodal lognormal size distribution with $\sigma_g = 1.5$ is assumed. The median radius is retrieved by comparing the 525 to 1020-nm extinction ratios from SAGE II version 7.0 with a lookup table consisting of extinction ratios calculated for the same wavelengths using Mie routines (Oxford University, 2022). By using the extinction ratio, the unknown number density cancels out from the retrieval which can subsequently be calculated from the retrieved median radius, the assumed geometric standard deviation, and the extinction coefficient (Eq. (6)). The effective radius is then calculated using Eq. (5).

We chose the extinction coefficients at 525 and 1020 nm because they have the lowest uncertainties of the four available extinction coefficients over almost all altitude levels. Additionally, this selection allows to retrieve a unique median radius in the largest possible radius range. At other wavelengths, Mie resonances may result in extinction ratios yielding multiple possible median radius solutions, especially for median radii larger than roughly 0.425 µm.

The Mie calculations are performed for spherical droplets consisting of 75 % sulphuric acid and 25 % water. The real refractive index is taken from Palmer and Williams (1975), corrected for temperature using Lorentz-Lorenz-corrections as described by Steele and Hamill (1981). The imaginary refractive index is set to zero, i. e., no absorption.

## 5.3 SAGE III-M3M

The Stratospheric Aerosol and Gas Experiment III (SAGE III) was operational on the Russian Meteor-3M (M3M) satellite from 2002 to 2005. The spacecraft was launched on 10 December 2001 into a sun-synchronous orbit at an altitude of 1020 km with an inclination of 99.5 ° and an ascending node equatorial crossing time of 9:15 am (Roberts et al., 1996).

The SAGE III-M3M instrument performed solar occultation measurements from 0.5 to 100 km altitude. The solar irradiance was measured by a grating spectrometer at 86 wavelengths from 280 to 1040 nm with a spectral resolution of 1 to 2 nm. An Indium Gallium Arsenide photodiode additionally measured the irradiance at 1550 nm with a bandwidth of 30 nm. A horizontal slit limited the field of view at the tangent height location to 0.7 km in the vertical and to 1.5 km in the horizontal direction. The orbit of the satellite resulted in sunrise measurements being confined within roughly 60 °S and 35 °S and sunset measurements occurring roughly between 40 °N and 80 °N (McCormick and Chu, 2004; Thomason et al., 2010).

The SAGE III version 4.0 algorithm provides aerosol extinction coefficient profiles at nine wavelengths between 384 and 1544 nm at 90 altitude levels from the Earth's surface to an altitude of 45 km (Thomason et al., 2010). The algorithm for processing the measurements works similarly to the SAGE II version 7.0 algorithm.

The PSD parameters are retrieved by a method similar to the SAGE II DWE approach described above. For that, SAGE III version 4.0 extinction ratios are compared with those from a lookup table calculated by Mie theory. However, due to the
broad wavelength range covered by the nine available wavelengths, the use of extinction coefficients at three wavelengths is feasible. Two sets of extinction ratios, namely 449 to 756 nm and 1544 to 756 nm, are used instead of only one, creating a two-dimensional field of median radii and geometric standard deviations for combinations of the two extinction ratios. Thus, the geometric standard deviation does not have to be assumed, but can instead be retrieved simultaneously with the median radius. The effective radius is then calculated from Eq. (5). The method has already been successfully applied to SAGE III
measurements aboard the International Space Station (ISS) and is described in detail in Wrana et al. (2021). In analogy to the DWE algorithm, it is referred to as the triple-wavelength extinction (TWE) ratio approach in the following.

## 5.4 OSIRIS version 7.2

The Optical Spectrograph and InfraRed Imager System (OSIRIS) operates aboard the Odin satellite. It was launched on 20 February 2001 into a sun-synchronous orbit at an altitude of 610 km with a local descending node equator crossing time of
6 am. OSIRIS has been measuring the scattered radiance of the terrestrial limb since November 2001.

The relevant data comes from the optical spectrograph. It is a grating spectrometer that covers the wavelength range from 280 to 810 nm with a spectral resolution of about 1 nm. The horizontal slit in the entrance optics limits the field of view at the limb tangent height to 40 km in the horizontal and to 1 km in the vertical direction. By continuously tilting the entire satellite, OSIRIS scans the limb from 7 km to 75 km with a sampling interval of about 2 km, depending on the measurement integration
time. One profile scan takes about 40 seconds and covers approximately 400 km along the satellite track at the ground. Sunlit observations in the mid and high latitudes are not available in the winter hemisphere. Global coverage from 82 °N to 82 °S is only achieved in spring and autumn. Further information about the instrument is provided in Warshaw et al. (1996), Llewellyn et al. (2004), and McLinden et al. (2012).

This paper uses the OSIRIS version 7.2 aerosol extinction coefficients at 750 nm (Rieger et al., 2019). The algorithm is a
multi-wavelength retrieval assuming a fixed unimodal lognormal PSD with $r_\mathrm{g} = 0.08\,\mu$m and $\sigma_\mathrm{g} = 1.6$. During the iterative procedure, the number density is retrieved and converted to the extinction coefficient using the assumed particle size. The measurement vector contains reference-height-normalized limb measurements. These are additionally normalized by radiances of a variable wavelength combination to reduce noise and decrease sensitivity to the PSD assumptions.

## 6 Sensitivity tests

The retrieval of the aerosol PSD parameters $r_\mathrm{g}$ and $\sigma_g$ from SCIAMACHY observations described in Sect. 4 is based on two assumptions. First, the surface is assumed to be Lambertian. Second, the number density profile is fixed at the beginning of the

retrieval and is unalterable. Both assumptions do not have to be correct for any given SCIAMACHY observation. Therefore, we test the retrieval for its sensitivity to both assumptions using a synthetic measurement set.

This synthetic data set has been created by using SCIATRAN. It contains the nadir and limb radiances for the illumination and observation geometries of one randomly chosen SCIAMACHY orbit. The time of the orbit is irrelevant, just as there is no need for multiple orbits to account for seasonality. The reason lies in the single-scattering angle, the crucial angle of the illumination and observation geometry that affects the quality of the retrieval. Since this angle varies more within an orbit than per year at constant latitude, the seasonal range of the single-scattering angle – and therefore the seasonal dependency of the retrieval – is covered by the variation of the single-scattering angle along an orbit.

The nadir and limb radiances are simulated assuming a vegetated surface. Its anisotropic reflectance is defined by the bidirectional reflectance distribution function (BRDF). The BRDF describes the scattering of the incident irradiance from an infinitesimal solid angle into the infinitesimal solid angle of another direction (Schaepman-Strub et al., 2006). The BRDF is calculated by the Ross-Li semi-empirical model (Lucht et al., 2000) with parameters $f_{\mathrm{iso\,/\,vol\,/\,geo}} = 0.36\,/\,0.24\,/\,0.03$. The values are based on the Moderate Resolution Imaging Spectroradiometer (MODIS) MCD43A1 data set over a vegetated surface of Amazonia (Lorente et al., 2018). The simulated surface is characterized by an enhanced backscattering and a lowered forward-scattering reflectance. At a solar zenith angle of $35\,°$, the BRDF ranges between 0.09 and 0.12 at VZA of $26\,°$ and between 0.10 and 0.11 at VZA of $9\,°$. The black-sky albedo, as defined by the albedo of the surface that is illuminated only by the sun without any atmospheric contribution, is 0.33. Larger solar zenith angles entail a higher anisotropy and albedo values.

The simulated atmosphere contains aerosols between 12 and $46\,\mathrm{km}$ altitude. Their PSD profiles are specified below (Tab. 1 for testing the sensitivity to the Lambertian surface assumption, Fig. 2 black lines for testing the sensitivity to the aerosol number density). Pressure and temperature profiles as well as the extraterrestrial solar irradiance are chosen in accordance with the randomly selected SCIAMACHY orbit.

### 6.1 Sensitivity to Lambertian surface assumption

To investigate the influence of the Lambertian surface assumption on the retrieved aerosol PSD parameters, the nadir and limb radiances are simulated with four different aerosol loads: "small", "background", "unperturbed", and "volcanic". The cases are selected similar to Malinina et al. (2018). The PSD paramaters $R_{\mathrm{mod}}$, $r_{\mathrm{g}}$, and $\sigma_{\mathrm{g}}$ are constant with altitude and are summarized in Table 1. In the "unperturbed" case, the PSD parameters are the same as the a priori values. The number density profile is based on balloon-borne measurements over Wyoming (Deshler et al., 2019). The profile is shown later in Fig. 2. This number density profile is also used as a priori in the subsequent retrieval procedure.

The simulated nadir radiances are used to derive the effective Lambertian surface albedo. The nadir measurements observing the surface closer to its forward-scattering region slightly underestimate the correct surface albedo while nadir measurements observing the surface closer to its backscattering region overestimate the correct surface albedo. The error increases from VZA of 9 to $26\,°$ and is almost independent of the aerosol amount.

**Table 1.** Selected scenarios of aerosol loads (adapted from Malinina et al. (2018)).

| Aerosol load | $R_{\mathrm{mod}}$, μm | $r_{\mathrm{g}}$, μm | $\sigma_{\mathrm{g}}$ | $w$, μm | SAOD |
|---|---|---|---|---|---|
| Small | 0.060 | 0.080 | 1.700 | 0.052 | 0.017 |
| Background | 0.080 | 0.100 | 1.600 | 0.055 | 0.027 |
| Unperturbed | 0.110 | 0.121 | 1.370 | 0.041 | 0.021 |
| Volcanic | 0.200 | 0.207 | 1.200 | 0.039 | 0.130 |

In the second step, the retrieved albedo is used as the a priori Lambertian surface albedo to retrieve $r_{\mathrm{g}}$ and $\sigma_{\mathrm{g}}$ from simulated limb radiances. Unlike the description in Sect. 4, the albedo retrieval is switched off in this step in order to better investigate the effects of an incorrectly assumed Lambertian surface.

Figure 1 shows the true and derived aerosol characteristics for the four studied aerosol loads (colours). The results are shown along the orbit, i. e., as a function of the event number. Except $Ext$, aerosol characteristics at different retrieval heights are presented by the colour tints. Since the true values of $Ext$ depends on the altitude due to the $N$ profile (Eq. (6)), we decide to show the results of $Ext$ only for an altitude of 21.8 km. Left panels contain the aerosol characteristics of the profiles at the western edge of the swath, while right panels contain the aerosol characteristics of the profiles at the eastern edge of the swath. We selected the profiles at the edges of the swath as corresponding nadir measurements have VZAs of $\pm 26\,°$. For those limb profiles, the errors in the a priori surface albedo are the largest and thus have the strongest influence on the PSD retrieval. SCIAMACHY moved southwards, i. e., with an increasing event number the considered aerosol profile is located further south and the single-scattering angle becomes larger. We chose this special format for presentation to emphasize how the accuracy of the retrieval behaves with different single scattering angles.

The aerosol characteristics are correctly derived up to a single-scattering angle of about 96 $°$. This angular range corresponds to latitudes north of 26 $°$N in summer and 23 $°$S in winter. The relative deviation between derived and true aerosol characteristics, i. e., the relative error, is usually less than 10 % in case of $r_{\mathrm{g}}$ and $Ext$ and even less than 5 % in case of $\sigma_{\mathrm{g}}$ and $r_{\mathrm{eff}}$. Limb radiances at single-scattering angles greater than 96 $°$ are less sensitive to aerosols due to much smaller phase function values (Rieger et al., 2014, 2019). This makes it difficult to retrieve the PSD parameters separately which leads to more incorrect retrieval values for $r_{\mathrm{g}}$ and $\sigma_{\mathrm{g}}$. Their relative errors point in opposite directions: while $r_{\mathrm{g}}$ is overestimated, $\sigma_{\mathrm{g}}$ is underestimated or vice versa. Interestingly – and probably for this reason –, $Ext$ and $r_{\mathrm{eff}}$ calculated from $r_{\mathrm{g}}$ and $\sigma_{\mathrm{g}}$, are estimated similarly well as at single-scattering angles below 96 $°$, except for retrievals at high aerosol loads. The local maximum in $Ext$ and $r_{\mathrm{eff}}$ at single-scattering angles of about 85 $°$ (Fig. 1 (f, h)) is due to the defined limitation of $r_{\mathrm{g}}$ (Sect. 4). In this angular range, $r_{\mathrm{g}}$ should be lower than the limit of $R_{\mathrm{mod}} = 0.05\,\mu m$, but is held back during the retrieval: The retrieval parameter is set to a value that corresponds to the average of the lower retrieval limit and the mode radius from the previous iteration.

The retrieved $r_{\mathrm{g}}$, $\sigma_{\mathrm{g}}$, and calculated $r_{\mathrm{eff}}$ values are mostly constant with altitude as intended. An incorrect altitude dependency can only be observed for single scattering angles greater than 96 $°$, when the retrieval error increases. Here, aerosol

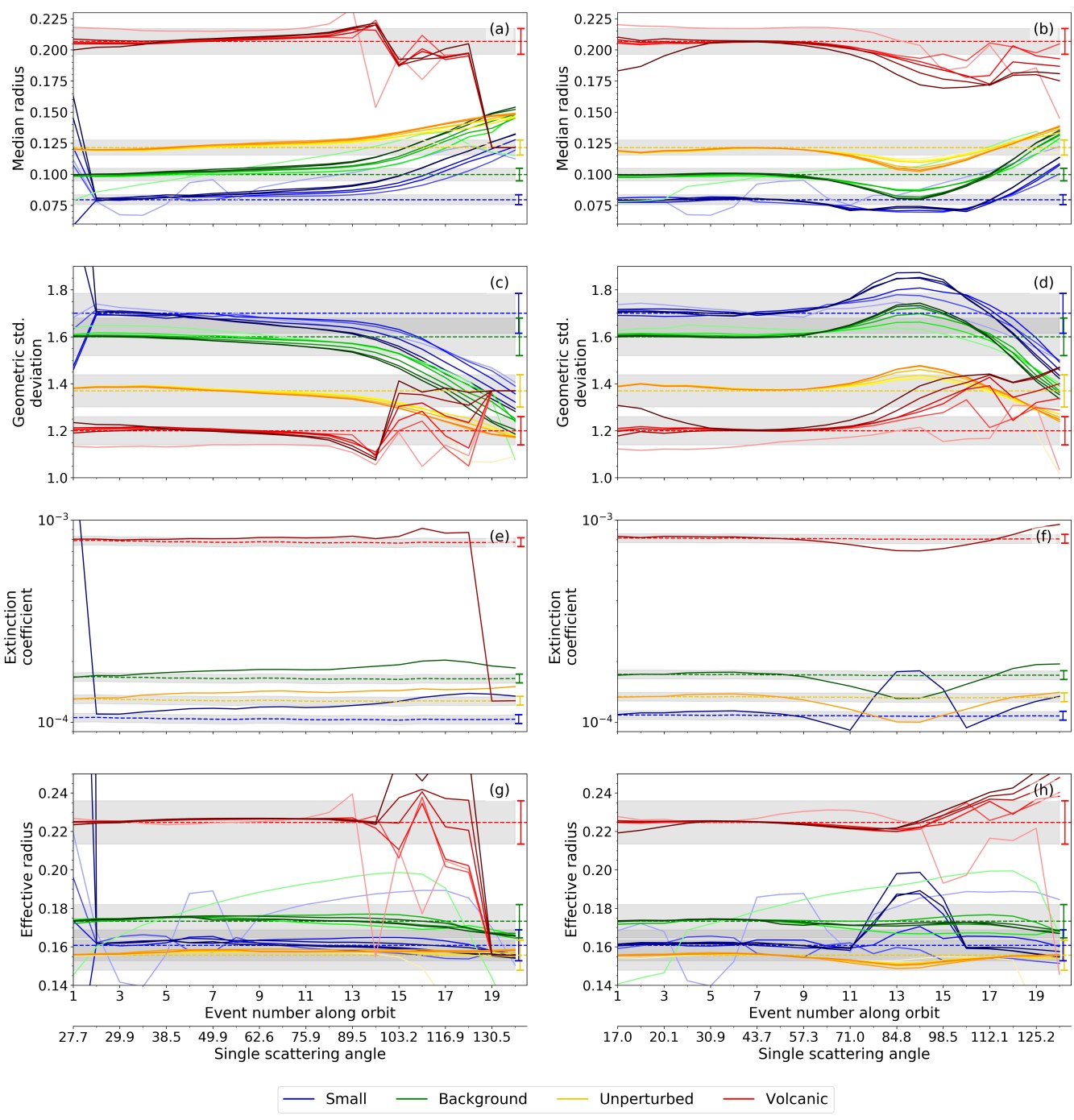

**Figure 1.** Aerosol characteristics for different artificial aerosol load scenarios given in Tab. 1. Dashed lines show the true values, solid lines the retrieval results at the western (a,c,e,g) and eastern edge (b,d,f,h) of one randomly selected SCIAMACHY orbit. Different retrieval heights are represented by colour tints, the lightness increases with the altitude. Extinction coefficients (e,f) are only shown for an altitude of 21.8 km. Shading areas framed by error bars depict the 5 % uncertainty.

characteristics of lower retrieval heights deviate more strongly from the true value. Unstable retrieval results can appear at the uppermost retrieval height ($\approx 35$ km). This is due to the low sensitivity of the limb radiances to aerosols at this altitude. Deviations of calculated and true $Ext$ exhibit a similar altitude dependence as the other three aerosol characteristics. However, they are not presented here for reasons of clarity.

It is worth emphasizing that strongly anisotropic surfaces not only cause a bias in the derived aerosol characteristics. They

also induce a slope of the bias in the across-track direction, as can be seen by comparing the left panels of Fig. 1 with right panels. We will now only focus on single-scattering angles up to 96 °. At the western edge of the swath (left panels), the corresponding nadir observations are from the near forward-scattering region of the surface, where surface reflectances are lower than the surface albedo. The retrieved effective Lambertian surface albedo is therefore underestimated. Using this underestimated value as the a priori Lambertian surface albedo in the PSD retrieval leads to a mean overestimation of $r_{\mathrm{g}}$ by

up to 11.2 % and a mean underestimation of $\sigma_{\mathrm{g}}$ by mostly up to 2.7 %. Vice versa, nadir observations at the eastern edge of the swath are from the near backscattering region of the surface, where surface reflectances are larger than the surface albedo. The retrieved surface albedo is therefore overestimated, resulting in a mean underestimation of $r_{\mathrm{g}}$ by up to 5.6 % and a mean overestimation of $\sigma_{\mathrm{g}}$ by up to 4.1 % (right panels). By calculating $Ext$ and $r_{\mathrm{eff}}$ from $r_{\mathrm{g}}$ and $\sigma_{\mathrm{g}}$, the former two also present a slight distortion in the across-track direction.

However, it is assumed that for real measurements, the upwelling radiance scattered into the instrument's field of view originates from different anisotropic surface types. The integration of these radiances likely smoothes out the anisotropic reflectance contribution of each individual surface type, resembling a Lambertian surface. This reduces the impact of the Lambertian surface assumption on the PSD parameter retrieval.

## 6.2 Sensitivity to aerosol number density

We repeat the simulation of nadir and limb radiances by assuming altitude-dependent profiles of $r_{\mathrm{g}}$, $\sigma_{\mathrm{g}}$, and $N$. These are based on balloon-borne measurements over Wyoming between 1989 and 2001 (Deshler et al., 2019) and are shown in Fig. 2 as black lines. The PSD parameters $r_{\mathrm{g}}$ and $\sigma_{\mathrm{g}}$ are retrieved using different a priori $N$ profiles. Note that the albedo is also retrieved in the second step of the retrieval algorithm as described in Sect. 4. The obtained PSD is used to calculate $Ext$ and $r_{\mathrm{eff}}$ by Eqs. (6) and (5).

Figure 2 shows the resulting profiles of $r_{\mathrm{g}}$, $\sigma_{\mathrm{g}}$, $Ext$, and $r_{\mathrm{eff}}$ using the correct $N$ profile (red), halving (yellow) or doubling (green) the correct $N$ profile, or using the $N_{\mathrm{ECSTRA}}$ profile (blue) as a priori in the retrieval procedure. The latter profile is based on the ECSTRA model climatology for aerosol background conditions (Fussen and Bingen, 1999). The results are averaged over all profiles of the randomly chosen SCIAMACHY orbit within an orbital segment of single-scattering angles between 20 and 96 °. The lower angular limit is based on instabilities that occurred during the retrieval at smaller single-scattering

angles. According to Sect. 6.1, a separate retrieval of $r_{\mathrm{g}}$ and $\sigma_{\mathrm{g}}$ at single-scattering angles larger than 96 ° is challenging due to a reduced sensitivity of limb radiances to PSD parameters. Therefore, they are not included in Fig. 2. Since the specified orbital segment comprises 12 limb observation sequences of 4 profiles each, 48 profiles are considered in the averaging. The biases in

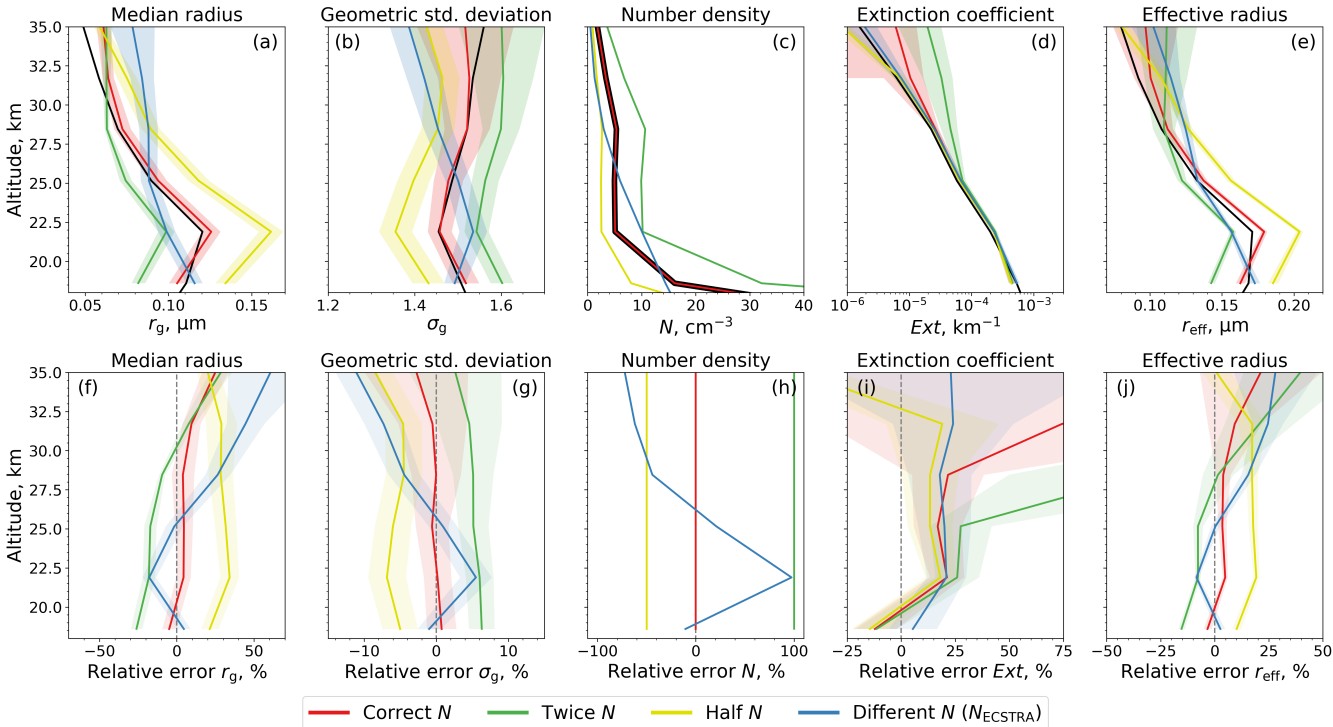

**Figure 2.** Profiles of the aerosol characteristics (a-e) and their relative errors (f-j). Black: true artificial aerosol load scenario. Colours: retrievals using a correct (red), twice as high (green), half as high (yellow), and differently shaped (blue) number density profile as a priori. Relative errors are calculated as (retrieval - true) / true $\times$ 100 % before averaging. Solid lines are averages over 48 profiles of one randomly chosen SCIAMACHY orbit, shading areas depict their standard deviations.

the retrieved ($r_\mathrm{g}$, $\sigma_\mathrm{g}$) and calculated data ($Ext$, $r_\mathrm{eff}$) are mainly due to the biases in the a priori $N$ profile and the uncertainties in the retrieval algorithm itself. Their standard deviations are mainly due to the assumption of a Lambertian surface.

If the correct $N$ profile is used as the a priori, errors in $r_\mathrm{g}$, $\sigma_\mathrm{g}$, $Ext$, and $r_\mathrm{eff}$ are mostly smaller than 9.5 %, 0.7 %, 20 % and 9.2 %, respectively. Errors in $Ext$ can exceed 20 % at altitudes above 28 km. This is explainable by aerosols smaller than $r_\mathrm{g} < 0.06\,\mu\mathrm{m}$ whose sizes are below the sensitivity limit of SCIAMACHY.

A larger assumed $N$ profile leads to wider PSDs with smaller aerosols in the logarithmic space, a smaller assumed $N$ profile leads to narrower PSDs with larger aerosols. If the a priori and the true $N$ profile differ by a factor of two, the derived aerosol

characteristics differ by $\pm 20$-30 % ($r_\mathrm{g}$), $\pm 4$-6 % ($\sigma_\mathrm{g}$), and $\pm 10$-20 % ($r_\mathrm{eff}$). Merely $Ext$ below 25 km altitude seems to be almost independent of the choice of the a priori $N$. Larger changes at higher altitudes are rather caused by too small aerosols than by errors in the assumed $N$. If the shape of the a priori $N$ profile also changes, the shapes of the aerosol characteristics profiles differ from the correct ones, especially in case of $r_\mathrm{g}$ and $\sigma_\mathrm{g}$.

To conclude, the retrieved ($r_\mathrm{g}$, $\sigma_\mathrm{g}$) and calculated ($r_\mathrm{eff}$) aerosol characteristics depend on the assumed a priori $N$ profile. The

closer this assumption is to reality, the more precisely the aerosol characteristics can be derived. However, the number density

varies in the reality to an unknown extent. Therefore, it is impossible to quantitatively estimate the retrieval uncertainty caused by the number density assumption. We can only provide uncertainty limits.

Regarding the balloon-borne OPC measurements from 2002 to 2012, a variation of $N$ by a factor between 0.5 and 2.0 encloses $\approx 80\%$ of the variation observed in the Wyoming record. This means the above mentioned uncertainties are repre-
sentative for most of the SCIAMACHY record. In the remaining 20% of cases, the balloon-borne OPC record reveals number densities in around 18 km altitude that are about four times larger than the $N_{\mathrm{ECSTRA}}$ profile assumed in the SCIAMACHY retrieval. They originate from major volcanic eruptions of Tavurvur (2006), Kasatochi (2008), and Nabro (2011).

In the aftermath of volcanic eruptions, an underestimation of the a priori $N$ profile by a factor of four does not necessarily lead to a doubling of the above mentioned uncertainties. The latter rather depend on the true PSD of the aerosol plume, i. e.,
the interplay between $r_{\mathrm{g}}$, $\sigma_{\mathrm{g}}$, and $N$. We repeated the simulations using the altitude-dependent aerosol profiles shown in Fig. 2 (black lines) but perturbed the profiles below 25 km altitude according to the OPC measurements and SAGE III/ISS retrievals (Wrana et al., 2021) in post-eruption periods. The number density at 18.4 km altitude was four times larger than the assumed a priori number density $N_{\mathrm{ECSTRA}}$. We considered cases with increasing and decreasing volcanic particle size. In the best case, the retrieval uncertainties at 18.4 km are at 30% ($r_{\mathrm{g}}$), -8% ($\sigma_{\mathrm{g}}$), and 18% ($r_{\mathrm{eff}}$). In the worst case, they are twice as large.
Uncertainties in $Ext$ are between -70 and 50% (not shown).

To summarize Sect. 6, the PSD parameters $r_{\mathrm{g}}$ and $\sigma_{\mathrm{g}}$ can be accurately retrieved up to a single-scattering angle of about 96°. This corresponds to latitudes north of 26°N in summer and 23°S in winter. Beyond this threshold, limb radiances are less sensitive to aerosols making it difficult to retrieve the PSD parameters separately. They can be subject to large uncertainties and should therefore be treated with caution. In contrast, the accuracy of $r_{\mathrm{eff}}$ and $Ext$ depends only slightly on the single-scattering
angle. The two aerosol characteristics have reasonable results for both hemispheres.

The PSD retrieval is sensitive to the assumption of a Lambertian surface and the a priori number density profile. The latter effect exceeds the former one. However, the effect of the Lambertian surface assumption can only be estimated for ideal cases, i. e., homogeneous surface types. Its quantitative estimation in real situations is not possible. Moreover, the spatio-temporal distribution of the stratospheric aerosol number density is unknown. The SCIAMACHY retrieval uses assumptions that lead to
errors in the derived aerosol characteristics. Radiative transfer simulations where the a priori and true $N$ differ by up to a factor of four imply uncertainties of $\pm 30\%$ ($r_{\mathrm{g}}$), $\pm 8\%$ ($\sigma_{\mathrm{g}}$), and $\pm 20\%$ ($r_{\mathrm{eff}}$) during volcanically quiescent and some post-eruption periods. However, uncertainties in volcanic plumes can also double depending on the PSD. $Ext$ has an uncertainty of $\pm 25\%$ during volcanically quiescent periods and of -70 - 50% during post-eruption periods.

## 7   Evaluation

Stratospheric profiles of aerosol characteristics are retrieved ($r_{\mathrm{g}}$, $\sigma_{\mathrm{g}}$) and calculated ($Ext$, $r_{\mathrm{eff}}$) for the entire SCIAMACHY observation period between 2002 and 2012. Their results are shown in Fig. 3 for an altitude of 18.4 km. The changes in the aerosol characteristics after volcanic eruptions are readily identified. The injected masses usually increase $r_{\mathrm{g}}$, $Ext$, and $r_{\mathrm{eff}}$, and

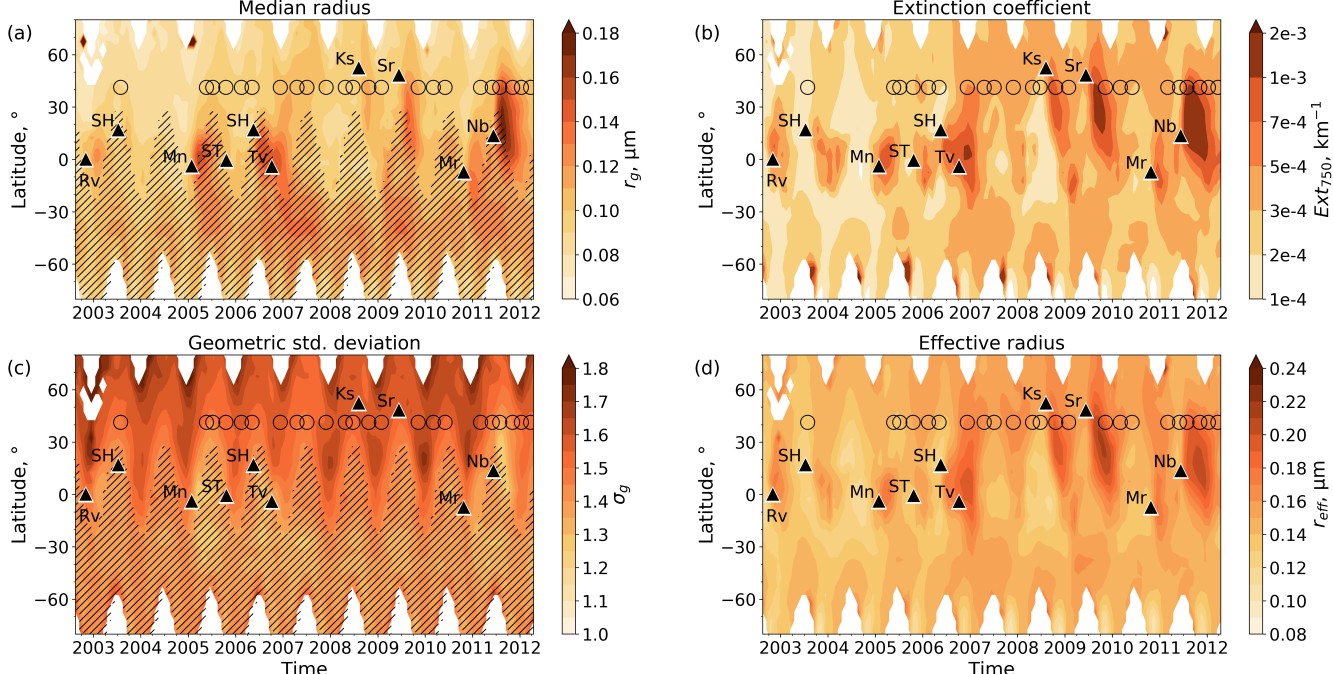

**Figure 3.** Temporal evolution of the retrieved (a,c) and calculated (b,d) v2.0 aerosol characteristics from SCIAMACHY measurements between August 2002 and April 2012 at an altitude of 18.4 km: (a) median radius, (b) extinction coefficient at 750 nm excluding those greater than 0.1 km$^{-1}$ (c) geometric standard deviation, (d) effective radius. Hatched aerosol characteristics are retrieved at single scattering angles larger than 96° and should be considered with caution. Black circles depict location and time of the balloon-borne OPC measurements. Black triangles indicate major volcanic eruptions: Rv - Reventador, SH - Soufrière Hills, Mn - Manam, ST - Santo Tomás, Tv - Tavurvur, Ks - Kasatochi, Sr - Sarychev, Mr - Merapi, Nb - Nabro.

reduce $\sigma_g$. Their temporal developments are determined spatially by advection and microphysically by nucleation, coagulation, condensation, and sedimentation.

In this section, we evaluate the SCIAMACHY v2.0 aerosol characteristics with balloon-borne measurements and satellite observations from SAGE II, SAGE III-M3M, OSIRIS, and our previous SCIAMACHY v1.4 product. The comparison of aerosol extinction coefficients is performed at 750 nm, where a direct comparison with the OSIRIS and our previous SCIA-MACHY v1.4 $Ext$ is possible. Therefore, the SAGE II extinction coefficients are converted to 750 nm via the Ångstrom exponent (Eq. (7)) calculated for the extinction ratio of 525 to 1020 nm, respectively. In case of SAGE III-M3M, the extinction

coefficients at 755 nm are used. The aerosol extinction coefficient from balloon-borne measurements is calculated by Eq. (6).

### 7.1   Comparison with balloon-borne measurements

The comparison of SCIAMACHY aerosol observations and balloon-borne measurements includes 23 collocated profiles over Wyoming between 2003 and 2012 with a maximum distance of 750 km and a maximum time mismatch of 12 hours. The

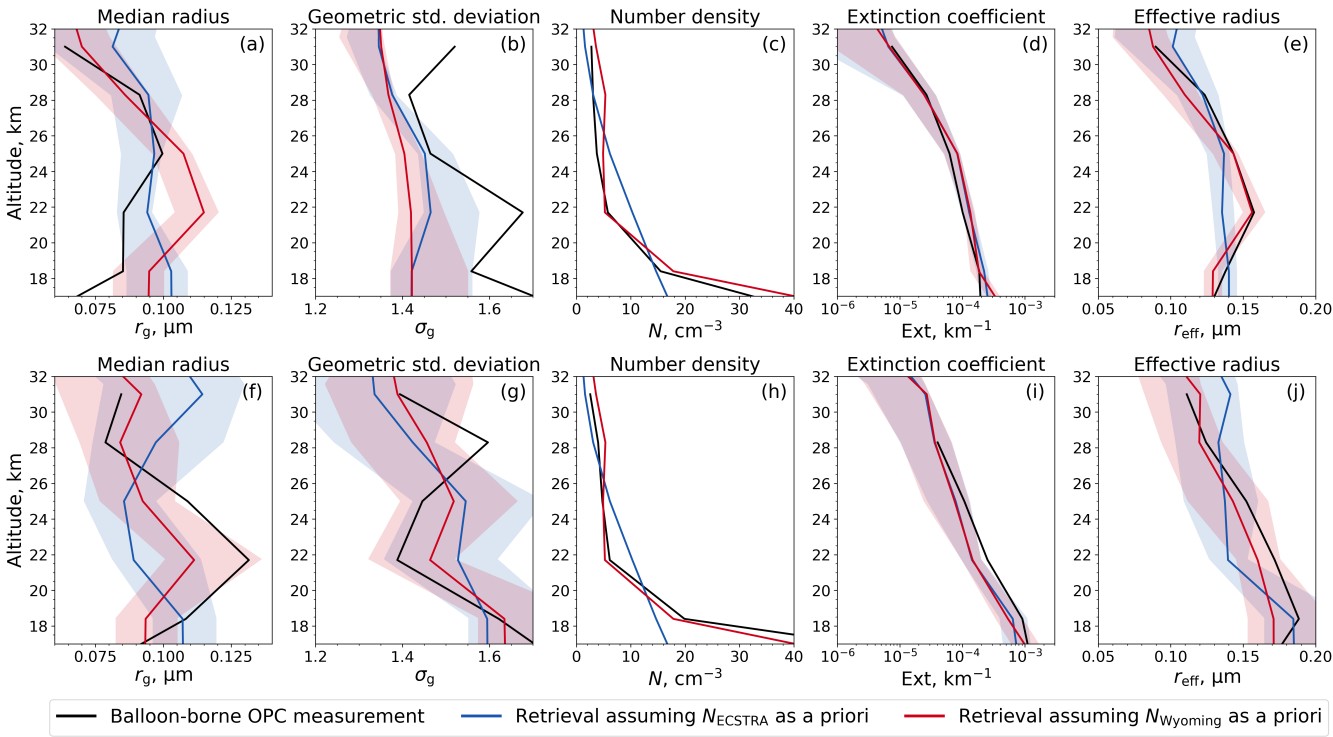

**Figure 4.** Balloon-borne-measured and SCIAMACHY-observed aerosol characteristics over Laramie, Wyoming, USA, on 7 July 2005 under aerosol background conditions (a-e) and on 7 November 2009 after the Sarychev eruption in June (f-j) (from left to right: median radius, geometric standard deviation, number density, extinction coefficient at 750 nm, effective radius). Black: Balloon-borne measurements. Blue, red: Collocated SCIAMACHY observations assuming a priori number densities shown in panel (c,h). Shading areas: Value range of the surrounding SCIAMACHY profiles with a maximum distance of 1500 km.

profiles are distributed over all seasons as shown in Fig. 3. Most of the balloon-borne profiles have been measured during
volcanically quiescent periods. Four profiles originate from volcanically perturbed situations albeit only from the peripheral area of the volcanic aerosol plume. Figure 4 shows the aerosol characteristics of two profiles, one from 7 July 2005 under aerosol background conditions (a-e) and one from 7 November 2009 after the Sarychev eruption in June (f-j). The aerosol characteristics obtained from the balloon-borne measurements are smoothed by a moving average, using a boxcar function of 3 km width, and subsequently interpolated onto the SCIAMACHY vertical grid (black lines).

Due to the Sarychev eruption, $N$ at altitudes around 18.4 km is significantly higher than that measured under background conditions. The aerosol characteristics $r_g$, $Ext$, and $r_{\mathrm{reff}}$ are also increased. At higher altitudes, there is only an indiscernible difference in $N$ between the volcanically quiescent and perturbed profile. This is not the case for $r_g$ and $\sigma_g$ which show distinct differences between the volcanically quiescent and perturbed profile. Around 21.7 km, both parameters exhibit local extrema on 7 November 2009, which can be attributed to the Sarychev eruption and the Kasatochi eruption one year earlier (Aug 2008).
These lead to a slight increase in the $Ext$ and $r_{\mathrm{reff}}$.

Figure 4 also shows the collocated SCIAMACHY aerosol profiles assuming two different a priori number density profiles, one based on the ECSTRA model climatology (Fussen and Bingen, 1999, Fig. 4(c,h) - blue) and one based on balloon-borne measurements over Wyoming between 1989 and 2001 (Fig. 4(c,h) - red). A direct comparison of the SCIAMACHY and OPC aerosol characteristics is challenging, as there is a spatio-temporal mismatch between the two data sets, albeit a small one. Within that time and space, the stratospheric aerosol condition can change slightly. It is also important to note the difference in the measurement footprint size between SCIAMACHY and OPC due to the fact that the OPC data on 7 November 2009 were only recorded at the edge of the aerosol plume. For this reason, we have displayed the value range of SCIAMACHY profiles from a larger area around the location of the OPC profiles with a maximum distance of 1500 km in Fig. 4 (shading areas). The larger number of SCIAMACHY profiles shown thus increases the probability that a SCIAMACHY profile among them has detected aerosols in a similar air mass as OPC.

On 7 July 2005, during the volcanically quiescent period, the SCIAMACHY aerosol PSD parameters are retrieved more accurately (Fig. 4 (a,b)) by assuming the a priori $N$ based on the ECSTRA model despite its overestimation at altitudes between 18.4 and 28.3 km. On 7 November 2009, during the volcanically perturbed period, the SCIAMACHY aerosol PSD parameters are retrieved more accurately (Fig. 4 (f,g)) by assuming the a priori $N$ based on balloon-borne measurements. Below 25 km, $r_{\mathrm{g}}$ is overestimated by SCIAMACHY during the volcanically quiescent period and underestimated after the Sarychev eruption. The opposite is the case for $\sigma_{\mathrm{g}}$. Remarkable are the similar profile shapes from SCIAMACHY and OPC on 7 November 2009 in case of $r_{\mathrm{g}}$ and $\sigma_{\mathrm{g}}$ by assuming the a priori $N$ based on balloon-borne measurements. This is due to the similarity of the SCIAMACHY-assumed and OPC-measured $N$ profiles. Assuming here the a priori $N$ based on the ECSTRA model provides a completely different shape of $r_{\mathrm{g}}$.

One central statement of Fig. 4 is the agreement between the extinction coefficients from OPC and SCIAMACHY, with deviations within 33 % (Fig. 4 (d)) and 43 % (Fig. 4 (i)) regardless of the assumed a priori number density profile. It can be explained by the strong correlation of the PSD parameters. For example, if $N$ is overestimated, $r_{\mathrm{g}}$ and $\sigma_{\mathrm{g}}$ change accordingly and to a certain extent. This enables the correct calculation of aerosol characteristics such as $Ext$ from the combination of all three PSD parameters. Due to the anti-correlation of the errors of $r_{\mathrm{g}}$ and $\sigma_{\mathrm{g}}$, a good agreement is also achieved for $r_{\mathrm{eff}}$ (Fig. 4 (e,j)). Only at 21.7 km, $r_{\mathrm{eff}}$ is significantly underestimated if the a priori number density based on the ECSTRA model is assumed.

Since the difference in aerosol characteristics between volcanically perturbed and quiescent profiles are not large, we consider all 23 collocated profiles in one comparison. Figure 5 shows the retrieved profiles of the median radius (a) and the geometric standard deviation (b) as well as the calculated profiles of the extinction coefficient at 750 nm (d), and the effective radius (e) from SCIAMACHY. Again, two a priori number density profiles are assumed for obtaining the aerosol characteristics, one based on the ECSTRA model climatology (Fussen and Bingen, 1999, Fig. 5(c) - blue) and one based on balloon-borne measurements over Wyoming between 1989 and 2001 (Fig. 5(c) - red). The aerosol characteristics obtained from the balloon-borne measurements are smoothed and interpolated onto the SCIAMACHY vertical grid as described above. Radiation scattered by aerosols with PSDs of $r_{\mathrm{g}} < 0.06\,\mu\mathrm{m}$ is below the sensitivity limit of SCIAMACHY (Malinina et al., 2019). Corresponding OPC measurements and collocated SCIAMACHY retrievals are therefore not in the comparison.

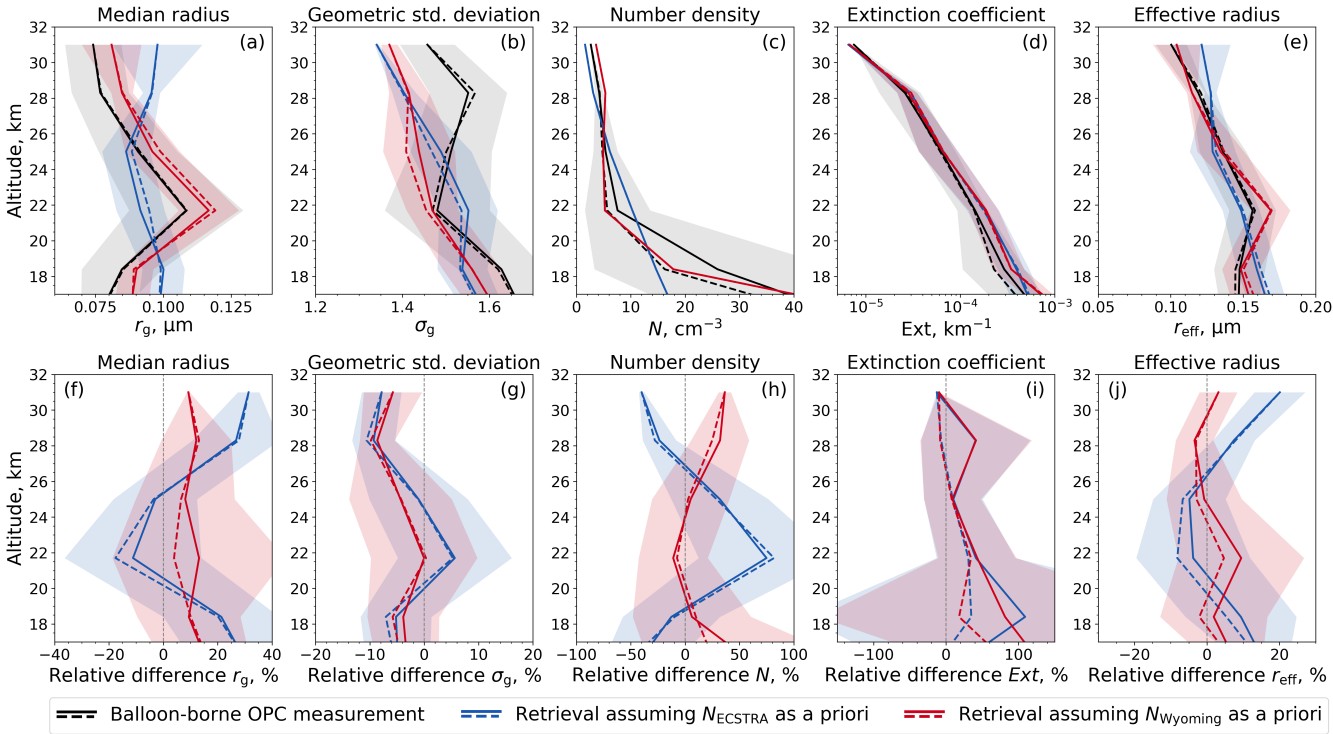

**Figure 5.** Comparison of balloon-borne-measured and SCIAMACHY-observed aerosol characteristics over Laramie, Wyoming, USA, between 2003 and 2012 (from left to right: median radius, geometric standard deviation, number density, extinction coefficient at 750 nm, effective radius). Absolute values (a-e) and relative differences (f-j) are averaged over 23 profiles with $r_g > 0.06\,\mu m$ (solid lines). The median is represented by dashed lines, the standard deviations by shading areas. Relative differences are calculated as (SCIAMACHY - balloon)/balloon $\times\,100\,\%$ before averaging. Black: Balloon-borne measurements. Blue, red: SCIAMACHY observations assuming a priori number densities shown in panel (c).

The relative differences of $r_g$ and $\sigma_g$ are mostly below 26.8 % and 9.3 %, respectively (Fig. 5 (f-g)). The smaller the error in the a priori $N$ profile, the smaller the differences. A more accurate a priori $N$ profile can usually improve the accuracy by more than a factor of 2. Furthermore, $r_g$ and $\sigma_g$ are highly anti-correlated: If $r_g$ is underestimated, $\sigma_g$ is overestimated and vice versa. This relation leads to a more accurate estimation of $r_{\text{eff}}$ (Fig. 5 (e)) and reduces their maximum relative differences mostly down to 9.4 % (Fig. 5 (j)). Relative differences in $Ext$ can exceed 100 % (Fig. 5 (i)). This large value is a result of the calculation method which is not robust against outliers. The median of the relative differences is below 20 %. Further, the relative differences in $Ext$ depend only slightly on the choice of the a priori $N$ profile.

The benefit of evaluating the aerosol characteristics obtained by SCIAMACHY with in-situ balloon-borne measurements is limited by the fact that the latter come from only one measurement site. There are also some balloon-borne measurements over Kiruna, Sweden, but at the times of OPC measurements, SCIAMACHY observations are sparse: Only 3 collocations of balloon-borne measurements with SCIAMACHY observations with distances of up to 3000 km are available. Due to the small

amount of data, a comparison of aerosol characteristics over Kiruna is not carried out. Instead, independent satellite data sets are used for a global evaluation.

## 7.2 Comparison of satellite retrieved aerosol extinction coefficients

Comparisons of satellite data products include data from a large spatial and temporal range. However, they have a decisive disadvantage compared to the comparisons with balloon-borne OPC measurements in Sect. 7.1: Similar to the SCIAMACHY v2.0 aerosol PSD retrieval product, the reference satellite data are not measured directly, but are retrieved from the satellite-measured radiances. Those retrievals are themselves subject to uncertainties, which creates an additional layer of ambiguity. A difference between two satellite retrieved aerosol products does not allow any conclusions to be drawn as to which product

is the more accurate. In order to limit ambiguity, this section is restricted to the comparison of aerosol extinction coefficients. Here, most of the reference data sets are retrieved directly (Sect. 5). Section 7.3 then deals with the comparison of aerosol sizes. In this case, the reference data sets are obtained from aerosol extinction coefficients, i. e., they are secondarily retrieved data products, which add another layer of ambiguity.

To compare the aerosol extinction coefficients from SCIAMACHY with those from SAGE II, SAGE III-M3M, and OSIRIS,

the SCIAMACHY profiles are collocated with each of the other satellite instruments. The maximum collocation time offset is 12 h. In case of SAGE III and OSIRIS, a maximum distance of 200 km is used. For SAGE II, the maximum distance from SCIAMACHY is increased to 500 km. This yields 4 255 collocations from the years 2002 to 2005. For the SAGE III - SCIAMACHY comparison, there are 5909 collocated profiles for the same time period. For the OSIRIS - SCIAMACHY comparison, almost 200,000 coincident profiles are available from 2002 to 2012.

The data with extinction coefficients greater than $0.1 \, \text{km}^{-1}$ are excluded from the comparison to reduce cloud effects. Note that this cloud filter is too simplistic to successfully eliminate all cloud contaminations. However, it prevents aerosol enriched retrievals from beeing incorrectly identified as clouds and excluded from the data set. Since the measurements of OSIRIS and both SAGE instruments provide a higher vertical resolution than SCIAMACHY, they are smoothed by a moving average, using a boxcar function of 3 km width as a weighting function, and are subsequently interpolated onto the SCIAMACHY vertical

grid.

Figures 6 and 7 show the differences between $Ext$ from SAGE II, SAGE III, and OSIRIS and those from SCIAMACHY. Note that in contrast to Figs. 2 and 5 the calculation of the difference is changed here and in the following figures. This is because we do not know which satellite data product is correct. We therefore now refer to deviations between the products, instead of calculating errors by using one satellite data product as the 'true' reference.

Figure 6 shows the differences as monthly zonal means in latitude ranges of $5°$ at different altitudes. In Fig. 7, the differences are averaged over latitudinal bins of $20°$. Due to the inclination of the Meteor-3M orbit, the SAGE III profiles are asymmetrically distributed over the northern and southern hemisphere and there are no profiles between $40°N$ and $35°S$.

The extinction coefficients from SCIAMACHY and SAGE II, SAGE III, or OSIRIS mostly agree within $30\%$, on average (Fig. 7). The differences are smaller in the middle latitudes ($\approx 30\text{-}50°N/S$) below 25 km altitude because the stratospheric

aerosols in these altitudes and regions are impacted to a lesser extent by volcanic activity than in other latitudes. Largest

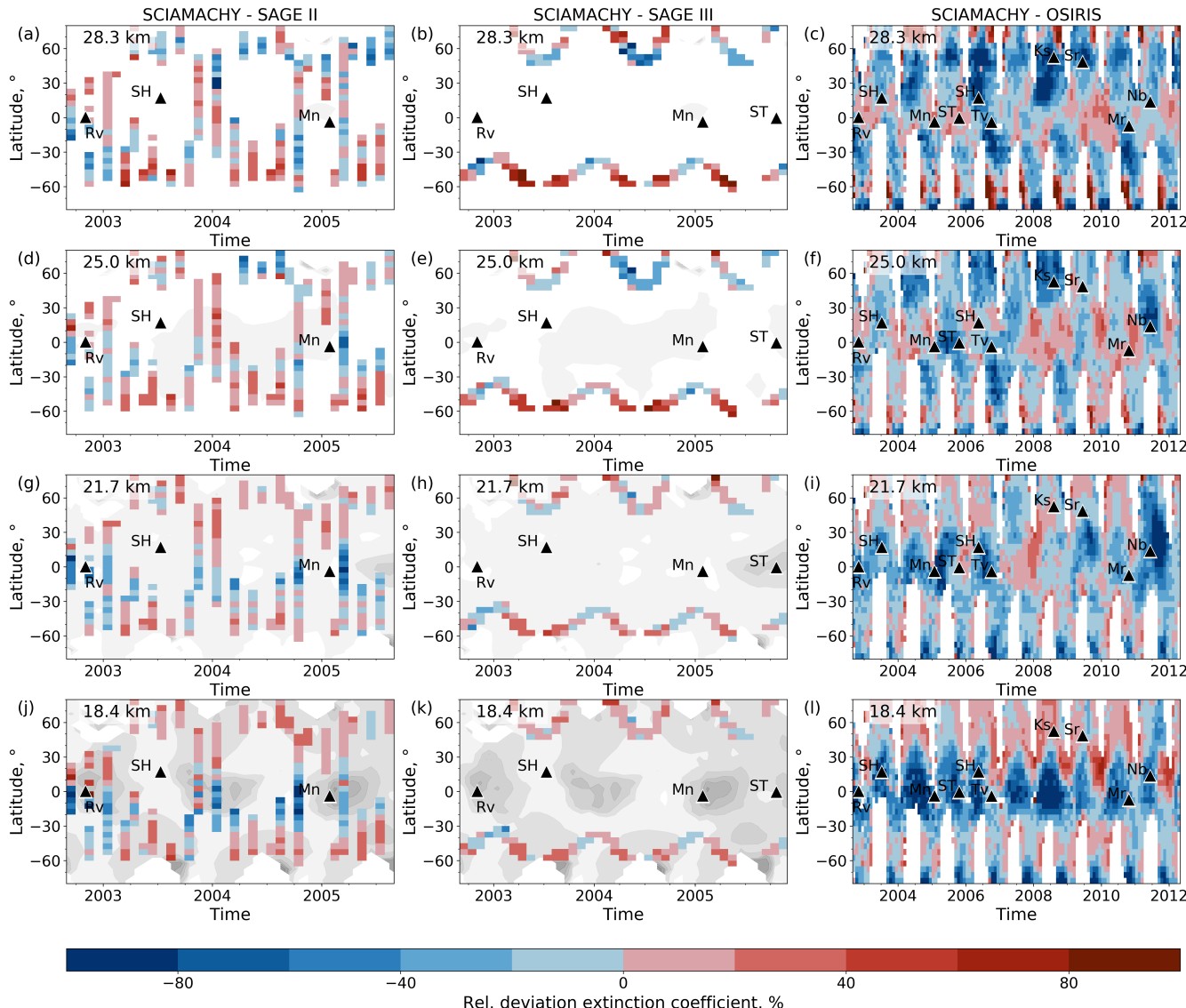

**Figure 6.** Monthly zonal means of differences in extinction coefficients from SCIAMACHY v2.0 and SAGE II (left column), SAGE III-M3M (middle column), and OSIRIS (right column) at different altitudes (rows). The latitude bin size is 5 °. Extinction coefficients are at 750 nm, in case of SAGE III-M3M at 755 nm. Relative deviations are calculated as (SCIAMACHY v2.0 - Instrument) / (SCIAMACHY v2.0 + Instrument) × 200 % before averaging. The contour plots in the background show the absolute extinction coefficient from SCIAMACHY for orientation. Black triangles indicate major volcanic eruptions. Note the different time scaling.

differences occur at altitudes above 28 km due to smaller absolute values. And they also occur in the tropics at altitudes below 22 km, although the differences are smaller in the northern than in the southern tropics. Here, $Ext$ from SCIAMACHY is smaller than SAGE II, SAGE III, and OSIRIS, especially after volcanic eruptions (Fig. 6). This is consistent with the

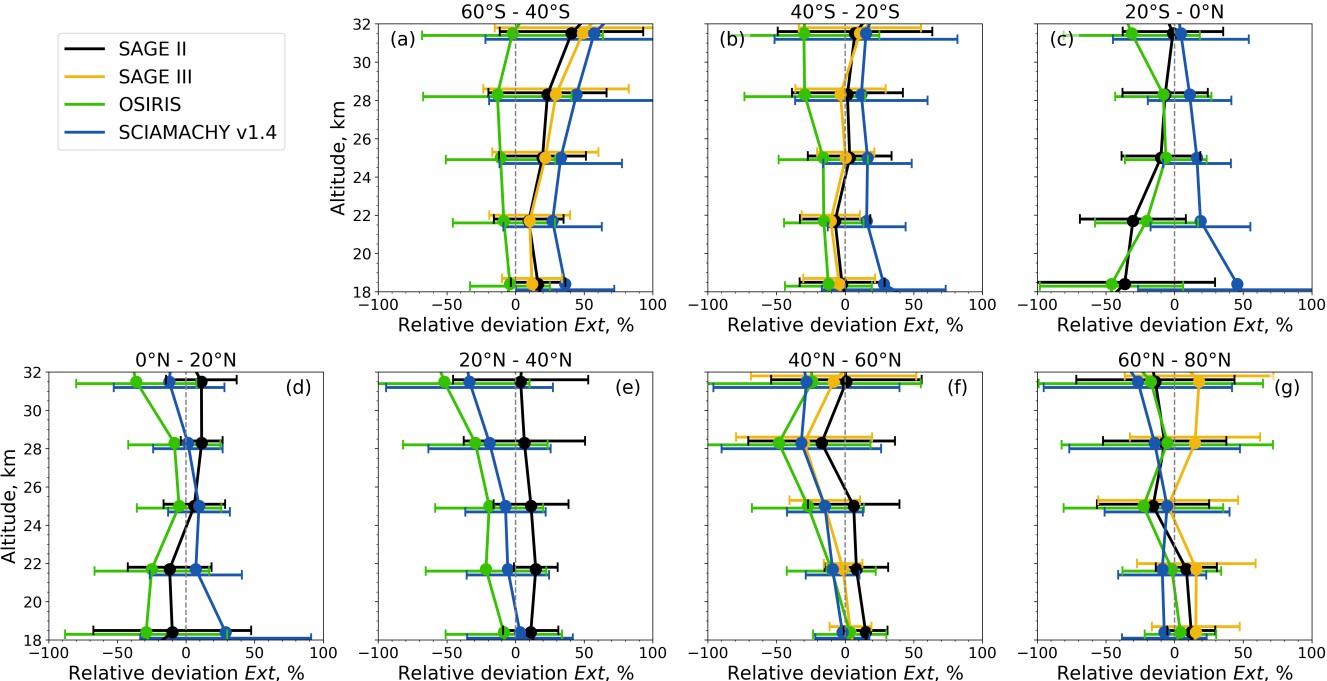

**Figure 7.** Comparison of extinction coefficients from SCIAMACHY v2.0 with those from SAGE II, SAGE III-M3M, OSIRIS, and SCIA-MACHY v1.4. The extinction coefficients are at 750 nm, in case of SAGE III-M3M at 755 nm. Relative deviations are calculated as (SCIA-MACHY v2.0 - Instrument)/(SCIAMACHY v2.0 + Instrument) × 200 % before averaging. Solid lines and filled circles show the averages within indicated latitudinal bins, error bars depict the standard deviations and are slightly shifted vertically for better readability.

comparison of SCIAMACHY and OPC data during volcanic perturbation (Fig. 4 (i)). However, there are also strong negative
differences during volcanic quiescent periods at 18.4 km altitude, e. g., before the Reventador (2002) and the Manam eruption
(2005) as well as between March 2007 and December 2008. This is most probably due to remaining cloud effects.

Because of the usually smaller $Ext$ from SCIAMACHY at 18.4 km after volcanic eruptions, the three post-volcanic eruption
periods of Kasatochi (2008), Sarychev (2009), and Nabro (2011) are outstanding due to positive differences (Fig. 6). The next
higher sampling level (21.7 km) shows negative differences in the same time periods. This might reveal a possible effect of the
wrongly assumed number density profile. However, the comparison with balloon-borne measurements in Fig. 4 (i) has shown
that SCIAMACHY slightly underestimates $Ext$ after the Sarychev eruption (2009) at all altitudes below 28.3 km. A positive
bias in Fig. 6 could therefore also be a result of uncertainties in the OSIRIS $Ext$ product.

In addition to the SCIAMACHY v2.0 aerosol PSD retrieval product, we also included the SCIAMACHY v1.4 $Ext$ retrieval
product (Rieger et al., 2018) in Fig. 7. In contrast to $Ext$ v2.0, $Ext$ v1.4 is retrieved directly from the SCIAMACHY radiances
at 750 nm that are normalized to measurements at the reference tangent altitude of 38 km. During the retrieval, a fixed unimodal
lognormal PSD with $r_\mathrm{g} = 0.08\,\mu\mathrm{m}$ and $\sigma_\mathrm{g} = 1.6$ is assumed. The retrieval algorithm is described in detail in von Savigny et al.
(2015) and Rieger et al. (2018).

It can be seen from Fig. 7 that in the northern hemisphere the difference in $Ext$ between SCIAMACHY v2.0 and v1.4 is about -10 % and is comparable to the difference in $Ext$ between SCIAMACHY v2.0 and other satellite instruments. In the

southern hemisphere, however, $Ext$ v2.0 can be on average more than 30 % larger than $Ext$ v1.4. Due to the good agreement of SCIAMACHY v2.0 $Ext$ with the SAGE II, SAGE III, and OSIRIS products, we conclude that SCIAMACHY v1.4 highly underestimates $Ext$ in southern latitudes while the new algorithm version provides more accurate $Ext$ values.

### 7.3 Comparison of satellite retrieved aerosol size parameters

We now focus on the comparison of satellite retrieved aerosol size parameters, i. e., the PSD parameters and the effective radius.

As already mentioned, this comparison uses secondarily retrieved size parameters as reference data sets. They are subject to uncertainties caused by two retrievals, firstly, that of $Ext$ and, secondly, that of the PSD parameters or $r_{\text{eff}}$ from $Ext$ (Sect. 5). Thus, differences between the aerosol size parameters from SCIAMACHY v2.0 and those from the reference data products may be larger than the differences in the aerosol extinction coefficients.

In principle, both the DWE and TWE approaches provide PSDs from SAGE observations that can be compared with those

retrieved from SCIAMACHY observations. However, SCIAMACHY v2.0 and the SAGE II DWE approach rely on different assumptions. The former utilizes a fixed number density profile based on the ECSTRA model climatology. The latter uses a fixed geometric standard deviation. These assumptions have a significant impact on the PSD shape finally retrieved. Therefore, a comparison of SCIAMACHY and SAGE II-retrieved PSDs is more an evidence of the (in)correctly assumed parameters than an evaluation of the SCIAMACHY PSD product per se. Therefore, we limit ourselves to the comparison of PSD data retrieved

from SCIAMACHY v2.0 and the SAGE III TWE approach (Fig. 8). The comparison is further limited to the latitude bins 40 - 60 °N and 60 - 80 °N due to the restricted global distribution of SAGE III observations (Sect. 5.3) and the limitation of the satisfactory separate retrieval of $r_{\text{g}}$ and $\sigma_{\text{g}}$ to the northern hemisphere (Sect. 6.2).

Figure 8 shows mean differences in $r_{\text{g}}$ and $\sigma_{\text{g}}$ between SCIAMACHY and SAGE III of less than 26.5 % and 10.5 %, respectively. The mean deviation in the a priori $N$ can be up to 51.1 %. Analogous to the sensitivity test (Fig. 2) and the

comparison with balloon data (Fig. 5), the differences in $r_{\text{g}}$ and $\sigma_{\text{g}}$ correlate with the differences between SCIAMACHY-assumed and SAGE III-retrieved number densities: If the assumed $N$ from SCIAMACHY is greater than the derived $N$ from SAGE III, $r_{\text{g}}$ from SCIAMACHY is usually smaller and $\sigma_{\text{g}}$ is usually larger than those from SAGE III. The magnitudes of differences align with the relative errors shown in Figs. 2 and 5.

The differences in $N$ show a broad distribution (Fig. 8 (e,f)). This is due to the variability of the retrieved $N$ from SAGE III

since the $N$ profile from SCIAMACHY is invariant. According to the sample distribution width, a fixed $N$ profile for the SCIAMACHY retrieval seems to be questionable, because in some cases, it can be more than twice as large or small than the retrieved $N$ profile from SAGE III. As a result, the differences of $r_{\text{g}}$ (Fig. 8 (a,b)) and $\sigma_{\text{g}}$ (Fig. 8 (c,d)) also show a significant spread, albeit less than in $N$.

The comparison of $r_{\text{eff}}$ from SCIAMACHY v2.0 with that from the SAGE series data is shown in Figs. 9 and 10. In case of

SAGE II, two different $r_{\text{eff}}$ products, one retrieved with the SAGE II v7.0 algorithm from NASA (Damadeo et al., 2013) and

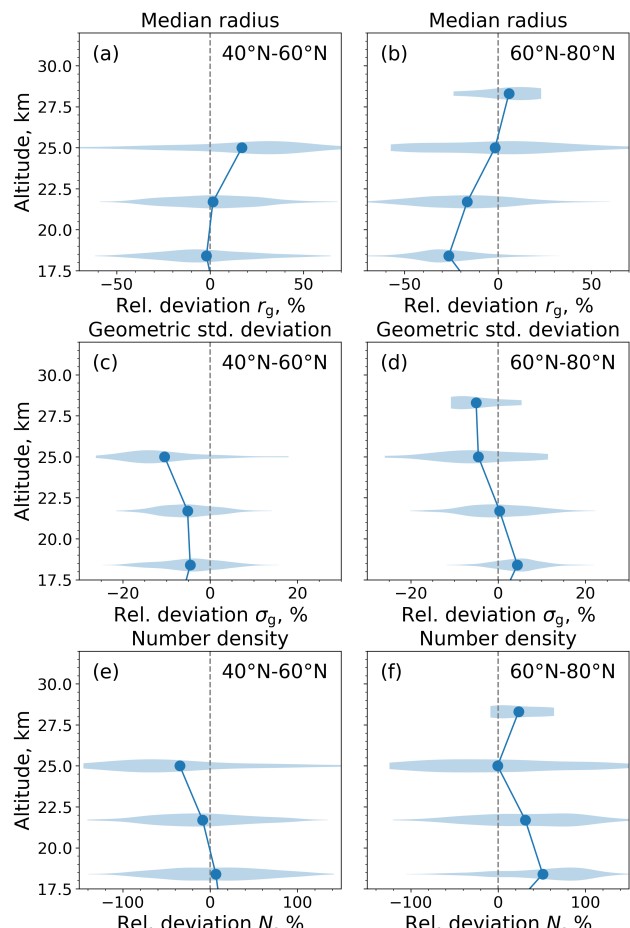

**Figure 8.** Comparison of aerosol PSD parameters from SCIAMACHY and SAGE III-M3M (from top to bottom: median radius, geometric standard deviation, number density). Relative deviations are calculated as (SCIAMACHY - SAGE)/(SCIAMACHY + SAGE) × 200 % before averaging. Solid lines and filled circles show the averages in the latitudinal range of 40 - 60 °N (a,c,e) and 60 - 80 °N (b,d,f). Shading areas depict the sample distribution.

one retrieved with the DWE approach described in Sect. 5.2, are compared to $r_{\mathrm{eff}}$ from SCIAMACHY. In case of SAGE III, $r_{\mathrm{eff}}$ is retrieved with the TWE approach (Wrana et al., 2021).

The effective radii from SCIAMACHY are systematically lower than those from SAGE II and SAGE III. At 31.5 km altitude, $r_{\mathrm{eff}}$ from SAGE II and SCIAMACHY agree well with differences below 17.7 % at latitudes from 40 °N to 40 °S and below 43 % at higher latitudes (Fig. 10). Best agreement is achieved in the tropics. The differences becoming larger with decreasing altitude south of 40 °N due to a faster increase of $r_{\mathrm{eff}}$ from SAGE II compared to SCIAMACHY. The reason is still unknown. The altitude dependency is most pronounced in the tropics. Here, the differences can increase up to 45.6 % (v7.0 NASA) and 57.0 % (DWE). The differences at 18.4 km seem to be independent of the volcanic perturbation (Fig. 9). Best agreement is


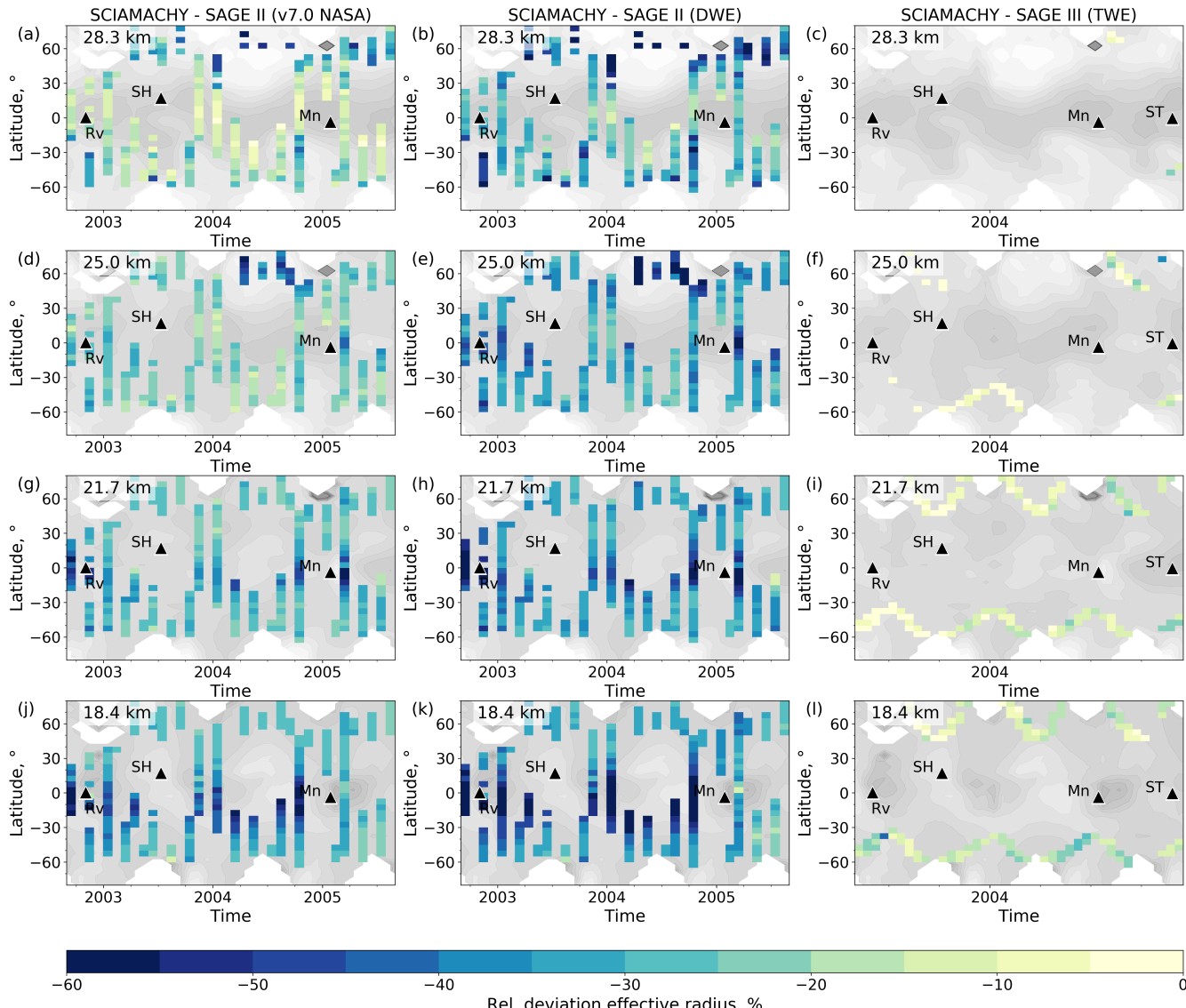

**Figure 9.** Monthly zonal means of differences in effective radii from SCIAMACHY v2.0 aerosol PSD retrieval and SAGE II v7.0 NASA (left column), SAGE II DWE (middle column), and SAGE III-M3M TWE (right column) at different altitudes (rows). The latitude bin size is 5°. Relative deviations are calculated as (SCIAMACHY v2.0 - Instrument) / (SCIAMACHY v2.0 + Instrument) × 200 % before averaging. The contour plots in the background show the absolute effective radius from SCIAMACHY for orientation. Black triangles indicate major volcanic eruptions. Note the different time scaling.

achieved between SCIAMACHY and SAGE III with deviations of 1.6 to 17.1 %. Only at latitudes from 40 °S to 60 °S, the differences are slightly larger at 18.4 and 28.3 km altitude. These small deviations are remarkable when one considers the large


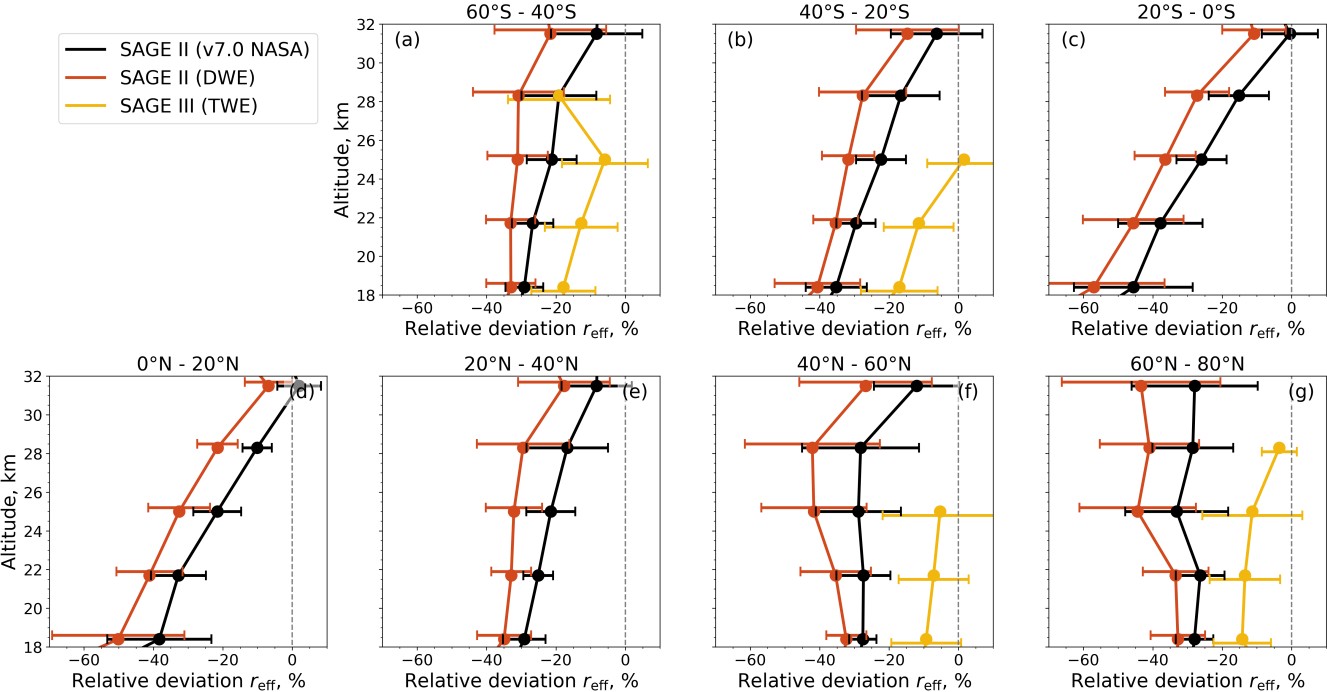

**Figure 10.** Comparison of effective radii from SCIAMACHY v2.0 with those from SAGE II and SAGE III-M3M. Relative deviations are calculated as (SCIAMACHY v2.0 - Instrument) / (SCIAMACHY v2.0 + Instrument) × 200 % before averaging. Solid lines and filled circles show the averages within indicated latitudinal bins, error bars depict the standard deviations and are slightly shifted vertically for better readability.

differences in the PSD parameters (Fig. 8). This is the advantage of comparing $r_{\text{eff}}$. Firstly, its calculation is independent of $N$ according to Eq. (4). Secondly, the anti-correlation of $r_{\text{g}}$ and $\sigma_{\text{g}}$ compensates the uncertainties in $r_{\text{eff}}$.

Missing (in case of SAGE III-M3M) or highly variable (in case of SAGE II) effective radii at higher altitudes are artefacts of the DWE and TWE approaches. These altitudes are characterized by a typically low aerosol content that leads to low signal-to-noise ratios in the satellite measurements and thus to noisy extinction ratios. Using two of them in the SAGE III TWE approach reduces the likelihood of a successful retrieval of the effective radius. In contrast, the SAGE II DWE approach requires only one extinction ratio. Though this fact increases the probability of a successful retrieval of the effective radius it is associated with large uncertainties. Remaining cloud effects in the tropics also provide large variability in the effective radius at the altitudes below 22 km albeit to a much smaller extent than for the extinction coefficient (Fig. 7).

## 7.4 Temporal comparison

We now focus on the temporal evolution of the aerosol extinction coefficient and effective radius. This is shown in Fig. 11 between 2002 and 2012 using the collocations of SCIAMACHY and OSIRIS (lines), SCIAMACHY and SAGE II (filled

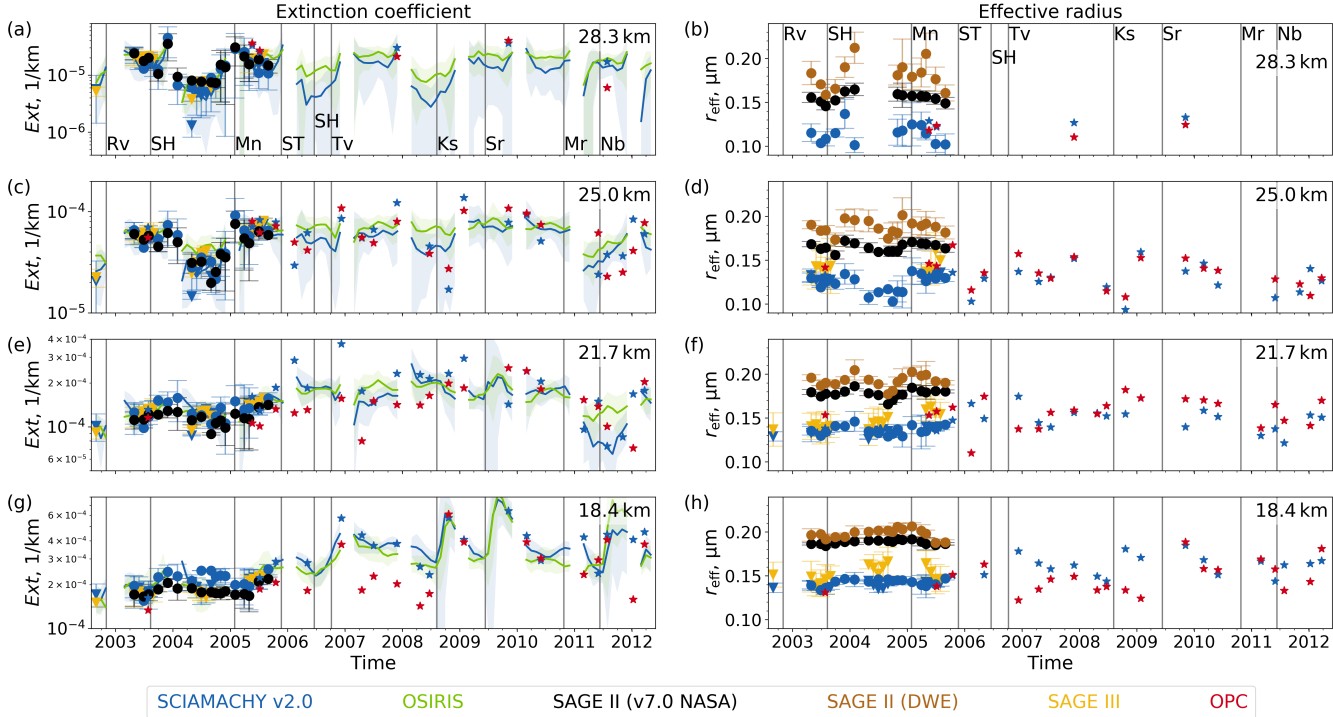

**Figure 11.** Time series of extinction coefficients (a,c,e,g) and effective radii (b,d,f,h) at different altitudes between $40°$-$60°$N. Colours indicate the instrumental source and retrieval version if necessary. Symbols indicate the data sets to be compared: SCIAMACHY is compared to OSIRIS (lines), SAGE II (filled circles), SAGE III-M3M (triangles), and OPC (stars). Intersatellite aerosol product comparisons are monthly means (symbols) with standard deviations (bars, shading areas). Comparisons with OPC data (stars) are instantaneous observations. Grey vertical lines indicate major volcanic eruptions. Note the changes in the scales for the aerosol extinction coefficient.

circles), as well as SCIAMACHY and SAGE III-M3M (triangles). The results are shown at four different altitudes. The data are presented as monthly averages within the latitude range of $40°$-$60°$N. We have chosen this latitude range because it allows the additional comparison of individually collocated SCIAMACHY and balloon-borne OPC profiles (stars). The high resolution profiles of OPC, SAGE II, SAGE III, and OSIRIS are smoothed by a moving average, using a boxcar function of 3 km width, and are subsequently interpolated onto the SCIAMACHY vertical grid. Note that for illustration purposes, the scale of the aerosol extinction coefficient is adjusted in the individual figure panels.

All data products show a temporally synchronous development of $Ext$ (Fig. 11, left panel), which increases after volcanic eruptions such as Reventador (Nov 2002), Manam (Jan 2005), Kasatochi (Aug 2008), Sarychev (Jun 2009), and Nabro (Jun 2011). As already shown in Figs. 5-7, SCIAMACHY has on average slightly larger extinction coefficients at lower altitudes ($\leq 21.7$ km) than SAGE II, SAGE III, and OPC while SCIAMACHY and OSIRIS usually agree on average. At higher altitudes ($\geq 25.0$ km), SCIAMACHY has on average slightly lower extinction coefficients. Although extinction coefficients from

SAGE II and III are not directly comparable, as their mapped profiles are from different locations and times, their temporal
trajectories indicate an agreement of both products.

In contrast to the matching extinction coefficients, the $r_{\text{eff}}$ products exhibit a multi-track behaviour (Fig. 11, right panel). It
is most probably caused by the retrieval assumptions as discussed in Sect. 8.2. At 18.4 km altitude, the effective radii of both
SAGE II algorithms agree well with only a minor offset of 0.01 µm. The effective radii of SCIAMACHY, SAGE III, and the
OPC measurements are also similar, although SAGE III $r_{\text{eff}}$ is occasionally slightly larger and OPC $r_{\text{eff}}$ slightly smaller than
SCIAMACHY $r_{\text{eff}}$. However, both SAGE II products are 0.04-0.05 µm larger than $r_{\text{eff}}$ of SCIAMACHY, SAGE III, and OPC.
While $r_{\text{eff}}$ follows the temporal development of $Ext$ - the effective radius increases with a larger extinction coefficient and
vice versa -, the offset between the satellite data products remains nearly constant in time.

The shift in $r_{\text{eff}}$ is also present at other altitudes. The offset between the individual products varies, so that $r_{\text{eff}}$ changes from a
two-track behaviour to a multi-track behaviour with increasing altitude. Largest $r_{\text{eff}}$ is retrieved by the SAGE II DWE approach,
the smallest $r_{\text{eff}}$ by the SCIAMACHY v2.0 aerosol PSD algorithm. The continued consistency of the SCIAMACHY-derived
and OPC-measured $r_{\text{eff}}$ is noteworthy (Fig. 11 (b,d,f)).

A slight but significant upward trend in the effective radius from SAGE III can be observed especially at the altitude of
21.7 km. This comes along with an increasing median radius and a decreasing geometric standard deviation (not shown). Such
a significant evolution of the aerosol particle size is not observed in SCIAMACHY and both SAGE II (v7.0 NASA, DWE) data
sets. A possible reason might be that in all of the latter three retrieval algorithms one of the PSD parameters is assumed to be
constant, namely $N_{\text{ECSTRA}}$ in the SCIAMACHY retrieval, the total $N$ of 20 cm$^{-3}$ in the v7.0 NASA retrieval, and $\sigma_{\text{g}} = 1.5$
in the DWE approach.

## 8  Discussion

The SCIAMACHY version 2.0 aerosol PSD algorithm successfully retrieves the median radius and geometric standard devia-
tion in the northern hemisphere and calculates globally the extinction coefficient and the effective radius of aerosols between 18
and 35 km altitude. The extinction coefficient of SCIAMACHY v2.0 agrees better with independent satellite observations than
that of the algorithm version 1.4 (Rieger et al., 2018). The temporal development of the effective radius is consistent with the
other observations. However, the effective radii from SCIAMACHY v2.0, SAGE II, and SAGE III-M3M reveal biases that are
most prominent between SCIAMACHY / SAGE III and both SAGE II products (Figs. 10, 11). The cause of the biases remains
to be clarified. The accuracy of the retrieved median radius and geometric standard deviation depends on uncertainties in the
assumed fixed number density. Uncertainties in both PSD parameters may blow up in case of a disregarded strong increase of
the number density, e. g., after volcanic eruptions or biomass burning events. All these topics are discussed in the following.

### 8.1  Aerosol extinction coefficient improvement

Extinction coefficients from the SCIAMACHY v2.0 aerosol PSD retrieval agree better with independent satellite observations
than those of the v1.4 $Ext$ retrieval (Rieger et al., 2018) for two reasons. Firstly, the v2.0 aerosol PSD algorithm normalizes

the measured radiances by the extraterrestrial solar irradiance (Eq. (8)) and not by a measurement at an upper tangent height, as in version 1.4. This fact makes the v2.0 aerosol PSD retrieval independent of a variable aerosol situation at the normalization altitude (Rieger et al., 2018). The disadvantage of this approach is a stronger influence of the surface albedo on the retrieval results. However, this effect can be mitigated by the prior determination of the effective surface albedo from nadir measurements.

Secondly, the v2.0 aerosol PSD algorithm utilizes multiple wavelengths between 748 and 1306 nm to retrieve the aerosol PSD parameters while version 1.4 utilizes a single wavelength of 750 nm. The multi-wavelength approach stabilizes the retrieval, increases the sensitivity to aerosols and decreases the sensitivity to PSD assumptions (Rieger et al., 2018).

## 8.2 Effective radius offset

The right panel of Fig. 11 presented distinct biases between the effective radii from the different satellite data products. Several
reasons may be responsible for these offsets.

### 8.2.1 A priori assumptions

We tested different a priori conditions and minor algorithm adjustments. The result is exemplified in Fig. 12 at an altitude of 21.7 km. Instead of the number density profile based on the ECSTRA model climatology (blue line in Fig. 2), we used the profile based on balloon-borne measurements over Wyoming (black line in Fig. 2) as a priori in the SCIAMACHY v2.0
aerosol PSD algorithm. The resulting effective radius at 21.7 km altitude is 0.02 µm larger, but the effect is too small to be solely responsible for the bias between the effective radii of SCIAMACHY and SAGE II.

Next, the a priori geometric standard deviation from the SAGE II DWE approach was increased from 1.5 to 1.6. This results in a 0.02 µm smaller effective radius, which is similar to that from SAGE II v7.0 NASA (Fig. 12).

In a third test, we checked the differences between the DWE approach of SAGE II and the TWE approach of SAGE III-
M3M as a possible reason for a bias. The SAGE III effective radius is based on a three-wavelength algorithm while the SAGE II product, which has fewer channels, is based on a two-wavelength algorithm. Additionally, the latter algorithm requires an a priori assumption of the geometric standard deviation. Therefore, we repeated the retrieval of the effective radius from SAGE III data using the two-wavelength retrieval algorithm (DWE) with different a priori geometric standard deviations (1.5 and 1.6) as well as two different wavelength combinations (520/1021 nm and 449/1544 nm).

The change from a three- to a two-wavelength retrieval algorithm changes the effective radii depending on the mismatch between the assumed (DWE) and the retrieved (TWE) geometric standard deviation. The latter decreases on a yearly average from 1.66 in 2003 to 1.61 in 2004 and to 1.57 in 2005 (not shown). Assuming a geometric standard deviation of 1.5 and using the wavelength pair 520/1021 nm, the DWE approach yields effective radii that are 0.01 µm (2005) to 0.03 µm (2003) larger than those from the TWE algorithm (Fig. 12). Better agreement between TWE and DWE data is achieved when a geometric
standard deviation of 1.6 is assumed, especially in year 2004. The positive trend in effective radii as seen by the SAGE III TWE approach disappears when using the DWE approach. The selection of the wavelength pair has only minor influence on the retrieved effective radius (not shown). Remarkable is the difference of about 0.02 µm in the effective radius between SAGE II and SAGE III when using the identical retrieval algorithm.

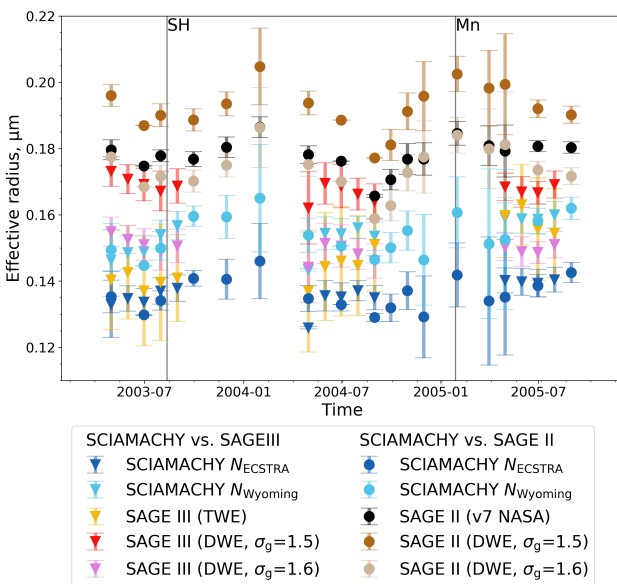

**Figure 12.** Effective radii same as in Fig. 11(f) but using different a priori conditions and minor algorithm adjustments in the retrieval of effective radii from SCIAMACHY v2.0 (bluish), SAGE II (black, ochreous), and SAGE III-M3M (yellow, reddish).

To conclude, Fig. 12 clearly demonstrates that the comparison of the effective radii is dominated by the a priori retrieval assumptions. Those may slightly distort the retrieval data. However, the comparison of SAGE II and SAGE III-M3M data from the same retrieval algorithm indicates that the individual retrieval algorithms – and the a priori assumptions – are not the only reason for the systematic biases in the effective radius.

### 8.2.2 Varying measurement sensitivities

Another reason can be found in the different sensitivities of limb scatter and occultation measurements to stratospheric aerosol particle sizes (e. g., Thomason and Poole, 1993; Rieger et al., 2014; Malinina et al., 2019). While the transmitted solar radiance measured in the occultation geometry depends only on the aerosol extinction coefficient, the scattered radiation measured in limb geometry depends at a first approximation on the product of the aerosol phase function and the aerosol scattering coefficient, both of which are functions of the aerosol PSD. Thus, limb radiances in the visible and near-infrared range are more sensitive to the aerosol size than occultation measurements. Towards smaller particles, the sensitivity decreases and falls below the detection limit faster for occultation measurements than for limb measurements (Malinina et al., 2019). As a result, small particles of a certain size can still be detected in limb geometry, but not in occultation geometry.

This fact might lead to larger particle sizes retrieved from occultation measurements than from limb scatter measurements, but only in cases where the aerosol loading is dominated by small particles and the assumed PSD shape in the retrieval algorithm differs from the true one (von Savigny and Hoffmann, 2020). This might explain at least a part of the difference in the effective radius between SAGE II and SCIAMACHY, but contradicts the similar values from SAGE III-M3M and SCIAMACHY.

Note that the comparison of retrieved extinction coefficients is not significantly influenced by the different sensitivities of limb scatter and occultation measurements. According to Eq. 6, the extinction coefficient is determined by the number density and optical cross section. Thus, larger particles contribute much stronger to the extinction coefficient than small ones. For the former, the difference in the sensitivity of limb scatter and occultation measurements to aerosols is small (Rieger et al., 2019).

### 8.2.3 Low-distorted extinction coefficients

We compared the 520 to 1020-nm extinction ratios of SAGE II with respective 520 to 1021-nm extinction ratios of SAGE III-M3M. The latter were found to be greater due to lower $Ext(1021\,\text{nm})$ values. It is not obvious, whether the SAGE II or the SAGE III extinction coefficients are closer to the truth.

A simple explanation would be a slight overestimation of the SAGE II $Ext(1020\,\text{nm})$ values which leads to uncertainties in the effective radius. On the other hand, SAGE III-measured transmissions are associated with some additional uncertainties due to an etalon effect, caused by a solar attenuator plate in the entrance optics. The solar attenuator was a neutral density filter where one side should be wedged by less than 1 arcmin. Due to the actual plane-parallel alignment of the filter sides, the attenuator acted like an etalon and caused interference patterns on the charge-coupled device (CCD) image sensor.

Thomason et al. (2010) have reported on an impact of the etalon effect on the water vapor retrieval from SAGE III-M3M. The etalon induced interference pattern was most influential when attempting to resolve fine spectral absorption features such as the water vapor or oxygen A-band retrievals, particularly because the temperature of the attenuator changed during an occultation event. Most of the designated SAGE III-M3M aerosol channels are effectively broadband or in areas where significant gaseous absorption does not exhibit fine spectral structure thus likely averaging out any interference patterns from the etalon effect. The actual impact on each aerosol product has never been fully assessed but should be negligible in most channels, with the possibility of some theoretical minor influence in the 449, 602, and 756 nm channels (Robert Damadeo, personal communication).

Considering this, we cannot provide explicit reasons for the differences in the extinction coefficients of SAGE II and SAGE III, but we can emphasize that they may contribute to the offsets between the different effective radius products.

### 8.3 Natural aerosol perturbations

Sulfur-rich volcanic eruptions and biomass burning events significantly enlarge the aerosol number density in the stratosphere. However, this increase is not taken into account in the retrieval – the number density profile for aerosol background conditions is still assumed (Sect. 4). Thus, the retrieval may return intensified deviating or more uncertain PSD parameters as investigated in Sect. 6.2. The derived aerosol characteristics should therefore be considered with caution in areas with high aerosol loading.

Obtaining more accurate aerosol characteristics in areas of strong aerosol burden is of great interest to the scientific community. It requires an optimization of the a priori information. Independently observed or simulated aerosol data sets could be used to adapt the a priori aerosol profiles in the retrieval. This supposedly simple approach is challenging for several reasons, some of which are explained below.

First, there is currently no practical approach that describes how an independent data set can be used to adapt the a priori data set. Second, in layers with strong aerosol perturbations, the required strength of regularization correlates in particular with the aerosol particle size. Some retrievals therefore require smaller covariance values to keep them stable. How to adapt the a priori covariance depending on the aerosol load is unknown. And thirdly, following from the previous point, the retrieval result may depend significantly on the a priori value if the a priori covariance is chosen too small.

## 9   Conclusions

A global data set of stratospheric aerosol characteristics has been obtained from SCIAMACHY limb observations. It contains the median radius, the geometric standard deviation, the extinction coefficient, and the effective radius between 18 and 35 km altitude. The median radius and the geometric standard deviation are directly retrieved by a multi-wavelength non-linear regularized inversion. The assumed number density profile does not change during the retrieval. The extinction coefficient at 525, 750, and 1020 nm and the effective radius are subsequently calculated from the PSD parameters. All obtained aerosol characteristics depend only marginally on the surface albedo since the PSD retrieval employs the pre-retrieved surface albedo from SCIAMACHY nadir observations.

A sensitivity study based on synthetic retrievals clearly demonstrates the operational capability of the SCIAMACHY retrieval algorithm. The median radius and the geometric standard deviation are accurately retrieved for single-scattering angles smaller than 96°, i. e., at latitudes north of 26 °N in summer and 23 °S in winter (Fig. 1). At larger single-scattering angles, limb radiances are less sensitive to aerosols. That leads to increasing uncertainties in the retrieved PSD parameters which should be treated with caution. The extinction coefficient and the effective radius benefit from the anti-correlation of the uncertainties – while the median radius is underestimated, the geometric standard deviation is overestimated and vice versa. They can therefore be retrieved satisfactorily in both, the northern and southern hemispheres. The assumption of a Lambertian surface and a number density profile in the algorithm compromises an accurate retrieval. Regarding single profile retrievals, the Lambertian surface assumption introduces an uncertainty of about 11 % (4 %) in the retrieved median radius (geometric standard deviation), depending on the anisotropy of the surface. Errors in the pre-assumed number density profile control the quality of the retrieved PSD parameters (Fig. 2).

The SCIAMACHY-derived aerosol characteristics have been evaluated with respective balloon-borne measurements over Wyoming as well as global satellite data products from SAGE II, SAGE III-M3M, and OSIRIS. The median radius differs by less than 27 % and the geometric standard deviation by less than 11 % from balloon-borne measurements and SAGE III retrievals. In case of the balloon data comparison, a more accurate a priori number density profile can typically reduce the differences by more than a factor of 2. SCIAMACHY extinction coefficients at 750 nm deviate by less than 35 % from the other available satellite data products. This deviation is smaller than when comparing SAGE II, SAGE III-M3M, and OSIRIS data with the extinction coefficient from the SCIAMACHY retrieval version 1.4 (Rieger et al., 2018). The effective radii from SCIAMACHY, balloon-borne measurements, and SAGE III agree within about 18 %.

Particularly worth mentioning is the distinct bias between the effective radius from SAGE II and that from SCIAMACHY / SAGE III-M3M. It is about 0.05 μm and can not be fully explained by the different types of the retrieval algorithms applied. An incorrect choice of a priori assumptions, different sensitivity of limb and occultation observations for aerosols, and a potential distortion of SAGE II or SAGE III-M3M extinction coefficients are discussed as potential reasons. However, a clear identification of the cause is hindered by the fact that the true effective radii are not known.

The PSD parameters, the aerosol effective radius, and the extinction coefficients at 525, 750, and 1020 nm wavelengths are publicly available at https://www.iup.uni-bremen.de/DataRequest/. This data set significantly expands the limited knowledge of stratospheric aerosol properties and enables a better understanding of aerosol microphysical processes. Currently, the retrieval algorithm is being adapted with the goal to derive all three parameters of the lognormal PSD simultaneously, namely the median radius, the geometric standard deviation, and the number density, from SCIAMACHY observations. Without any pre-assumption on the PSD parameters, the retrieval can provide lower uncertainties. It will result in more reliable PSD parameters that can be used independently of each other for interpretation purposes, at least in the northern hemisphere.

*Data availability.*  Aerosol characteristics from the SCIAMACHY v2.0 aerosol PSD retrieval are available at https://www.iup.uni-bremen.de/DataRequest/. SCIAMACHY v1.4 *Ext* is outdated and will be made available on request. Balloon-borne measurements are published in Deshler (2023), SAGE II v7.0 and SAGE III-M3M v4.0 at https://asdc.larc.nasa.gov/, and OSIRIS v7 at https://arg.usask.ca/docs/osiris_v7/. SAGE II and SAGE III-M3M aerosol characteristics based on the DWE and TWE approach will be made available on request.

*Author contributions.*  CP optimized the SCIAMACHY retrieval algorithm, performed test calculations, retrieved SCIAMACHY v2.0 aerosol PSD data products, performed the data analysis and comparison, and wrote the manuscript. FW retrieved and provided aerosol data products from SAGE II DWE and SAGE III TWE/DWE, provided valuable contribution to the SCIAMACHY - SAGE II/III comparison, and wrote Sect. 5.2 and 5.3 of the manuscript. AR developed the retrieval software and retrieved and provided the effective Lambertian surface albedo from SCIAMACHY nadir radiances. TD helped with access to the publicly available OPC measurements and calculated OPC-measured unimodal lognormal PSDs. EM, LR, and AEB provided extinction coefficients from SCIAMACHY v1.4 and OSIRIS and gave scientific advice in limb scattering retrievals. AR, CvS, and JPB initiated and supervised the project. All authors contributed to the discussions, provided critical feedback on the manuscript, and added valuable suggestions to the final manuscript.

*Competing interests.*  CvS is a member of the editorial board of AMT. The peer-review process was guided by an independent editor, and the authors have also no other competing interests to declare.

*Acknowledgements.*  This research has been funded in parts by the Deutsche Forschungsgemeinschaft (DFG) via the Research Unit VolImpact (grant no. 398006378), by the ESA via CREST project, and by the University and State of Bremen. The balloon-borne measurements were

completed under funding the US National Science Foundation and National Aeronautics and Space Administration. All SCIAMACHY v2.0 aerosol PSD retrievals reported here were performed on HPC facilities of the IUP, University of Bremen, funded under DFG/FUGG grant

INST 144/379-1 and INST 144/493-1. We thank Larry W. Thomason for his SAGE II and SAGE III expertise, and Robert Damadeo and two anonymous reviewers for their valuable comments.

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
