# Peer review of "Stratospheric aerosol characteristics from SCIAMACHY limb observations: 2-parameter retrieval"

_Atmospheric Measurement Techniques, 2023_

## Referee Comment (RC3)

**Overview**

The authors present an improvement on a previous technique for inferring particle size distribution (PSD) parameters from the SCIAMACHY data record. The proposed technique performs a 2-parameter ($r_g$ and $\sigma_g$) retrieval that is *not* limited to the tropics (an improvement over Malilina et al. 2018). The authors evaluated the impact their assumptions have on the end products and compare their PSD values (including the calculated effective radius ($r_e$)) to those measured directly by the University of Wyoming optical particle counter (UWY OPC) as well as those inferred from the SAGE II and SAGE III/M3M instruments. Further, they used their derived PSD parameters to calculate extinction coefficients (they labeled this *Ext*, I refer to this a k herein) and compared these to extinction coefficients as measured by the SAGE instruments and OSIRIS.

Overall this is an interesting paper that the community will benefit from. Overall, it is well written and the major claims are more-or-less substantiated (details below) and I believe this should be published if the authors can satisfactorily address the more salient points below.

Finally, I want to congratulate Dr. Pohl and the coauthors for the high-quality work that went into preparing this manuscript.

**Major Remarks**

The stated goal of this work is to extract PSD parameters (for the sake of brevity I include $r_e$ under the PSD label as appropriate) from SCIAMACHY data. This can be best achieved through comparison with the UWY OPC record, but comparison with OPC data is very limited. Instead, the authors devote a substantial portion of the paper to comparing their PSD parameters to those derived from the SAGE missions. Ultimately, this results in an evaluation of the assumptions in each model, which is a distraction from the intended purpose of this work. Direct comparison with the OPC record removes at least half of these assumptions and gets to the heart of the matter. This comparison should be expanded.

An alternative to direct comparison with OPC data is using the SCIAMACHY-derived PSD parameters to calculate extinction coefficient. While the authors did this the evaluation was presented in a bulk-statistics manner (Figures 5 & 8) and it would have been much more informative to expand these figures to include more meaningful information (please see specific comments below). Further, this type of comparison removes the assumptions that went into the SAGE estimates and thereby provides a more robust and meaningful comparison.

Another alternative is to use the SCIAMACHY-derived PSD parameters to calculate backscatter coefficients and compare those directly to the numerous ground-based lidars as well as CALIOP. This seems like a grossly overlooked opportunity (a potential gold mine of data) that would have significantly increased the number of intercomparison opportunities as well as the geographic coverage. If this should not be don then can the authors at least address, in the paper, why this should not be done?

There are numerous ambiguities throughout the paper that must be addressed before publication. Without correction the reader cannot understand the presented work and cannot reproduce it.

Finally, the current version of this manuscript suffers from a substantial flaw that prevents the reader from understanding and appreciating the impact and applicability of this work. The authors state that the intent of this method is to infer $r_g$ and $\sigma_g$ from SCIAMACHY data, but limit their discussion of these parameters. Instead, the authors spend more time discussing the

comparison to SAGE-derived PSD parameters that are strongly dependent on the assumptions that go into the SAGE algorithm. Further, the reader is not afforded the opportunity to see the overall performance of the SCIAMACHY-derived PSD estimates. Indeed, inclusion of PSD time-series plots (whether line plots with error bars or contour/mesh plots) would communicate a wealth of information to the reader not only on how the PSD parameters changed over the lifetime of the instrument (including volcanic impacts), but would also inform the reader of the stability of this retrieval algorithm. Such a figure would no longer limit the authors to coinciding with other instruments and would allow the authors to display the entire SCIAMACHY PSD record (all within a single figure!). In my view this paper cannot be published without this type of information content. To publish without this information leaves the reader with a knowledge gap that should not be there. Why is this so important? Because this informs the reader and end-user of how stable and reliable these estimates are. I agree that, in aggregate, comparison with SAGE/OSIRIS is generally good, but there are many unanswered questions. Does the PSD algorithm become unstable at certain altitudes? Does it perform better seasonally? How much do volcanic perturbations influence the PSD estimates? What about wildfires events? Without answering these questions I cannot ascertain the utility of these estimates.

This is a potentially fantastic paper, but in its current state it is incomplete. For these reasons I recommend that the paper undergo major revisions and be resubmitted for review.

**Specific Remarks**

- **page 2, line 44:** "catalysers" should be "catalysts"?

- **Section 4:** There is a lot of repetition within this section. It reads as if it was written twice and never cleaned up. Please consolidate the information and rewrite to be concise and precise.

- **page 7, line 209:** Why a relative humidity of 0%? Does your algorithm have some dependence in RH? This seems important since an RH of 0% is never true.

- **page 7, line 212:** Please define $R_{mod}$.

- **page 7, line 212:** You stated that $R_{mod}$ and $\sigma_g$ are arbitrarily chosen, but why these specific numbers ($\sigma_g$ of 1.37 is a precise number, why not 1.4)?

- **page 8, line 220:** What is meant by "Above 35 km, the PSD profile remains unchanged."? Please provide a reference to support this claim.

- **page 8, lines 220–222:** You use both $r_g$ and $R_{mod}$ in this sentence. I understand the difference between the 2, but it seems you use them interchangeably here. Please clarify.

- **page 8, lines 220–224:** Herein you stated that $r_g$ has a lower-limit of 0.05 $\mu$m and $\sigma_g$ is not limited. Surely $r_g$ has an upper limit as well as a lower limit. Surely $\sigma_g$ was also limited. Not to be pedantic, but $\sigma_g$ cannot be less than 1...could it, in theory, be >5? Could $r_g$ be 10$\mu$m? These values must be fundamentally limited by your model, please provide those limits here.

- **page 8, line 225:** Here you state that "step 2" solves for 3 parameters ($r_g$, $\sigma_g$, and albedo). Section 2 stated that "step 2" *only* solves for $r_g$ and $\sigma_g$. Please clarify.

– **page 9, line 255:** Please confirm that the 0.15 and 10.0 $\mu$m values are radii and not diameters (newer OPC instruments have a lower limit diameter of $\approx$150 nm).

– **page 10, lines 301–302:** This is not correct. The grating spectrometer did not include the 1550 nm channel. The 1550 nm channel was an InGaAs photodiode. Please clarify for the reader.

– **page 12, line 356:** "Their PSD profiles are specified below." I assume this refers to Table 1. Please include reference to appropriate table or figure.

– **Figure 1:** I am not colorblind, but I still have a very difficult time reading this figure. This is one of the key figures of your paper. Please update to make it more readable.

– **page 13, line 375:** "...with latitudes north of 26°N in summer and 23°S in winter." It is unclear what this means. Do you never go farther south than 23°S? Please clarify.

– **Sections 6.1 and 6.2:** This is a critical aspect of this paper. Ultimately you do not know the PSD parameters so you cannot definitively calculate the error in your derived PSD parameters that is caused by the Lambertian assumption and the assumed N profile. Therefore, how do you propagate the error from these 2 assumptions into your estimates for $r_g$, $\sigma_g$, and extinction coefficient (e.g., in Fig. 4)? Is this uncertainty ignored for the rest of the paper, or is it accounted for?

– **page 14, lines 403–404:** Here you state that you averaged over all angles from 20°– 96°. If I understood Section 4 correctly the variation in scattering angle has no impact on this process. If that is correct then please ignore this comment. If there is a dependence then would you please clarify here.

– **Figures 2 & 4:** Do I understand this correctly that there is no variability in the inferred number density? You should be able to infer N as well (I thought that's what you did in Figure 7), so I expect some spread in the profiles. Please clarify.

– **page 15, line 413:** "...correct selection of the a priori N is crucial..." That's not how I interpret Figure 2. It looks like the algorithm is sensitive to the N profile and how close it is to reality. However, "crucial" may be an overstatement. The $r_g$ is mostly within $\pm$25% up to $\approx$30 km. The $r_e$ is even better. I think the text, as written, misleads the reader. Please revise.

– **page 15, line 418:** "...latitudes south of 26°N in summer and 23°S in winter." Please see similar comment above.

– **Figure 3:** Please consider plotting panel (a) on a log scale. Maybe not necessary, but it helps readability. If the authors disagree then please disregard this comment.

– **page 16, line 424:** "2002 and 2011" Why not "2002 and 2012"?

– **page 16, line 426:** "...these processes are evidenced in Fig. 3." This level of information cannot be ascertained from a simple extinction coefficient plot. Please revise.

– **Figure 4:** How were the relative error statistics calculated? Did you first calculate the relative errors then average, or did you calculate the average $r_g$ then the relative error?

– **Figure 4:** This is possibly the most important figure in this paper (comparing to OPC) and should receive more attention. You put all of the data into one plot (I realize the OPC data are sparse), but this leaves me wondering if information is lost in the bulk statistics. Did volcanic activity significantly change the performance of your method? Breaking this analysis into 2 paradigms (volcanically perturbed/not-perturbed) would be illuminating. Without this information it is difficult to realize the value of this method.

– **page 17, lines 441–442:** "...and one based on balloon-borne measurements over Wyoming before 2002..." This is confusing. This sounds like you use an OPC-based climatology as your N profile, which has the potential to significantly bias your evaluation. This raises several critical questions regarding the methodology.

1. Are you now using an OPC-based climatology for your N profile? If so, did you also use this climatological profile in creating Figure 2?

2. You stated this N profile is based on WY OPC data collected before 2002. Does this include data collected throughout the entire record (i.e., back to 1971)? If so, how do you handle the differing OPC instruments? How do you handle extreme outlier events like El Chichon and Pinatubo? How did these events influence the overall performance of your algorithm?

3. From Figure 4, it looks like the OPC-based N profile yields better performance than the ECSTRA profile (better errors, the profile shape is in better agreement with reality, etc.), yet you never state which N profile you use in the operational algorithm. I assumed you use the ECSTRA profile, but Figure 4 indicates the OPC profile is better. Please clarify.

– **page 18, lines 448–449:** "...which can be explained by a small reference value (Fig. 4 (i))." Did you mean Fig. 4 (d)? Also, I'm not sure Figure 4 supports this claim. Extinction coefficients on the order of 1E-4 (i.e., $z \leq 24$ km) are not small. What you are dealing with here is, in fact, relatively large differences in the derived extinction coefficients.

– **Figure 5:** You allude to this figure throughout the text and reference statistics from this figure. However, I cannot read this figure. There are so many colors on top of each other I cannot tell where they all begin/end...and I cannot tell if the quoted numbers quote are correct (I assume they are, I just cannot verify). Please improve the readability of this figure.

– **Figure 5:** Why the stark difference between this figure and Figure 4 (i)?

– **Figure 5:** Why not use the same method to calculate error as used in Figure 4?

– **page 19, lines 489–490** "...with the differences decreasing with altitude." This is not what Figure 6 shows. All differences increased (except panels f and g). Please clarify.

– **Figure 6:** This figure exemplifies why the comparison with OPC (or even a comparison of k) is of more value than comparing with the SAGE-derived PSD parameters. The extreme slope in panels b–e indicates a systematic error in the SAGE PSD values. How is the reader to draw meaning from this analysis? Can the authors account for this extreme slope? To be honest, the extinction comparison looked promising, so I was somewhat shocked to see this odd behavior in these profiles (I do note that the authors plotted here data from both SAGE missions and that this is a figure for $r_e$ (a derived product that is based on derived

products, which adds another layer of obfuscation)). Given the ambiguities, it is unclear how this figure is helpful, especially since the intended purpose of this work is to extract $r_g$ and $\sigma_g$ from SCIAMACHY...not $r_e$.

– **page 20, line 501:** "...utilizes a fixed number density profile..." What is this profile? Is it the OPC-based climatological profile, or the ECSTRA profile?

– **page 20, line 509:** "...can be by up to 51.1%." I have 2 points. First, should this be "...can be up to 51.1%"? Second, the error in N (including error bars) goes well beyond 51% (maybe even over 100%). Please clarify.

– **Section 7.3 and Figure 8:** It is unclear how the 1989–2002 time period is relevant to this study. This seems like wasted space and text. Figure 8 would be much more meaningful to the reader if the authors were to remove the 1989–2002 period (this would also allow them to use consistent scales) and create a single time series (i.e., 2002–2012). The figure could be further improved by breaking it into 2 figures (1 for extinction, 1 for $r_e$), with each figure containing sub-plots for different altitudes (e.g., panel (a) could be 30 km, panel (b) is 25 km, etc.). The value of doing this is it shows the reader the relevant information and provides the reader a much better understanding of the performance of this algorithm at multiple altitudes. Finally, showing this as a time series (instead of the aggregate profile statistics) allows the reader to appreciate the influence volcanic perturbations have on this method.

– **Figure 8:** It seems the first legend (titled "Symbols: Comparison of") is unnecessary. There is no "comparison" plotted in these panels, so it is unclear what the legend title means. The 2 legends could be consolidated to make the figure more easily interpreted.

– **page 23, lines 532–533:** "...due an increasing median radius with a simultaneously decreasing geometric standard deviation." First, should this be "...due to an..."? Second, this claim is conjecture and is not supported by the analysis. Since you are dealing with inferred values at this point I suspect that everything you see here, including the changes in $r_g$ and $\sigma_g$, are symptoms as opposed to the root cause. Can the authors provide additional support for this claim or reword this sentence?

– **page 23, line 535:** What is the constant in the SCIAMACHY process? Undoubtedly it is N, but which N (UWY OPC or ECSTRA) is it?

– **Figure 9:** Again, this figure is an excellent example of why the authors should make the comparison with the UWY OPC products their first priority and give this analysis the most weight. This is also why the second priority should be comparing SCIAMACHY-derived extinction coefficients with SAGE/OSIRIS extinction coefficients should receive more weight and be second priority. This figure clearly demonstrates that the comparison is dominated by the assumptions in the SAGE estimates.

– **page 27, lines 648–652:** I have several points about this text.

1. I saw nothing in the paper to indicate that "the median radius and geometric standard deviation are fully reliable *only* in the northern hemisphere." (emphasis mine). If this was in text then I sincerely apologize. Since the authors are limited to UWY OPC data (i.e., northern hemisphere) for validation I don't know how you could determine this. Please clarify.

2. If the southern hemisphere SCIAMACHY PSD parameters are "bad" then how do you justify inferring extinction coefficient and $r_e$ from them? It would seem you are getting a more-or-less right answer for the wrong reasons. Please clarify.

3. The previous 2 points seem to be contradictory. Please clarify in the text.

4. The authors stated that the intent of this method is to derive PSD parameter ($r_g$ and $\sigma_g$) from SCIAMACHY data...and do so globally. If the southern hemisphere $r_g$ and $\sigma_g$ are, per the authors' statement, unreliable then has the intent of this work failed? If so, the abstract must be updated to reflect the limited applicability of this method.

– **page 27, line 666:** The "link" must be updated.

– **Appendix A:** While this appendix is interesting, I fail to see how it makes a substantive contribution to the paper and should be removed (unless the authors can justify its inclusion, of course).

---

## Author Comment (AC1)

Dear Reviewer,

we thank for your very valuable comments. We revised our paper in light of your comments (in black). The answers are shown below in red.

Best wishes,

Pohl et al.

The paper is dedicated to improved retrievals of aerosol characteristics from SCIAMACHY limb observations. Compared to the previous version of (Malinina et al., 2018), the algorithm has been improved, and implemented to measurements not only in the tropics, but also on the entire globe.

The measurements that allow retrieval of information about aerosol particle size distributions are limited, while this information is important both for evaluation of climate response and also for retrievals from satellite measurements. This paper provides a valuable contribution to this topic.

The paper is well-structured and well-written. I recommend it for publications. Please find my minor comments below.

COMMENTS.

About assumption of fixed number density. While further in the text it becomes clear what you mean by "fixed number density", the first mentioning of this creates many questions (for example, P.3). It is worth to add something like "details are provided below " with the first mentioning the fixed number density assumption.

We adapt respective sentences:

Abstract: This assumes a fixed number density profile → This assumes a number density profile that does not change during the retrieval.

p3: … by assuming a fixed number density. → A number density profile is assumed that does not change during the retrieval.

Conclusion: … assuming a fixed number density profile. → The assumed number density profile does not change during the retrieval.

A related question: have you tried a maximum a posteriori inversion with three parameters retrieved (Bayesian approach with a priori information)? After obtaining the estimates of the parameters with your two-parameter retrievals, this might be a working approach.

We thank you for your suggestion. Unfortunately, such an approach is not working. The results of the 2-param retrieval (first inversion) depend on the a priori number density. That means, uncertainties in the a priori number density are largely adapted by the two retrieval parameters. Using these results in the 3-param retrieval (second inversion) will not iterate properly for the following reasons:

1) The determination of a solution is complicated by the fact that neither the a priori values (due to the a priori N dependence in the 2-param retrieval) nor the a priori covariance values (due to the lack of better knowledge) have to be correct in the 3-param retrieval.

2) A 3-param retrieval is already complicated in itself. The differences between simulated and measured radiances can be largely minimized by adjusting only 2 parameters. The third PSD parameter usually provides only little additional information. Thus, we do not get sufficient sensitivity for all 3 parameters.

Concerning point 3, we have added the following sentences at an appropriate position:

„While $r\_g$ and $sigma\_g$ are derived, N remains unchanged at the initial profile for two reasons.

First, the spectral signatures of the three parameters are strongly correlated. Changes in measured limb radiances can be largely described by adjusting only two PSD parameters. The third PSD parameter usually provides only little additional information. That means, a multitude of aerosol PSD profiles result in the similar measured limb radiance. Fixing one PSD parameter restricts this unambiguity and gives more weight to the other two PSD parameters when responding to the given limb radiance. Second, ..."

Line 78. " …on the entire globe, here".  "here" is not needed

Done.

Line 190 "Either"  -> "either"

Done.

Line 245: It is better to use  the word "data" instead of "products"

We have added the word „data" to the heading → Aerosol data products.

(And have also adapted other text passages, accordingly.)

Figure 3. It would be useful to add  letters near the triangle indicating volcanic eruptions, and to provide a table listing them.

Done.

Figure 8. Please indicate dates of volcanic eruptions in the figure, for example, by adding vertical lines.

Done.

Line 693: Please provide the link to the dataset.

Done.

---

## Author Comment (AC2)

Dear Robert Damadeo,
we thank for your very valuable comments. We revised our paper in light of your comments (in black). The answers are shown below in red.

One important note:
If you agree we will include your comment on the etalon effect into the manuscript, see below.

Best wishes,
Pohl et al.

The authors describe a new aerosol retrieval from SCIAMACHY as well as new retrievals of various aerosol PSD parameters and compare them with other ground- and space-based measurements and retrieved parameters. Knowledge of both the amount and size distribution of aerosols is of key importance not only for climate modeling but also the retrievals of both aerosols and trace gases from many different instruments. PSD parameters are particularly important for many satellite retrievals as most instruments rely upon assumptions, rather than measurements, of these parameters for their retrieval algorithms. This paper is well organized and presented and I would recommend it for publication. The following comments are minor and only offer up suggestions for improvement or clarifications.

Pg 07, Ln 197: "For the public, we also calculate the aerosol extinction coefficient at 525 and 1020 nm to enable a comparison with other satellite aerosol products"

How is this done exactly? Is this done using the measurements at 750 nm and 1020 nm to compute the Angstrom exponent to then relate 525 nm to one of those channels?

It is calculated by Mie theory. We refer to the corresponding equation (6).

Pg 07, Ln 209: "0% relative humidity"

Does this assumption impact the data quality at the bottom of the profiles?

Yes, please see next comment.

Pg 07, Ln 210: "They are specified as a mixture of 75% sulphuric acid and 25% water."

How does this assumption impact the results seeing as how recent measured estimates of this parameter from ACE-FTS show variability in this concentration?

To address this comment, we have added the following text:

„Both, the aerosol composition and the relative humidity, are idealistic assumptions. The percentage of sulphuric acid can vary slightly in reality (Turco et al., 1982, Steele et al., 2003, Doeringer et al., 2012). The Atmospheric Chemistry Experiment Fourier Transform Spectrometer (ACE-FTS) even occasionally detected sulphuric acid levels of less than 50 % after the Raikoke eruption 2019 (Boone et al., 2022). The stratospheric relative humidity is usually between 0 and 10 % (Steele et al., 1981). However, we stick to these conventions because the OPAC database does not offer more realistic compositions. The resulting retrieval uncertainty was estimated by comparing retrieved PSD parameters assuming a relative humidity of 0 % and 80 %. The latter value is exceedingly high, but allows a maximum uncertainty estimate of below 15 % for the mode radius and below 10 % for the geometric standard deviation (not shown). These values can also be regarded as an uncertainty estimate due to an incorrect aerosol composition. By increasing the relative humidity, the particles absorb water vapour, which reduces the percentage of sulphuric acid. As a result, the aerosol refractive index (Palmer and Williams, 1975) changes with a similar amplitude to that of an increase in relative humidity (Hess et al., 1998)."

Pg 08, Ln 216: If N remains fixed, how is the value determined? Is a single value retrieved

somewhere else for each measurement or is a single value used for all retrievals?

The N profile is not retrieved, but is preset and never changes. To make this statement clearer, we reorderd Sect. 4. The general description of the atmosphere is at the beginning („ a number density profile based on the ECSTRA model") followed by the retrieval description. Respective sentence is now:

„While r_g and s_g are derived, N remains unchanged at the initial profile for two reasons."

Pg 08, Ln 240: "The noise covariance matrix is assumed to be diagonal, i. e., the noise is spectrally and spatially uncorrelated."

   If there is known stray light, should this be the case? Or is the stray light sufficiently small as to ignore it completely? I am guessing there is some transition region near the upper end of the retrieval range.

To address this comment, we have added the following text:

„In the absence of better knowledge, the noise covariance matrix is assumed to be diagonal, i.e., the noise is spectrally and spatially uncorrelated. Since the influence of stray light below 35 km is small, this assumption should not have a negative impact on the retrieval. The diagonal elements contain..."

Pg 16, Ln 430: "Therefore, the SAGE II and SAGE III extinction coefficients are converted to 750 nm via the Ångstrom exponent"

   If you use the ~750 nm SAGE III channel for the TWE method of computing Reff, why not also use the extinction data from that channel instead of converting from the other two?

We have switched from converted extinction coefficients at 750 nm to extinction coefficients at 755 nm.

Pg 18, Ln 471: "Discrepancies are slightly higher in the tropics at altitudes below 22 km due to cloud effects"

   Why not apply some rudimentary cloud filtering (or omit all data below just above [e.g., 1 km] the tropopause)?

Cloud identification is complicated. Cloud filters with stronger efficacy than applied in the manuscript could be used but at the cost of missing some aerosol-rich plumes. We therefore argue that sticking with the simple cloud filter is preferable but also mention the possible cloud contamination:

„The data with extinction coefficients greater than 0.1 km$^{-1}$ are excluded from the comparison to reduce cloud effects. Note that this cloud filter is too simplistic to successfully eliminate all cloud contaminations. However, it prevents aerosol enriched retrievals from beeing incorrectly identified as clouds and excluded from the data set."

Pg 20, Ln 500: "However, SCIAMACHY v2.0 and the SAGE II DWE approach rely on different assumptions …"

   What about the SAGE II NASA approach? It appeared that the SCIAMACHY v2.0 Reff matched those better than the DWE approach.

The SAGE II NASA approach used here (v7.0) does not provide any PSD parameters.

Pg 20, Ln 509: "… the differences in rg and σg correlate with the differences between SCIAMACHY-assumed and SAGE III-retrieved number densities …"

   They correlate, but do their magnitudes align with the sensitivity tests shown in Fig. 2?

Yes, we have added the following sentences:
„If the assumed N from SCIAMACHY is greater than the derived N from SAGE III, rg from SCIAMACHY is usually smaller and σg is usually larger than those from SAGE III. The magnitudes of differences align with the relative errors shown in Figs. 2 and 5.“

Pg 23, Ln 532: "Note that effective radii from SAGE III increase slightly but significantly over time. It is due an increasing median radius with a simultaneously decreasing geometric standard deviation. Such an evolution of the aerosol particle size is not observed in SCIAMACHY and both SAGE II (v7.0 NASA, DWE) data sets. This might be because in those three retrieval algorithms one of the PSD parameters is assumed to be constant."

Is there another referenceable source that definitively shows mean radius/geometric SD systematically increasing/decreasing over this time period to show that the SAGE III data is correct and the SAGE II / SCIAMACHY data is incorrect?

Unfortunately not. The key message of this plot (new Figure number 11) should rather show differences than indicate which data set might be correct. The latter is not possible due to the lack of available data sets. We rephrased this passage based on another referee comment, who suspects these „are symptoms as opposed to the root cause.":

„A slight but significant upward trend in the effective radius from SAGE III can be observed especially at the altitude of 21.7 km. This comes along with an increasing median radius and a decreasing geometric standard deviation (not shown). Such a significant evolution of the aerosol particle size is not observed in SCIAMACHY and both SAGE II (v7.0 NASA, DWE) data sets. A possible reason might be that in all of the latter three retrieval algorithms one of the PSD parameters is assumed to be constant, namely N_ECSTRA in the SCIAMACHY retrieval, the total N of 20 1/cm³ in the v7.0 NASA retrieval, and s_g=1.5 in the DWE approach.“

Pg 26, Ln 615: "Thomason et al. (2010) have reported on an impact of the etalon effect on the water vapor retrieval. An additional influence of this effect on the extinction coefficient retrieval cannot be excluded."

It is unlikely that the etalon impacts either the 520 or the 1020 nm channels in a meaningful way. An etalon is a spectral interference pattern that can change with the thickness (correlated to temperature) of the attenuator. This interference pattern will be most influential when attempting to resolve fine spectral absorption features such as with the water vapor or oxygen A-band retrievals, particularly because the temperature of the attenuator will change during an occultation. The measurement of aerosol through the 520 and 1020 nm channels does not depend on resolving any spectral features and is effectively broadband thus likely averaging out any interference patterns.

We thank you for this valuable comment. We renamed and rewrote this section and – if you agree – included your comment in the manuscript. We have additionally included the 449, 756, and 1544 nm channels.

„8.2.3 Low-distorted extinction coefficients
We compared the 520 to 1020-nm extinction ratios of SAGE II with respective 520 to 1021-nm extinction ratios of SAGE III. The latter were found to be greater due to lower Ext(1021 nm) values. It is not obvious, whether the SAGE II or the SAGE III extinction coefficients are closer to the truth.
A simple explanation would be a slight overestimation of the SAGE II Ext(1020 nm) values which leads to uncertainties in the effective radius. On the other hand, SAGE III measured transmissions are associated with small uncertainties due to an etalon effect, caused by a solar attenuator plate in the entrance optics. The solar attenuator was a neutral density filter where one side should be wedged by less than 1 arcmin. Due to the actual plane-parallel alignment of the filter sides, the

attenuator acted like an etalon and caused interference patterns on the charge-coupled device (CCD) image sensor.

Thomason et al. (2010) have reported on an impact of the etalon effect on the water vapor retrieval from SAGE III-M3M. The etalon induced interference pattern was most influential when attempting to resolve fine spectral absorption features such as the water vapor or oxygen A-band retrievals, particularly because the temperature of the attenuator changed during an occultation event. The measurement of aerosol from the 449 to the 1544 nm channels used for the effective radius retrieval does not depend on resolving any spectral features. It is effectively broadband thus likely averaging out any interference patterns (Robert Damadeo, personal communication).

Considering this, we cannot provide explicit reasons for the differences in the extinction coefficients of SAGE II and SAGE III, but we can emphasize that they may contribute to the offsets between the different effective radius products."

Pg 26, Ln 628: "The retrieved median radii and geometric standard deviations should therefore be considered with caution in areas with high aerosol loading."

This is unfortunate as these scenarios tend to be of greater interest to the scientific community. Is there any way to iterate the retrieval and update the assumptions based on other retrieved parameters such as extinction and/or effective radius?

We thank you for this good question. Theoretically, it is possible to use an independent data set as an a priori data set for the 2-parameter retrieval. In practice, there is currently no approach of how the independent data set can be used to create the a priori data set. For example, one could assume that the a priori number density must be greater in areas with enhanced extinction coefficients, e.g., after volcanic eruptions. However, there is no information on how much the number density is to be increased. Even if such an approach can be found, there are other challenges within the retrieval, e.g., the strength of the regularization depends on the particle size or the dependence of the retrieval result on the a priori result. Therefore, we have added the following text passage:

„Retrieving more accurate aerosol characteristics in areas of strong aerosol burden is of great interest to the scientific community. It requires an optimization of the a priori information. Independently observed or simulated aerosol data sets could be used to adapt the a priori aerosol profiles in the retrieval. This supposedly simple approach is challenging for several reasons, some of which are explained below.

First, there is currently no practical approach that describes how an independent data set can be used to adapt the a priori data set. Second, in layers with strong aerosol perturbations, the strength of regularization correlates in particular with the aerosol particle size. Some retrievals therefore require smaller covariance values to keep them stable. How to adapt the a priori covariance depending on the aerosol load is unknown. And thirdly, following from the previous point, the retrieval result may depend significantly on the a priori value if the a priori covariance is chosen too small."

---

## Author Comment (AC3)

Dear Reviewer,
we thank for your very valuable comments. We revised our paper in light of your comments (in black). The answers are shown below in red.
Best wishes,
Pohl et al.

**Overview**

The authors present an improvement on a previous technique for inferring particle size distribution (PSD) parameters from the SCIAMACHY data record. The proposed technique performs a 2- parameter ($r_g$ and $\sigma_g$) retrieval that is not limited to the tropics (an improvement over Malilina et al. 2018). The authors evaluated the impact their assumptions have on the end products and compare their PSD values (including the calculated effective radius ($r_e$)) to those measured directly by the University of Wyoming optical particle counter (UWY OPC) as well as those inferred from the SAGE II and SAGE III/M3M instruments. Further, they used their derived PSD parameters to calculate extinction coefficients (they labeled this Ext, I refer to this a k herein) and compared these to extinction coefficients as measured by the SAGE instruments and OSIRIS.

Overall this is an interesting paper that the community will benefit from. Overall, it is well written and the major claims are more-or-less substantiated (details below) and I believe this should be published if the authors can satisfactorily address the more salient points below.

Finally, I want to congratulate Dr. Pohl and the coauthors for the high-quality work that went into preparing this manuscript.

**Major Remarks**

The stated goal of this work is to extract PSD parameters (for the sake of brevity I include $r_e$ under the PSD label as appropriate) from SCIAMACHY data. This can be best achieved through comparison with the UWY OPC record, but comparison with OPC data is very limited. Instead, the authors devote a substantial portion of the paper to comparing their PSD parameters to those derived from the SAGE missions. Ultimately, this results in an evaluation of the assumptions in each model, which is a distraction from the intended purpose of this work. Direct comparison with the OPC record removes at least half of these assumptions and gets to the heart of the matter. This comparison should be expanded.

We have expanded the comparison of the OPC data with the SCIAMACHY retrievals by separating that in volcanically unperturbed and perturbed situations – please see answer of your comment below for further information (page 7).

An alternative to direct comparison with OPC data is using the SCIAMACHY-derived PSD parameters to calculate extinction coefficient. While the authors did this the evaluation was presented in a bulk-statistics manner (Figures 5 & 8) and it would have been much more informative to expand these figures to include more meaningful information (please see specific comments below). Further, this type of comparison removes the assumptions that went into the SAGE estimates and thereby provides a more robust and meaningful comparison.

We have expanded the comparison of Ext products by additionally showing the

monthly zonal means of the differences depending on the time, latitude, and altitude. The new Figures are Figure 6 and Figure 9 in the revised manuscript. The text has been adapted accordingly (starting from line 577) and also adresses the comparison of Ext products in the post-eruption periods. Analogously, we have also introduced a Figure 9 showing a similar setup for comparing the effective radii.

Another alternative is to use the SCIAMACHY-derived PSD parameters to calculate backscatter coefficients and compare those directly to the numerous ground-based lidars as well as CALIOP. This seems like a grossly overlooked opportunity (a potential gold mine of data) that would have significantly increased the number of intercomparison opportunities as well as the geographic coverage. If this should not be don then can the authors at least address, in the paper, why this should not be done?

A comparison with CALIOP would indeed be very interesting, but does not necessarily add a further value to this already long paper that already contains comparisons with five other reference data sets. We will do this comparison for an upcoming paper investigating the aerosol characteristics after volcanic eruptions. Additionally during much of the SCIAMACHY time period stratospheric aerosol were primarily close to a background state. At these low aerosol loadings, backscatter instruments struggle to tease the signal out of the noise. Vernier et al., 2009 had to do a lot of averaging to obtain scattering ratios characteristic of a clean stratosphere.

Vernier, J. P., Pommereau, J. P., Garnier, A., Pelon, J., Larsen, N., Nielsen, J., Christensen, T., Cairo, F., Thomason, L. W., Leblanc, T., and McDermid, I. S.: The tropical stratospheric aerosol layer from CALIPSO lidar observations, J. Geophys. Res., 114, D00H10, doi:10.1029/2009JD011946, 2009.

There are numerous ambiguities throughout the paper that must be addressed before publication. Without correction the reader cannot understand the presented work and cannot reproduce it.

Finally, the current version of this manuscript suffers from a substantial flaw that prevents the reader from understanding and appreciating the impact and applicability of this work. The authors state that the intent of this method is to infer $r_g$ and $\sigma_g$ from SCIAMACHY data, but limit their discussion of these parameters. Instead, the authors spend more time discussing the comparison to SAGE-derived PSD parameters that are strongly dependent on the assumptions that go into the SAGE algorithm.

We have changed the priority of the comparisons according to your comment, please see the answer to your comment below for further information (page 11).

Further, the reader is not afforded the opportunity to see the overall performance of the SCIAMACHY-derived PSD estimates. Indeed, inclusion of PSD time-series plots (whether line plots with error bars or contour/mesh plots) would communicate a wealth of information to the reader not only on how the PSD parameters changed over the lifetime of the instrument (including volcanic impacts), but would also inform the reader of the stability of this retrieval algorithm. Such a figure would no longer limit the authors to coinciding with other instruments and would allow the authors to display the entire SCIAMACHY PSD record (all within a single figure!).

We have included a Figure (Fig. 3) containing all retrieved and calculated aerosol characteristics from SCIAMACHY period 2002-2012 at an altitude of 18.4 km.

In my view this paper cannot be published without this type of information content. To publish without this information leaves the reader

with a knowledge gap that should not be there. Why is this so important? Because this informs the reader and end-user of how stable and reliable these estimates are. I agree that, in aggregate, comparison with SAGE/OSIRIS is generally good, but there are many unanswered questions. Does the PSD algorithm become unstable at certain altitudes? Does it perform better seasonally?

This will be answered by a new Fig. 1. See answers to your comments below (page 5).

How much do volcanic perturbations influence the PSD estimates? What about wildfires events? Without answering these questions I cannot ascertain the utility of these estimates.

According to the new Figs. 4 (comparison of OPC and SCIAMACHY using profiles of a volcanically unperturbed and perturbed situation)  6, and 9 (time dependent comparisons of Ext and reff from SCIAMACHY, OSIRIS, and SAGE series data), an influence of volcanic eruptions on the SCIAMACHY-obtained aerosol characteristics is now being investigated.

Additionally, volcanic events will be subject of an upcoming paper. We implement an indication in Sect. 8.3:

„Investigating the quality of the retrieved extinction coefficient and effective radius after volcanic eruptions or biomass burning events is appropriate and will be the subject of a subsequent publication."

This is a potentially fantastic paper, but in its current state it is incomplete. For these reasons I recommend that the paper undergo major revisions and be resubmitted for review.

**Specific Remarks**

- page 2, line 44: "catalysers" should be "catalysts"?
  Changed.
- Section 4: There is a lot of repetition within this section. It reads as if it was written twice and never cleaned up. Please consolidate the information and rewrite to be concise and precise.
  We apologize that the section seemed to contain repetitions and present a new structure.
- page 7, line 209: Why a relative humidity of 0%? Does your algorithm have some dependence in RH? This seems important since an RH of 0% is never true.
  A relative humidity of 0% is of course never true but a relative humidity in the stratosphere can be and is often << 1%, effectively 0. To address this comment, we have added the following text:
  „Both, the aerosol composition and the relative humidity, are idealistic assumptions. The percentage of sulphuric acid can vary slightly in reality (Turco et al., 1982, Steele et al., 2003, Doeringer et al., 2012) and the stratospheric relative humidity is usually between 0 (<< 1%) and 10 % (Steele et al., 1981). However, we stick to this convention because the OPAC database does not offer more realistic compositions. The resulting retrieval uncertainty was estimated by comparing retrieved PSD parameters assuming a relative humidity of 0 % and 80 %. The latter value is exceedingly high, but allows a maximum uncertainty estimate. It is below 15 % for the mode radius and below 10 % for the geometric standard deviation (not shown). Note that these values can also be

regarded as an uncertainty estimate due to an incorrect aerosol composition: By increasing the relative humidity, the particles absorb water vapour, which reduces the percentage of sulphuric acid. As a result, the aerosol refractive index (Palmer and Williams, 1975) changes with a similar amplitude to that of an increase in relative humidity (Hess et al., 1998)."

- page 7, line 212: Please define $R_{mod}$.
  The definition of Rmod was in the annex and has been moved to Section 2 (Stratospheric aerosol characteristics).
- page 7, line 212: You stated that $R_{mod}$ and $\sigma_g$ are arbitrarily chosen, but why these specific numbers ($\sigma_g$ of 1.37 is a precise number, why not 1.4)?
  We corrected respective sentence:
  „Both values are based on balloon-borne measurements at background aerosol loadings (Deshler, 2008)."
- page 8, line 220: What is meant by "Above 35 km, the PSD profile remains unchanged."? Please provide a reference to support this claim.
  To address this comment, we have added the following text:
  „Above 35 km, a vertically constant PSD profile is assumed with Rmod=0.11 um and sigma_g=1.37. ... The aerosol parameterizations outside the altitude range of 18 – 35 km might be inadequate. However, they avoid unphysical aerosol size parameters in the lowermost (18 km) and uppermost retrieval height (35 km). Additionally, they have only a minor influence on the aerosol size parameters to be retrieved in between (Malinina et al., 2018)."
- page 8, lines 220–222: You use both $r_g$ and $R_{mod}$ in this sentence. I understand the difference between the 2, but it seems you use them interchangeably here. Please clarify.
  To address this comment, we have added the following text:
  „A vertically constant aerosol size is used as an initial condition. The mode radius is set to Rmod=0.11um. Note the convention used here - the retrieval is controlled externally by Rmod and not by r_g. The geometric standard deviation is set to sig_g=1.37."
- page 8, lines 220–224: Herein you stated that $r_g$ has a lower-limit of 0.05 µm and $\sigma_g$ is not limited. Surely $r_g$ has an upper limit as well as a lower limit. Surely $\sigma_g$ was also limited. Not to be pedantic, but $\sigma_g$ cannot be less than 1...could it, in theory, be >5? Could $r_g$ be 10µm? These values must be fundamentally limited by your model, please provide those limits here.
  Here we discuss mathematical rather than physical limits. This means, that the inversion algorithm is not allowed to produce Rmod values less than 0.05um. There are no maximum limits for sig_g and r_g as such limitations are mathematically useless. In contrast to the minimum r_g limit, the other limits do not help to keep the inversion stable when they are reached.
- page 8, line 225: Here you state that "step 2" solves for 3 parameters ($r_g$, $\sigma_g$, and albedo). Section 2 stated that "step 2" only solves for $r_g$ and $\sigma_g$. Please clarify.
  At the beginning of section 4, we include an additional sentence to describe more thoroughly the second retrieval step:
  „Retrieval errors in the aerosol parameters that may arise from the assumption of a Lambertian surface are mitigated by an adjustment of the surface albedo in the second retrieval step."

– page 9, line 255: Please confirm that the 0.15 and 10.0 μm values are radii and not diameters (newer OPC instruments have a lower limit diameter of ≈150 nm).
We have changed the respective sentence:
„The instrument itself is only sensitive to particles with radii between 0.15 and 10.0 um.“

– page 10, lines 301–302: This is not correct. The grating spectrometer did not include the 1550 nm channel. The 1550 nm channel was an InGaAs photodiode. Please clarify for the reader.
Corrected:
„The solar irradiance was measured by a grating spectrometer at 86 wavelengths from 280 to 1040 nm with a spectral resolution of 1 to 2 nm. An Indium Gallium Arsenide photodiode additionally measured the irradiance at 1550 nm with a bandwidth of 30 nm.“

– page 12, line 356: "Their PSD profiles are specified below." I assume this refers to Table 1. Please include reference to appropriate table or figure.
To address this comment, we have added the following text:
„Their PSD profiles are specified below (Tab. 1 for testing the sensitivity to the Lambertian surface assumption, Fig. 2 black lines for testing the sensitivity to the aerosol number density).“

– Figure 1: I am not colorblind, but I still have a very difficult time reading this figure. This is one of the key figures of your paper. Please update to make it more readable.
Done.

– page 13, line 375: "...with latitudes north of 26°N in summer and 23°S in winter." It is unclear what this means. Do you never go farther south than 23°S? Please clarify.
We completely revised Sect. 6.1 to make this statement clearer.

– Sections 6.1 and 6.2: This is a critical aspect of this paper. Ultimately you do not know the PSD parameters so you cannot definitively calculate the error in your derived PSD parameters that is caused by the Lambertian assumption and the assumed N profile. Therefore, how do you propagate the error from these 2 assumptions into your estimates for $r_g$, $\sigma_g$, and extinction coefficient (e.g., in Fig. 4)? Is this uncertainty ignored for the rest of the paper, or is it accounted for?
We give a summary at the end of Section 6.2:
„To summarise Sect. 6, the PSD parameters r_g and s_g can be accurately retrieved up to a single-scattering angle of about 96°. This corresponds to latitudes north of 26°N in summer and 23°S in winter. Beyond this threshold, limb radiances are less sensitive to aerosols making it difficult to retrieve the PSD parameters separately. They can be subject to large uncertainties and should therefore be treated with caution. In contrast, the accuracy of r_eff and Ext depends only slightly on the single-scattering angle. The two aerosol characteristics have reasonable results for both hemispheres.
The PSD retrieval is sensitive to the assumption of a Lambertian surface and the a priori number density profile. The latter effect exceeds the former one. However, the effect of the Lambertian surface assumption can only be calculated for ideal cases, i.e. homogeneous surface types. Moreover, the spatio-temporal distribution of the stratospheric aerosol number density is essentially unknown in reality. The SCIAMACHY retrieval has to rely on assumptions here that lead to errors in the

retrieved and calculated aerosol characteristics. A quantitative error estimation of both assumptions, the Lambertian surface and the a priori N profile, is therefore not possible for real retrievals. We can only point to these sources of uncertainty."

- page 14, lines 403–404: Here you state that you averaged over all angles from 20°– 96°. If I understood Section 4 correctly the variation in scattering angle has no impact on this process. If that is correct then please ignore this comment. If there is a dependence then would you please clarify here.
  To address this comment, we have added the following sentences: „According to [the new] Sect. 6.1, a separate retrieval of r_g and s_g at single-scattering angles larger than 96° is challenging due to a reduced sensitivity of limb radiances to PSD parameters. Therefore, they are not included in Fig. 2. The lower angular limit is based on instabilities that occurred during the retrieval at smaller single-scattering angles."

- Figures 2 & 4: Do I understand this correctly that there is no variability in the inferred number density? You should be able to infer N as well (I thought that's what you did in Figure 7), so I expect some spread in the profiles. Please clarify.
  N is not retrieved. The spread in Fig. 8 originates from the reference product:
  „The differences in N show a broad distribution (Fig. 8 (e,f)). This is due to the variability of the retrieved N from SAGE III since the N profile from SCIAMACHY is invariant. According to the distribution width, a fixed N profile from SCIAMACHY seems to be questionable, because in some cases, it can be more than twice as large or small than the retrieved N profile from SAGE III. As a result, the differences of rg (Fig. 8 (a,b)) and σg (Fig. 8 (c,d)) also show a significant spread, albeit less than in N."

- page 15, line 413: "…correct selection of the a priori N is crucial…" That's not how I interpret Figure 2. It looks like the algorithm is sensitive to the N profile and how close it is to reality. However, "crucial" may be an overstatement. The $r_g$ is mostly within +/-25% up to ≈30 km. The $r_e$ is even better. I think the text, as written, misleads the reader. Please revise.
  To address this comment, we have added the following text: „To conclude, the retrieved (r_g, s_g) and calculated aerosol characteristics (r_eff, Ext) depend on the assumption of the a priori N profile. The more correct this assumption is, the more precisely the aerosol characteristics can be retrieved."

- page 15, line 418: "…latitudes south of 26°N in summer and 23°S in winter." Please see similar comment above.
  See answers above.

- Figure 3: Please consider plotting panel (a) on a log scale. Maybe not necessary, but it helps readability. If the authors disagree then please disregard this comment.
  Done.

- page 16, line 424: "2002 and 2011" Why not "2002 and 2012"?
  This was a mistake. The data set has been extended to 2012.

- page 16, line 426: "…these processes are evidenced in Fig. 3." This level of information cannot be ascertained from a simple extinction coefficient plot. Please revise.
  To address this comment, we have added the following text:

„Figure 3 shows the results at 18.4 km altitude. The changes in aerosol characteristics after volcanic eruptions are particularly striking. The injected masses usually increase r_g, r_eff, and Ext, and reduce s_g. Their temporal developments are determined spatially by advection and microphysically by nucleation, coagulation, condensation, and sedimentation."

- Figure 4: How were the relative error statistics calculated? Did you first calculate the relative errors then average, or did you calculate the average $r_g$ then the relative error?
  To address this comment, we have added the following text:
  „Relative errors are calculated as (SCIAMACHY – balloon) / balloon x 100% before averaging."

- Figure 4: This is possibly the most important figure in this paper (comparing to OPC) and should receive more attention. You put all of the data into one plot (I realize the OPC data are sparse), but this leaves me wondering if information is lost in the bulk statistics. Did volcanic activity significantly change the performance of your method? Breaking this analysis into 2 paradigms (volcanically perturbed/not-perturbed) would be illuminating. Without this information it is difficult to realize the value of this method.
  We thank you for the valuable comment. Splitting [old] Fig. 4 in perturbed & non-perturbed profiles is a good idea. However, it is difficult to realize for the following reasons:
  (1) Out of 23 profiles, only 4 contain volcanic plumes. Two of those are measured at volcanic plume edges and are not strongly influenced by volcanic aerosols. A statisticial analysis of the volcanically perturbed profiles for such a small set of data is not very meaningful.
  (2) We have noted that the small number of volcanically perturbed OPC profiles are not sufficient for a meaningful statistical analysis. This is especially the case when the balloon OPC recorded aerosols are at the cloud edge. Consequently, the comparison with the collocated SCIAMACHY profile should be treated carefully due to a) spatio-temporal mismatch (<12 hours, <750 km) b) the different measurement footprints of the instruments. These issues have to be taken into account.

  That is why we decide to show only a small comparison. We have included an additional Figure in the revised manuscript (new Fig. 4), which shows the comparison of two balloon-profiles with SCIAMACHY retrievals. One profile was measured during aerosol background condition (7 May 2005), one profile was measured after the Sarychev eruption (7 Nov 2009). The text in lines 495 – 531 explains the differences between the volcanically perturbed and unperturbed profiles.

  A much more detailed analysis about the behaviour of SCIAMACHY retrievals in post-volcanic eruption periods is in progress and is the subject of a planned subsequent paper.

- page 17, lines 441–442: "…and one based on balloon-borne measurements over Wyoming before 2002…" This is confusing. This sounds like you use an OPC-based climatology as your N profile, which has the potential to significantly bias your evaluation. This raises several critical questions regarding the methodology.

1. Are you now using an OPC-based climatology for your N profile? If so, did you also use this climatological profile in creating Figure 2?
   Yes, this is now explicitly stated.
2. You stated this N profile is based on WY OPC data collected before 2002. Does this include data collected throughout the entire record (i.e., back to 1971)? If so, how do you handle the differing OPC instruments? How do you handle extreme outlier events like El Chichon and Pinatubo? How did these events influence the overall performance of your algorithm?
   We explicitly stated the length of the time series:
   „... based on balloon-borne measurements over Wyoming between 1989 and 2001.“
   The large eruption of Mount Pinatubo is within this period. We averaged all profiles within the time frame, thus the impact of Mount Pinatubo is limited. In addition, some outliers in the averaged profile were manually filtered to get a smoothed profile. The main goal of this approach was to create an aerosol profile which might be representative of reality. To avoid correlations between the created number density profile and the profiles to be retrieved, we took the precaution of omitting the period 2002- 2012 when creating the profile.
3. From Figure 4, it looks like the OPC-based N profile yields better performance than the ECSTRA profile (better errors, the profile shape is in better agreement with reality, etc.), yet you never state which N profile you use in the operational algorithm. I assumed you use the ECSTRA profile, but Figure 4 indicates the OPC profile is better. Please clarify.
   In Sect. 4, it is now stated at the beginning:
   „... the number density profile is based on the ECSTRA model climatology for aerosol background conditions (Fussen and Bingen, 1999).“
- page 18, lines 448–449: “...which can be explained by a small reference value (Fig. 4 (i)).” Did you mean Fig. 4 (d)? Also, I'm not sure Figure 4 supports this claim. Extinction coefficients on the order of 1E-4 (i.e., z ≤24 km) are not small. What you are dealing with here is, in fact, relatively large differences in the derived extinction coefficients.
   We have added the following sentences:
   „Relative errors in Ext can exceed 100 % (Fig. 5(i)). This large value is a result of the calculation method which is not robust against outliers. The median of the relative errors is below 20 %. Further, the relative errors in Ext depend only slightly on the choice of the a priori N profile.“
- Figure 5: You allude to this figure throughout the text and reference statistics from this figure. However, I cannot read this figure. There are so many colors on top of each other I cannot tell where they all begin/end...and I cannot tell if the quoted numbers quote are correct (I assume they are, I just cannot verify). Please improve the readability of this figure.
   Done. Instead of using shading areas, we use error bars. Those are slightly shifted vertically for better readability.
- Figure 5: Why the stark difference between this figure and Figure 4 (i)?
   There is no stark difference. In [new] Fig.5, the relative error is calculated with respect to the OPC. In [new] Fig. 7, the relative difference of two

observations is given with respect to the average of the two observations. The reason is given in the next comment.

- Figure 5: Why not use the same method to calculate error as used in Figure 4?
  To address this comment, we have added the following sentences: „Note that in contrast to Figs. 2 and 5 the calculation of the difference is changed here and in the following figures. This is because we do not know which satellite data product is correct. We therefore now refer to deviations between the products, instead of calculating errors by using one satellite data product as the 'true' reference.“

- page 19, lines 489–490 "…with the differences decreasing with altitude." This is not what Figure 6 shows. All differences increased (except panels f and g). Please clarify.
  The referee is refering to the new Fig. 10, where indeed all the differences become smaller with altitude between 40°S and 40°N. We have corrected the sentences:
  „The effective radii from SCIAMACHY are systematically lower than those from SAGE II and SAGE III. At 31.5 km altitude, $r\_eff$ from SAGE II and SCIAMACHY agree well with differences below 17.7 % at latitudes from 40°N to 40°S and below 43 % at higher latitudes (Fig. 10). Best agreement is achieved in the tropics. The differences becoming larger with decreasing altitude south of 40 °N due to a faster increase of $r\_eff$ from SAGE II compared to SCIAMACHY. The reason is still unknown. The altitude dependency is most pronounced in the tropics. Here, the differences can increase up to 45.6 % (v7.0 NASA) and 57.0 % (DWE). The differences at 18.4 km seem to be independent of the volcanic perturbation (Fig. 9).“

- Figure 6: This figure exemplifies why the comparison with OPC (or even a comparison of k) is of more value than comparing with the SAGE-derived PSD parameters. The extreme slope in panels b–e indicates a systematic error in the SAGE PSD values. How is the reader to draw meaning from this analysis? Can the authors account for this extreme slope?....
  Please see comment above.
  ….To be honest, the extinction comparison looked promising, so I was somewhat shocked to see this odd behavior in these profiles (I do note that the authors plotted here data from both SAGE missions and that this is a figure for $r_e$ (a derived product that is based on derived products, which adds another layer of obfuscation)). Given the ambiguities, it is unclear how this figure is helpful, especially since the intended purpose of this work is to extract $r_g$ and $\sigma_g$ from SCIAMACHY…not $r_e$.
  While the intended first purpose of this work is to derive $r\_g$ and sigma\_g from the SCIAMACHY data a secondary purpose is to then use those quantities to derive useful geophysical parameters such as $r\_eff$, and then it is natural to compare such results with other derivations of similar quantities. This figure is just to illustrate such comparisons and their differences.
  To address this comment, we have added the following text:
  „Best agreement is achieved between SCIAMACHY and SAGE III with deviations of 1.3 to 17.9 %. This is remarkable when one considers the large differences in the PSD parameters (Fig. 8). This is the advantage of comparing reff. Firstly, its calculation is independent of N according to Eq. (4). Secondly, the anti-correlation of rg and σg compensates the

uncertainties in reff.“
- page 20, line 501: “...utilizes a fixed number density profile...” What is this profile? Is it the OPC-based climatological profile, or the ECSTRA profile?
To address this comment, we have added the following sentence:
„The former utilizes a fixed number density profile based on the ECSTRA model climatology.“
- page 20, line 509: “...can be by up to 51.1%.” I have 2 points. First, should this be “...can be up to 51.1%”? Second, the error in N (including error bars) goes well beyond 51% (maybe even over 100%). Please clarify.
We incorporate the word „mean“:
„The mean deviation in the a priori N can be up to 51.1 %.“
- Section 7.3 and Figure 8: It is unclear how the 1989–2002 time period is relevant to this study. This seems like wasted space and text. Figure 8 would be much more meaningful to the reader if the authors were to remove the 1989–2002 period (this would also allow them to use consistent scales) and create a single time series (i.e., 2002–2012). The figure could be further improved by breaking it into 2 figures (1 for extinction, 1 for $r_e$), with each figure containing sub-plots for different altitudes (e.g., panel (a) could be 30 km, panel (b) is 25 km, etc.). The value of doing this is it shows the reader the relevant information and provides the reader a much better understanding of the performance of this algorithm at multiple altitudes. Finally, showing this as a time series (instead of the aggregate profile statistics) allows the reader to appreciate the influence volcanic perturbations have on this method.
Done. We revised Sect. 7.3 – which is now chapter 7.4.
- Figure 8: It seems the first legend (titled “Symbols: Comparison of”) is unnecessary. There is no “comparison” plotted in these panels, so it is unclear what the legend title means. The 2 legends could be consolidated to make the figure more easily interpreted.
Done.
- page 23, lines 532–533: “...due an increasing median radius with a simultaneously decreasing geometric standard deviation.” First, should this be “...due to an...”? Second, this claim is conjecture and is not supported by the analysis. Since you are dealing with inferred values at this point I suspect that everything you see here, including the changes in $r_g$ and $\sigma_g$, are symptoms as opposed to the root cause. Can the authors provide additional support for this claim or reword this sentence?
We revised the sentences:
„A slight but significant upward trend in the effective radius from SAGE III can be observed especially at the altitude of 21.7 km. This comes along with an increasing median radius and a decreasing geometric standard deviation (not shown).“
- page 23, line 535: What is the constant in the SCIAMACHY process? Undoubtedly it is N, but which N (UWY OPC or ECSTRA) is it?
To address this comment, we have added the following text:
„A possible reason might be that in all three retrieval algorithms one of the PSD parameters is assumed to be constant, namely N_ECSTRA in the SCIAMACHY retrieval, the total N of 20 1/cm³ in the v7.0 NASA retrieval, and s_g=1.5 in the DWE approach.“
- Figure 9: Again, this figure is an excellent example of why the authors

should make the comparison with the UWY OPC products their first priority and give this analysis the most weight. This is also why the second priority should be comparing SCIAMACHY-derived extinction coefficients with SAGE/OSIRIS extinction coefficients should receive more weight and be second priority. This figure clearly demonstrates that the comparison is dominated by the assumptions in the SAGE estimates.

We have changed the priority of the comparisons according to your comment:

→ Sect. 7.1 Comparison with balloon-borne measurements changes according to the comments above.

→ Sect. 7.2 Comparison of satellite retrieved aerosol extinction coefficients
We include the following text at the beginning:
„Comparisons of satellite data products include data from a large spatial and temporal range. However, they have a decisive disadvantage compared to the comparisons with balloon-borne OPC measurements in Sect. 7.1: Similar to the SCIAMACHY v2.0 aerosol product, the reference satellite data sets cannot be measured directly, but are retrieved from the satellite-measured radiances. Those retrievals are themselves subject to uncertainties, which creates an additional layer of ambiguity. A difference between two satellite retrieved aerosol products does not allow any conclusions to be drawn as to which product is the more accurate. In order to limit ambiguity, this section is restricted to the comparison of aerosol extinction coefficients. Here, most of the reference data sets are retrieved directly (Sect. 5). Section 7.3 then deals with the comparison of aerosol sizes. In this case, the reference data sets are obtained from aerosol extinction coefficients, i. e., they are secondarily retrieved data products, which add another layer of ambiguity."

→ Sect. 7.3 Comparison of satellite retrieved aerosol size parameters
We include the following text at the beginning:
„We now focus on the comparison of satellite retrieved aerosol size parameters, i. e., the PSD parameters and the effective radius. As already mentioned, this comparison uses secondarily retrieved size parameters as reference data sets. They are subject to uncertainties caused by two retrievals, firstly, that of Ext and, secondly, that of the PSD parameters or reff from Ext. Thus, differences between the aerosol size parameters from SCIAMACHY v2.0 and those from the reference data products may be larger than the differences in the aerosol extinction coefficients."

→ Sect. 7.4 Temporal comparison
We include the following text at the beginning:
„We now focus on the temporal evolution of the aerosol extinction coefficient and effective radius. This is shown in Fig. 11..."

→ Concerning Fig. 12:
We include the following text:
„To conclude, Fig. 12 clearly demonstrates that the comparison of the effective radii is dominated by the a priori retrieval assumptions. Those may slightly distort the retrieval data."

- page 27, lines 648–652: I have several points about this text.
  1. I saw nothing in the paper to indicate that "the median radius and geometric standard deviation are fully reliable only in the northern hemisphere." (emphasis mine). If this was in text then I sincerely apologize. Since the authors are limited to UWY OPC data (i.e., northern hemisphere) for validation I don't know how you could determine this. Please clarify.
  2. If the southern hemisphere SCIAMACHY PSD parameters are "bad" then how do you justify inferring extinction coefficient and $r_e$ from them? It would seem you are getting a more-or-less right answer for the wrong reasons. Please clarify.
  3. The previous 2 points seem to be contradictory. Please clarify in the text.
  4. The authors stated that the intent of this method is to derive PSD parameter ($r_g$ and $\sigma_g$) from SCIAMACHY data…and do so globally. If the southern hemisphere $r_g$ and $\sigma_g$ are, per the authors' statement, unreliable then has the intent of this work failed? If so, the abstract must be updated to reflect the limited applicability of this method.
     To address this comment, we have added the following text:
     „The median radius and the geometric standard deviation are accurately retrieved for single-scattering angles smaller than 96°, i.e., at latitudes north of 26°N in summer and 23°S in winter (Fig. 1). At larger single-scattering angles, limb radiances are less sensitive to aerosols. That leads to increasing uncertainties in the retrieved PSD parameters which should be treated with caution. The extinction coefficient and the effective radius benefit from the anti-correlation of the uncertainties - while the median radius is underestimated, the geometric standard deviation is overestimated and vice versa. They can therefore be retrieved satisfactorily in both, the northern and southern hemispheres."
- page 27, line 666: The "link" must be updated.
  Done.
- Appendix A: While this appendix is interesting, I fail to see how it makes a substantive contribution to the paper and should be removed (unless the authors can justify its inclusion, of course).
  It is deleted.

---

## Referee Report (RR1)

I want to thank Dr. Pohl and her coauthors for the work they have done in writing this paper. Further, I want to thank them for the work that went into this revision. Finally, I want to thank them for putting up with my myriad questions and comments.

**Overview**

This is a challenging paper to review because of its size and the complexity and detail of the analysis. There are multiple variables at play throughout and correctly interpreting the results requires considering the data from multiple facets. With that in mind I readily acknowledge that I may have miss interpreted something, which has led to my confusion. However, I must also admit that I struggle to see the utility of this method for the reasons below.

**Major Issues**

This methodology is highly sensitive to the assumed, a priori, number density profile as demonstrated in Fig. 2. Here, the authors showed that changing the a priori N profile (by factors of 0.5 and 2) changes $r_g$ by $\approx \pm 30\%$, $\sigma_g$ by $\approx \pm 6\%$ (no big deal), $r_e$ by $\approx \pm 15\text{-}20\%$, and extinction by -45% to +140% (it is interesting to note that this scaled nearly linearly). Granted, all bodes well when the a priori N matches current conditions (as the authors demonstrated in Fig. 5 and elsewhere). However, under volcanically active conditions it is entirely reasonable that the a priori is more than a factor of 2 different (looking at the Wyoming OPC record I see changes in excess of a factor of 10 after eruptions within the SCIAMACHY time period). Taken to an extreme, how would this method perform after Pinatubo or Hunga Tonga?

While the authors evaluated the influence of an incorrect a priori (via Fig. 2) there remains 1 glaring shortcoming of the method: the "real" N profile is unknown therefore we do not know how much uncertainty this introduces to the retrieval and we cannot quantify the uncertainty of the inferred PSD values and the derived extinction coefficients. What we do know is that this uncertainty *can* be substantial. If the authors were to limit their analysis to conditions when N is stable then they could make reasonable guesses for their a priori (that's basically what they do here, using the OPC record). However, that is not interesting. The interesting bits are in the post-eruption atmosphere when the N profile and PSD parameters are **most** dynamic! In short, we know that the PSD parameters and extinction coefficients as derived from SCIAMACHY data are wrong... but we have no gauge for how wrong they are. Unfortunately it's not *just* the quantitative results that are suspect but we must also suspect and qualitative interpretation of the data as well.

I really like this paper so this leaves me with a dilemma. I am left questioning how I would use this data and what is the ultimate purpose of this paper (i.e., what does the community now know that we did not before). I think what we now is this: PSD parameters can be inferred from SCIAMACHY data and these inferred parameters have modest sensitivity to the a priori N. The calculated extinction is much more sensitive. However, given the extreme range of N in the Wyoming OPC record after major eruptions within your time period I have to conclude that this methodology is useful only during stable/background conditions and not reliable in the aftermath of volcanic eruptions. This limitation is systemic throughout the paper and is inherent within the methodology itself and I currently see no path to salvaging it. It is for this reason that I cannot recommend this paper for publication.

I recognize that my view may be in the minority and, should the editor decide to allow publication, then I fully support him in this decision.

**Specific and Minor Comments**

- **page 7, line 197:** "unambiguity" should be "ambiguity"?

- **page 9, lines 268–269:** "the retrieved PSD parameters and the assumed number density are used to calculate the effective radius (Eq. (5)) and the extinction coefficient (Eq. (6)) of the aerosol particles" Earlier in the manuscript you stated that N can be fixed through space/time because N plays a minor role in the retrieval process (fair enough). However, here you see how N plays a crucial role in calculating some derived parameters (especially extinction). Perhaps a statement regarding this dependence is appropriate here.

- **page 10, line 283:** "classes" should be "bins"?

- **page 17, lines 475–479:** "A quantitative error estimation of both assumptions, the Lambertian surface and the a priori N profile, is therefore not possible for real retrievals. We can only point to these sources of uncertainty." This is an accurate statement and it is highly unfortunate. In my view, this is the dominant shortcoming of this method: you know the numbers are wrong but you don't know by how much. This uncertainty will be more pronounced immediately after major events (N can change by a factor of 10 or more, which is FAR more than the "doubling" you modeled). It may be necessary to explicitly tell the reader of this shortcoming post-eruptions.

- **page 18, lines 483–484:** In what way is this "striking"? Fig. 2 showed that underestimating number density (yellow line) results in over estimation of particle size and under estimation of distribution width. This results in an over estimation of approx 100% in the extinction coefficient and an over estimation of approx 20% for $r_e$. Undoubtedly, the a priori N value in your model is too low after these eruptions, which puts you squarely in the situation I just described (i.e., over estimation of $r_g$, over estimation of extinction, etc.). I don't doubt that extinction increased, I don't doubt that particles became bigger, and I don't doubt that distribution width decreased. However, particles do not always get bigger after eruptions as some of your co-authors have demonstrated (https://acp.copernicus.org/articles/23/9725/2023/). Therefore, this leaves the reader wondering how much of the variabilty shown in Fig. 3 is a by-product of a wrong number density. Given the level of uncertainty in this method I do not believe that you can say that your data unambiguously proves (much less quantified) these changes occurred. Undoubtedly changes are expected, but at this point I think that the most defenceable statement that can be made, based on your product, is that things changed. . . by some amount.

- **page 19, Figure 4 caption:** The OPC record reports 2 modes (fine and coarse mode). Did you use both modes in calculating the extinction coefficient? The coarse mode can have a disproportionate impact on extinction.

- **page 19, Figure 4:** Panels (d) and (i) do not make sense. Why do the red and blue lines cross each other at ≈18 km at not at ≈20 km (i.e., where the 2 a priori N lines cross each other in panels (c) and (h))? All other panels have the red/blue intersection at the same altitude so why are (d) and (i) different?

- **page 19, Figure 4 caption:** What about the light red and light blue colors? Can you define those here so the reader need not search the text for the explanation?

– **page 20, lines 522–524:** "Remarkable are the similar profile shapes from SCIAMACHY and OPC on 7 November 2009 in case of $r_g$ and $\sigma_g$ by assuming the a priori N based on balloon-borne measurements. This is due to the similarity of the SCIAMACHY-assumed and OPC-measured N profiles." This is no surprise (as the authors state, this is due to the extreme similarity between the current OPC N profile and the climatology, that was based on OPC data). What this tells me, yet again, is that the profile is entirely dependent on the a priori N profile.

– **page 20, lines 526–527:** "good agreement of the extinction coefficient from OPC and SCIAMACHY, regardless of the assumed a priori number density (Fig. 4(d,i))" I disagree on the interpretation of this figure, but I admit that I am struggling to find an interpretation and more information would be helpful. Panel (d) certainly looks promising (SCIAMACHY and OPC are in good agreement) and I would be surprised if it were bad. However, panel (i) is less impressive. While the shapes are in good agreement the OPC extinction is larger by a factor of 2-3 (is this what is meant by good agreement)? Undoubtedly all of this variability is driven by differing number densities. However, it is important for the reader to understand how the OPC extinction coefficients were calculated here: did you only use the first mode or both modes? If you only used the first mode then, especially after an eruption, we can reasonably expect the second/coarse mode to be enhanced and have a disproportionate impact on extinction. Here is the point: currently the difference between SCIAMACHY ext and OPC ext is 200%-300%, if you include the second OPC mode in calculating ext the difference will become larger and this does not qualify as "good agreement".

– **page 20, line 528:** "...the three PSD parameters remain consistent with each other." I do not understand what is meant by "consistent." Can you please clarify?

– **page 21, Figure 5 caption (and corresponding text):** "...relative errors..." This assumes that the OPC is correct, which it is not. You stated on the previous page that disagreement between OPC/SCIAMACHY was driven, in part, to differing sampling volumes and because the OPC only sampled the edge of the aerosol plume while some of the SCIAMACHY profiles were collected within the plume. Should be reworded to "percent difference" (or something comparable) here and throughout the text.

– **page 21, lines 539–540:** "Since SCIAMACHY is not sensitive to stratospheric aerosols with $r_g$ ¡ 0.06 $\mu$m (Malinina et al., 2019), corresponding OPC PSDs are excluded from the comparison" This is not correct and these smaller $r_g$'s should be included. They will not contribute much to extinction at 750 nm, so no big deal, but, as written, you are not doing a valid comparison.

– **page 21, lines 547–548:** "...Ext depend only slightly on the choice of the a priori N profile." This is not what Fig. 2 tells me; N is highly important.

– **page 22, lines 566–567:** "...to obtain a sufficient number of collocations." Sufficient for what? You now have 4255 coincident profiles...how many did you have when the maximum distance was only 200 km?

– **page 23, Figure 6:** This behavior is exactly what I expected based on Figure 2.

– **page 24, line 585:** "Discrepancies are smaller in the middle latitudes. . ." Yes, this makes sense because these latitudes are the least impacted by volcanic activity. Therefore, your a priori N profile is more similar to the N profile of these latitudes.

– **page 34, lines 774–775:** "The retrieved median radii and geometric standard deviations should therefore be considered with caution in areas with high aerosol loading." This is true for extinction as well.

---

## Author Response (AR2)

Dear Reviewer,
we thank for your suggestions, comments, questions, and for your criticism. Here, we would like to clarify and address your main criticisms (blue). This is then followed by revisions of our paper (red) made in light of your comments (in black).
Yours sincerely,
Christine Pohl, on behalf of all co-authors

1) Uncertainty in Ext can exceed 100 %.
→ The reviewer bases his/her comments and conclusions in particular on Fig. 2, which shows uncertainties between -45 and 140 % in the aerosol extinction coefficient. We thank the reviewer for motivating us to focus on these numbers because in rechecking them we noticed a mistake in this Figure. The wrong number density profiles that we used to calculate the extinction coefficients led to the large uncertainties. After the correction, the uncertainties in the extinction coefficients reduce to -15 - + 25 % (see revision below). Although this does not change the other results of the manuscript or the reviewer's comments, the magnitude of the range of uncertainty is much smaller than that addressed by the reviewer. In particular, we disagree with the reviewer's statement that the uncertainties of the extinction coefficients are greater than 100% (see revision below).

2) The number density N has to be assumed.
→ No remote sensor can measure or retrieve N without making any assumption. Even for the occultation technique, where the assumptions are not so strong, the overall shape of the particle size distribution (typically unimodal lognormal) and shape and composition of the aerosol particles (typically spherical particles consisting of sulfate acid) needs to be assumed.
In addition the lack of sensitivity for particles smaller than ~ 10 nm might be an issue.

In general, there are only a few in situ instruments that can measure N, one balloon-borne and 2-3 air-borne. Remote sensors using the scattering technique can be used to infer N by assuming some other parameter of the particle size distribution, such as median radius and/or distribution width. This, however, does not eliminate the uncertainty related to one fixed parameter. In terms of the used assumptions, the SCIAMACHY retrieval is not significantly different from any other retrieval for remote sensing data. Thus disqualifying SCIAMACHY based on this limitation would also disqualify many other published and well-recognized retrieval algorithms.

3) The number density N of the Wyoming OPC can vary by up to a factor of 10.
→ The measurements made indicate that N does not vary by a factor of 10. This is clearly shown in the attached plots (see p. 10-11). A variation of 0.5 - 2.0 is a better assessment of the observed range. A paper was recently submitted to JGR which reviews the history of total aerosol measurements from Wyoming (Norgren et al., in review: Measurements of total aerosol concentration in the stratosphere: a new balloon-borne instrument and a report on the existing measurement record). A figure in that paper shows that at 500 K +/- 20 K, except for 5 measurements out of 132, there were no

**Major Issues**

This methodology is highly sensitive to the assumed, a priori, number density profile as demonstrated in Fig. 2. Here, the authors showed that changing the a priori N profile (by factors of and 2) changes $r_g$ by ≈ +-30%, sig_g by ~ +-6% (no big deal), r_e by ~ +- 15-20 %, and extinction by -45% to + 140 % (it is interesting to note that this scaled nearly linearly).

We apologize here, a mistake was made when calculating the extinction coefficients. Instead of using the N_ECSTRA profile (blue line) which was assumed for the retrieval, the „true" N profiles (coloured lines) were mistakenly used to calculate the retrieved Ext. Ext is the product of the particle number density and the particle scattering cross section integrated over the number size distribution. This leads to the linearly scaled uncertainties in the retrieved Ext.

Granted, all bodes well when the a priori N matches current conditions (as the authors demonstrated in Fig. 5 and elsewhere). However, under volcanically active conditions it is entirely reasonable that the a priori is more than a factor of 2 different (looking at the Wyoming OPC record I see changes in excess of a

factor of 10 after eruptions within the SCIAMACHY time period). Taken to an extreme, how would this method perform after Pinatubo or Hunga Tonga?

N does not vary by a factor of 10. This is clearly shown in the attached plots (p. 10-11). A variation of 0.5 - 2.0 is much more indicative of the record. So we are not sure which part of the record the reviewer is referring to to make that statement. A paper was recently submitted to JGR which reviews the history of total aerosol measurements from Wyoming (Norgren et al., in review: Measurements of total aerosol concentration in the stratosphere: a new balloon-borne instrument and a report on the existing measurement record). A figure in that paper shows that at 500 K +/- 20 K, except for 5 measurements out of 132, there were no changes > about a factor of 2, including the Pinatubo period. For the SCIAMACHY period there were 3 such measurements between 2007 and 2009, but these profiles provide „only" 4 times greater number densities than assumed.

While the authors evaluated the influence of an incorrect a priori (via Fig. 2) there remains 1 glaring shortcoming of the method: the "real" N profile is unknown therefore we do not know how much uncertainty this introduces to the retrieval and we cannot quantify the uncertainty of the inferred PSD values and the derived extinction coefficients. What we do know is that this uncertainty can be substantial. If the authors were to limit their analysis to conditions when N is stable then they could make reasonable guesses for their a priori (that's basically what they do here, using the OPC record). However, that is not interesting. The interesting bits are in the post-eruption atmosphere when the N profile and PSD parameters are most dynamic! In short, we know that the PSD parameters and extinction coefficients as derived from SCIAMACHY data are wrong. . . but we have no gauge for how wrong they are.

Yes, but in these conditions all instruments are struggling with limited information to help. And because of the limited range of variability of N, the retrieval errors will be smaller than anticipated by the reviewer (see new sensitivity tests in the current version of the manuscript).

Unfortunately it's not just the quantitative results that are suspect but we must also suspect and qualitative interpretation of the data as well.
I really like this paper so this leaves me with a dilemma. I am left questioning how I would use this data and what is the ultimate purpose of this paper (i.e., what does the community now know that we did not before). I think what we now is this: PSD parameters can be inferred from SCIAMACHY data and these inferred parameters have modest sensitivity to the a priori N. The calculated extinction is much more sensitive. However, given the extreme range of N in the Wyoming OPC record after major eruptions within your time period I have to conclude that this methodology is useful only during stable/background conditions and not reliable in the aftermath of volcanic eruptions.

As already mentioned above, most of the balloon-borne recorded N is quite stable. Accordingly, the methodology is not only applicable in stable conditions, but also in the aftermath of some volcanic eruptions and biomass burning events.
We investigated the influence of volcanic eruptions on the SCIAMACHY PSD retrieval based on a newly created synthetic data set. We added the following

text to the manuscript:

To conclude, the retrieved (r_g, σ_g) and calculated aerosol characteristics (r_eff ) depend on the assumed a priori N profile. The closer this assumption is to reality, the more precisely the aerosol characteristics can be retrieved. However, the number density varies in the reality to an unknown extent. Therefore, it is impossible to quantitatively estimate the retrieval uncertainty caused by the number density assumption. We can only provide uncertainty limits.
Regarding the balloon-borne OPC measurements from 2002 to 2012, a variation of N by a factor between 0.5 and 2.0 encloses ≈ 80% of the variation observed in the Wyoming record. This means the above mentioned uncertainties are representative for most of the SCIAMACHY record. In the remaining 20% of cases, the balloon-borne OPC record reveals number densities in around 18 km altitude that are about four times larger than the N_ECSTRA profile assumed in the SCIAMACHY retrieval. They originate from major volcanic eruptions of Tavurvur (2006), Kasatochi (2008), and Nabro (2011).
In the aftermath of volcanic eruptions, an underestimation of the a priori N profile by a factor of four does not necessarily lead to a doubling of the above mentioned uncertainties. The latter rather depend on the true PSD of the aerosol plume, i. e., the interplay between r_g, σ_g, and N. We repeated the simulations using the altitude-dependent aerosol profiles shown in Fig. 2 (black lines) but perturbed the profiles below 25 km altitude according to OPC measurements and SAGE III/ISS retrievals (Wrana et al., 2021) in post-eruption periods. The number density at 18.4 km altitude was four times larger than the assumed a priori number density N_ECSTRA. We considered cases with increasing and decreasing volcanic particle size. In the best case, the retrieval uncertainties at 18.4 km are at 30% (r_g), -8% (σ_g), and 18% (r_eff ). In the worst case, they are twice as large. Uncertainties in Ext are between -70 and 50% (not shown).
…
To summarize […] the spatiotemporal distribution of the stratospheric aerosol number density is unknown. The SCIAMACHY retrieval uses assumptions that lead to errors in the derived aerosol characteristics. Radiative transfer simulations where the a priori and true N differ by up to a factor of four imply uncertainties of +-30% (r_g), +-8% (σ_g), and +-20% (r_eff) during volcanically quiescent and some post-eruption periods. However, uncertainties in volcanic plumes can also double depending on the PSD. Ext has an uncertainty of +-25% during volcanically quiescent periods and of -70 - 50% during post-eruption periods.

This limitation is systemic throughout the paper and is inherent within the methodology itself and I currently see no path to salvaging it. It is for this reason that I cannot recommend this paper for publication.
I recognize that my view may be in the minority and, should the editor decide to allow publication, then I fully support him in this decision.

We hope we have explained the intrinsic need for a priori knowledge to constrain the retrievals of the aerosol PSD. We have quantified the size of the error and use the best available knowledge from in situ climatologies to

constrain the retrievals.

**Specific and Minor Comments**

– page 7, line 197: "unambiguity" should be "ambiguity"?
Corrected.
– page 9, lines 268–269: "the retrieved PSD parameters and the assumed number density are used to calculate the effective radius (Eq. (5)) and the extinction coefficient (Eq. (6)) of the aerosol particles" Earlier in the manuscript you stated that N can be fixed through space/time because N plays a minor role in the retrieval process (fair enough). However, here you see how N plays a crucial role in calculating some derived parameters (especially extinction). Perhaps a statement regarding this dependence is appropriate here.

We have added the following text:
Note that both the retrieved ($r_g$, sigma_g) and the calculated parameters ($r_{eff}$, Ext) may slightly depend on the choice of the a priori number density profile. However, it will be shown in Sect. 6 that the strong correlation between the PSD parameters can compensate for retrieval errors in the calculated parameters, provided that the a priori number density profile does not deviate considerably from the true profile.

– page 10, line 283: "classes" should be "bins"?
Corrected.
– page 17, lines 475–479: "A quantitative error estimation of both assumptions, the Lambertian surface and the a priori N profile, is therefore not possible for real retrievals. We can only point to these sources of uncertainty." This is an accurate statement and it is highly unfortunate. In my view, this is the dominant shortcoming of this method: you know the numbers are wrong but you don't know by how much. This uncertainty will be more pronounced immediately after major events (N can change by a factor of 10 or more, which is FAR more than the "doubling" you modeled). It may be necessary to explicitly tell the reader of this shortcoming post-eruptions.

We do not agree with this statement. This is because our analysis of the Wyoming CN record shows that a doubling of the CN concentration accounts for 70-90% of the variation observed, and there are almost no observations that exceed a factor of 10 of the mean, especially during the SCIAMACHY years.

We have added the following text based on new simulations that have been done:
The SCIAMACHY retrieval uses assumptions that lead to errors in the derived aerosol characteristics. Radiative transfer simulations where the a priori and true N differ by up to a factor of four imply uncertainties of +-30% ($r_g$), +-8% ($\sigma_g$), and +-20% ($r_{eff}$) during volcanically quiescent and some post-eruption periods. However, uncertainties in volcanic plumes can also double depending on the PSD. Ext has an uncertainty of +-25% during volcanically quiescent periods and of -70 - 50% during post-eruption periods.

– page 18, lines 483–484: In what way is this "striking"?

We have clearly identified the volcanic signatures in the aerosol characteristics. We have consistent results for different volcanoes. The aerosol characteristics have a clear dependence on the eruption strength, as shown in the comparison of, e.g., Soufrière Hills with Nabro. Such eruptions are also unlikely to cause large perturbations in the total particle number density. Few eruptions do. So the uncertainty in number density is not a large and significant factor.

We have changed the sentence to:
The changes in the aerosol characteristics after volcanic eruptions are readily identified.

Fig. 2 showed that underestimating number density (yellow line) results in over estimation of particle size and under estimation of distribution width. This results in an over estimation of approx 100% in the extinction coefficient and an over estimation of approx 20% for $r_e$. Undoubtedly, the a priori N value in your model is too low after these eruptions, which puts you squarely in the situation I just described (i.e., over estimation of $r_g$, over estimation of extinction, etc.).

We apologize for a coding error. After correction, the overestimation of Ext is ~ 25 %. The a priori N value in our model is too low after volcanic eruptions. This does not necessarily result in a larger overestimation of Ext. Uncertainties in Ext depend not only on the underestimation of N but on uncertainties of all three PSD parameters. Underestimation of Ext is also possible as can be seen in, e.g., Fig. 6.

I don't doubt that extinction increased, I don't doubt that particles became bigger, and I don't doubt that distribution width decreased. However, particles do not always get bigger after eruptions as some of your co-authors have demonstrated (https://acp.copernicus.org/articles/23/9725/2023/). Therefore, this leaves the reader wondering how much of the variabilty shown in Fig. 3 is a by-product of a wrong number density. Given the level of uncertainty in this method I do not believe that you can say that your data unambiguously proves (much less quantified) these changes occurred. Undoubtedly changes are expected, but at this point I think that the most defenceable statement that can be made, based on your product, is that things changed. . . by some amount.
– page 19, Figure 4 caption: The OPC record reports 2 modes (fine and coarse mode). Did you use both modes in calculating the extinction coefficient? The coarse mode can have a disproportionate impact on extinction.

We assumed a single mode lognormal PSD to calculate Ext. This assumption does account for the coarse mode because the entire distribution is used to optimize the fitted three lognormal parameters. We could also compare the Ext with those calculated using the full bimodal distributions, but we doubt this would change our understanding significantly.

– page 19, Figure 4: Panels (d) and (i) do not make sense. Why do the red and blue lines cross each other at ≈18 km at not at ≈20 km (i.e., where the 2 a priori N lines cross each other in panels (c) and (h))? All other panels have the red/blue intersection at the same altitude so why are (d) and (i) different?

Thank you for being thorough. We have checked the Ext profiles (panels d and i) again, their calculation is correct. Calculating the normalized extinction coefficient (Êxt = Ext/N with N as the total number density) will result in red and blue lines crossing each other at ~20 km. Below 20 km, the red line (Êxt using N_Wyoming) is lower than the blue line (Êxt using N_ECSTRA). These profiles are then multiplied by the respective N profiles. Where N_Wyoming > N_ECSTRA. The absolute values depend on the size of factors leading to an intersection of the red and blue lines at ~18 km.

– page 19, Figure 4 caption: What about the light red and light blue colors? Can you define those here so the reader need not search the text for the explanation?

We have revised Figure 4. Light red and light blue colors have been eliminated by shading areas.

– page 20, lines 522–524: "Remarkable are the similar profile shapes from SCIAMACHY and OPC on 7 November 2009 in case of $r_g$ and $\sigma_g$ by assuming the a priori N based on balloon-borne measurements. This is due to the similarity of the SCIAMACHY-assumed and OPC-measured N profiles." This is no surprise (as the authors state, this is due to the extreme similarity between the current OPC N profile and the climatology, that was based on OPC data). What this tells me, yet again, is that the profile is entirely dependent on the a priori N profile.

This is certainly true as it has been pointed out in the paper already. No remote sensing instrument provides the true N profile, in fact there are only a few in situ instruments that can do that, one balloon-borne and 2-3 aircraft borne. Consequently, the SCIAMACHY retrieval algorithm requires an assumed number density profile.

– page 20, lines 526–527: "good agreement of the extinction coefficient from OPC and SCIAMACHY, regardless of the assumed a priori number density (Fig. 4(d,i))" I disagree on the interpretation of this figure, but I admit that I am struggling to find an interpretation and more information would be helpful. Panel (d) certainly looks promising (SCIAMACHY and OPC are in good agreement) and I would be surprised if it were bad. However, panel (i) is less impressive. While the shapes are in good agreement the OPC extinction is larger by a factor of 2-3 (is this what is meant by good agreement)?

We disagree with the statement that SCIAMACHY and OPC Ext differ by 200 – 300 %. The difference is < 33 % in Fig 4d and < 43 % in Fig 4 i. However, it could indeed be argued that this is not a „good" agreement. We have revised the relevant sentence:

One central statement of Fig. 4 is the agreement between the extinction coefficients from OPC and SCIAMACHY, with deviations within 33 % (Fig. 4d) and 43 % (Fig 4i), regardless of the assumed a priori number density.

Undoubtedly all of this variability is driven by differing number densities. However, it is important for the reader to understand how the OPC extinction coefficients were calculated here: did you only use the first mode or both modes? If you only used the first mode then, especially after an eruption, we

can reasonably expect the second/coarse mode to be enhanced and have a disproportionate impact on extinction. Here is the point: currently the difference between SCIAMACHY ext and OPC ext is 200%-300%, if you include the second OPC mode in calculating ext the difference will become larger and this does not qualify as "good agreement".

We use monomodal PSDs derived from OPC data to calculate Ext. We add the following sentence in Sect. 5.1:
The unimodal PSDs are used for comparison with SCIAMACHY-derived aerosol characteristics. From these, the aerosol extinction coefficients are calculated according to Eq. 6.

– page 20, line 528: ". . . the three PSD parameters remain consistent with each other." I do not understand what is meant by "consistent." Can you please clarify?

We have changed the sentence:
It can be explained by the strong correlation of the PSD parameters. For example, if N is overestimated, $r_g$ and $\sigma_g$ change accordingly and to a certain extent. This enables the correct calculation of aerosol characteristics such as Ext from the combination of all three PSD parameters.

– page 21, Figure 5 caption (and corresponding text): ". . . relative errors. . . " This assumes that the OPC is correct, which it is not. You stated on the previous page that disagreement between OPC/SCIAMACHY was driven, in part, to differing sampling volumes and because the OPC only sampled the edge of the aerosol plume while some of the SCIAMACHY profiles were collected within the plume. Should be reworded to "percent difference" (or something comparable) here and throughout the text.

Done. We have replaced „relative errors" by „relative differences".

– page 21, lines 539–540: "Since SCIAMACHY is not sensitive to stratospheric aerosols with $r_g$ ! 0.06 µm (Malinina et al., 2019), corresponding OPC PSDs are excluded from the comparison" This is not correct and these smaller $r_g$'s should be included. They will not contribute much to extinction at 750 nm, so no big deal, but, as written, you are not doing a valid comparison.

SCIAMACHY is not sensitive to aerosols smaller than $r_g = 0.06$ um. This fact is considered in the retrieval algorithm by using a lower threshold in the particle size retrieval: The mode radius cannot be smaller than 0.05 um. Accordingly, for the retrieved sig_g, this corresponds to a lower $r_g$ limit of 0.06 um.
If OPC-measured aerosol characteristics with $r_g$ lower than 0.06 um are compared with corresponding SCIAMACHY-retrievals, there will be discrepancies which are only caused by the user-defined retrieval constraints. This comparison will be meaningless and could be misleading for the reader. For this reason, we exclude the small aerosols.

We have revised the corresponding sentence:
Radiation scattered by aerosols with PSDs of $r_g<0.06$ um is below the sensitivity limit of SCIAMACHY (Malinina et al., 2019). Corresponding OPC measurements and collocated SCIAMACHY retrievals are therefore not in the

comparison.

– page 21, lines 547–548: ". . . Ext depend only slightly on the choice of the a priori N profile." This is not what Fig. 2 tells me; N is highly important.

This statement just refers to Fig. 5i) where it is obvious from the overlaying of the red and blue areas and lines that the initial choice has little impact.

– page 22, lines 566–567: ". . . to obtain a sufficient number of collocations." Sufficient for what? You now have 4255 coincident profiles. . . how many did you have when the maximum distance was only 200 km?

If the maximum distance was only 200 km, there will be ~ 1500 collocated SAGE II and SCIAMACHY profiles.
We have revised the respective sentence:
For SAGE II, the maximum distance from SCIAMACHY is increased to 500 km. This yields 4 255 collocations...

– page 23, Figure 6: This behavior is exactly what I expected based on Figure 2.
– page 24, line 585: "Discrepancies are smaller in the middle latitudes. . . " Yes, this makes sense because these latitudes are the least impacted by volcanic activity. Therefore, your a priori N profile is more similar to the N profile of these latitudes.

We added the explanation:
The differences are smaller in the middle latitudes (~30-50°N/S) below 25 km altitude because the stratospheric aerosols in these altitudes and regions are impacted to a lesser extent by volcanic activity than in other latitudes.

– page 34, lines 774–775: "The retrieved median radii and geometric standard deviations should therefore be considered with caution in areas with high aerosol loading." This is true for extinction as well.

We revised the sentence:
The retrieved and calculated aerosol characteristics should therefore be considered with caution in areas with high aerosol loading.